

**Change in Frozen Soils and Its Effect on Regional Hydrology in the**
**Upper Heihe Basin, on the Northeastern Qinghai-Tibetan Plateau**
Bing Gao[1], Dawen Yang[2]*, Yue Qin[2], Yuhan Wang[2], Hongyi Li[3], Yanlin Zhang[3], and
Tingjun Zhang[4]
[1] School of Water Resources and Environment, China University of Geosciences,
Beijing 100083, China
[2] State Key Laboratory of Hydroscience and Engineering, Department of Hydraulic
Engineering, Tsinghua University, Beijing 100084, China
[3] Cold and Arid Regions Environmental and Engineering Research Institute, Chinese
Academy of Sciences, Lanzhou, Gansu 730000, China
[4] Key Laboratory of West China's Environmental Systems (MOE), College of Earth
and Environmental Sciences, Lanzhou University, Lanzhou, 730000, China
*Correspondence to*:   Dawen Yang (yangdw@tsinghua.edu.cn)



**ABSTRACT:**
Frozen ground has an important role in regional hydrological cycles and ecosystems,
especially on the Qinghai-Tibetan Plateau, which is characterized by high elevation and
a dry climate. This study modified a distributed physically based hydrological model
and applied it to simulate the long-term (from 1971 to 2013) change of frozen ground
and its effect on hydrology in the upper Heihe basin located in the northeastern Qinghai-
Tibetan Plateau. The model was validated carefully against data obtained from multiple
ground-based observations. Based on the model simulations, we analyzed the changes
of frozen soils and their effects on the hydrology. The results showed that the permafrost
area shrank by 9.5% (approximately 600 km$^2$), especially in areas with elevation
between 3500 m and 3900 m. The maximum frozen depth of seasonally frozen ground
decreased at a rate of approximately 5.2 cm/10yr, and the active layer depth over the
permafrost increased by about 3.5 cm/10yr. Runoff increased significantly during cold
seasons (November-March) due to the increase in liquid soil moisture caused by rising
soil temperature. Areas where permafrost changed into seasonally frozen ground at high
elevation showed especially large changes in runoff. Annual runoff increased due to
increased precipitation, the base flow increased due to permafrost degradation, and the
actual evapotranspiration increased significantly due to increased precipitation and soil
warming. The groundwater storage showed an increasing trend, which indicated that
the groundwater recharge was enhanced due to the degradation of permafrost in the
study area.
**KEYWORDS:** permafrost; seasonally frozen ground; soil moisture; soil temperature;





runoff





## 1. Introduction

Global warming has led to significant changes in frozen soils, including both permafrost and seasonally frozen ground at high latitudes and high altitudes (Hinzman et al., 2013; Cheng and Wu, 2007). Changes in frozen soils can greatly affect the land-atmosphere interaction and the energy and water balances of the land surface (Subin et al., 2013; Schuur et al., 2015), altering soil moisture, water flow pathways and stream flow regime (Walvoord and Kurylyk, 2016). Understanding the changes in frozen soils and their impact on regional hydrology is important for water resources management and ecosystem protection in cold regions.

Previous studies based on either the experimental observations or long-term meteorological or hydrological observations have examined changes in frozen soils and their impacts on hydrology. Several studies reported that permafrost thawing might enhance base flow in the Arctic and the Subarctic (Walvoord and Striegl, 2007; Jacques and Sauchyn, 2009; Ye et al., 2009) and in northeast China (Liu et al., 2003; Duan et al., 2017). A few studies reported that permafrost thawing might reduce the river runoff, especially in the Qinghai-Tibetan Plateau (e.g. Qiu, 2012; Jin et al., 2009). Field experiments were usually carried out at small spatial scales over short periods, which lacked the regional pattern and long-term trends of the frozen soils, and the long-term meteorological and hydrological observations did not provide detailed data on soil freezing and thawing processes (McClelland et al., 2004; Liu et al., 2003; Niu et al., 2011). Therefore, the previous observation-based studies have not provided a sufficient understanding of the long-term changes in frozen soils and their impact on regional





hydrology (Woo et al., 2008).
Hydrological models have been coupled with soil freezing-thawing schemes to
simulate impacts of the changes in frozen soils on catchment hydrology. Several
hydrological models (Rawlins et al., 2003; Chen et al., 2008) used simple freezing-
thawing schemes, which could not simulate the vertical soil temperature profiles. The
SiB2 model (Sellers et al., 1996), the modified VIC model (Cherkauer and Lettenmaier,
1999) and the CLM model (Oleson et al., 2010) simulate the vertical soil freezing-
thawing processes, but they represent the hydrological processes especially the flow
routing at the catchment scale, in overly simple ways. Subin et al. (2013) and Lawrence
et al. (2015) used the CLM model to simulate the global change of permafrost. Cuo et
al. (2015) used the VIC to simulate frozen soil degradation and its hydrological impacts
at the plot scale in the headwater of the Yellow River. The GEOtop model (Endrizzi et
al., 2014) simulates three-dimensional water flux and vertical heat transfer in soil, but
it is difficult to apply to regional scales. Wang et al. (2010) and Zhang et al. (2013)
incorporated frozen soil schemes in a distributed hydrological model and showed
improved performance in a small mountainous catchment. More regional studies are
necessary for better understanding of the frozen soil changes and their impacts on the
regional hydrology and water resources.
The Qinghai-Tibetan Plateau is known as Asia's water tower, and runoff changes
on the plateau have significant impacts on water security in the downstream regions
(Walter et al., 2010), which have received an increasing amount of attention in recent
years (Cuo et al., 2014). Hydrological processes on the Qinghai-Tibetan Plateau, which



is characterized by high elevation and cold climate are greatly influenced by
cryospheric processes (Cheng and Jin, 2013; Cuo et al., 2014). In contrast with the
Arctic and Subarctic, the permafrost thickness on the Qinghai-Tibetan Plateau is
relatively thin and warm, and the frozen depth of the seasonally frozen soils is also
relatively shallow. As a result, the frozen soils on the Qinghai-Tibetan Plateau are more
sensitive to air the temperature rising (Yang et al., 2010), and the changes of frozen
soils may have more significant impacts on regional hydrology.

An evident increase in the annual and seasonal air temperature has been observed

in the Qinghai-Tibetan Plateau (Li et al., 2005; Liu and Chen, 2000; Zhao et al., 2004).
Several studies have shown the changes of frozen soils based on long-term observations.
For example, Cheng and Wu (2007) analyzed the borehole observations of soil
temperature profiles on the Qinghai-Tibetan Plateau and found that the active layer
thickness of frozen soils increased by 0.15-0.50 m during the period of 1996-2001.
Zhao et al. (2004) found a decreasing trend of freezing depth in the seasonally frozen
soils using observations at 50 stations. Several studies have analyzed the relationship
between the change of frozen soils and river discharge using the observed data (Zhang
et al., 2003; Jin et al., 2009; Niu et al., 2011). However, the spatio-temporal
characteristics of the long-term change in frozen soils are not sufficiently clear. Having
comprehensive experiments carried out by a major research plan, titled "Integrated
research on the ecohydrological processes of the Heihe basin" funded by the National
Natural Science Foundation of China (NSFC) (Cheng et al., 2014), a hydrological
model coupling cryospheric processes and hydrological processes has been developed

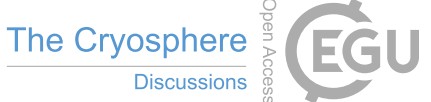

(Gao et al., 2016). This provides a solid basis upon which to analyze the spatio-temporal
changes in frozen soils and their impacts on the regional hydrology in the upper Heihe
basin located on the northeastern Qinghai-Tibetan Plateau.

On the basis of the previous studies, this study aims to: (1) explore the spatial and

temporal changes of frozen soils using a distributed hydrological model with
comprehensive validation and (2) analyze the hydrological responses to the change of
frozen soils during the past 40 years in the upper Heihe basin.
**2.    Study Area and Data**
**2.1 The Heihe River and the upper Heihe basin**
The Heihe River is one of the major inland basins in northwestern China. As shown in
Figure 1, the upper reaches of the Heihe River are located on the northeastern Qinghai-
Tibetan Plateau at an elevation of 2200 to 5000 m with a drainage area of 10,009 km$^2$.
The upper reaches provide the majority of the water supplied to the middle and lower
reaches (Cheng et al., 2014). The annual precipitation in the upper Heihe basin ranges
from 200 to 700 mm, and the annual mean air temperature ranges from -9 to 5℃.
Permafrost dominates the high elevation region above 3700 m (Wang et al., 2013), and
seasonal frozen ground covers other parts of the study area. Glaciers are found at an
elevation above 4000 m, covering approximately 0.8% of the upper Heihe basin. There
are two tributaries (East and West Tributaries) in the upper Heihe basin, on which two
hydrological stations are located, namely, Qilian (on the east tributary) and Zhamashike
(on the west tributary). The outlet of the upper Heihe basin has a hydrological station,
namely, Yingluoxia (see Figure 1).





## 2.2 Data used in the study

### (1) Forcing data of the hydrological model

The atmospheric forcing data used to drive the hydrological model include a 1-km gridded dataset of daily precipitation, air temperature, sunshine hours, wind speed and relative humidity. The gridded daily precipitation was interpolated from observations at meteorological stations (see Figure 1) provided by the China Meteorological Administration (CMA) using the method developed by Wang et al. (2017). The other atmospheric forcing data were interpolated by observations at meteorological stations using the inverse distance weighted method. The interpolation of air temperature considers the temperature gradient with elevation which was provided by the HiWATER experiment (Li et al., 2013).

The land surface data used to build the model include land use, topography, leaf area index, and soil parameters. The topography data were obtained from the SRTM dataset (Jarvis et al., 2008) with a spatial resolution of 90 m. The land use/cover data were provided by the Institute of Botany, Chinese Academy of Sciences (Zhou and Zheng, 2014). The leaf area index (LAI) data with 1-km resolution were developed by Fan (2014). The soil parameters were developed by Song et al. (2016); they include the saturated hydraulic conductivity, residual soil moisture content, saturated soil moisture content, soil sand matter content, soil clay matter content and soil organic matter content.

### (2) Data used for model calibration and validation

This study uses the observed daily river discharge data from the Yingluoxia, Qilian



and Zhamashike stations, the daily soil temperature of different depths from the Qilian
station and the daily frozen depths from the Qilian and Yeniugou stations for model
calibration and validation. Daily river discharge data were obtained from the Hydrology
and Water Resources Bureau of Gansu Province. Daily soil temperature data collected
at the Qilian station from January 1, 2004 to December 31, 2013, and daily frozen depth
data collected at the Qilian and Yeniugou stations from January 1, 2002 to December
31, 2013 were provided by CMA.
To investigate the spatial distribution of permafrost, boreholes were drilled during
the NSFC major research plan. Temperature observations from six boreholes, whose
location are shown in Figure 1, were provided by Wang et al. (2013). The borehole
depths are 100 m for T1, 69 m for T2, 50 m for T3, 90 m for T4, and 20 m for T5 and
T7. Monthly actual evapotranspiration data with 1-km resolution during the period of
2002-2012 estimated based on remote sensing data (Wu et al., 2012; Wu, 2013) were
used to evaluate the model-simulated evapotranspiration. We also used field
observations of the hourly liquid soil moisture to validate the model simulation of soil
moisture profiles. The HiWATER experiment (Li et al., 2013; Liu et al., 2011) provided
the soil moisture data from January 1 to December 31, 2014 at the A'rou Sunny Slope
station (100.52 E, 38.09 N).
**3.   Methodology**
**3.1 Brief introduction of the hydrological model**
This study used a distributed eco-hydrological model GBEHM (geomorphology-based
ecohydrological model), which was developed in an integrated research project under





the major research plan "Integrated research on the ecohydrological process of the
Heihe River Basin" (Yang et al., 2015; Gao et al., 2016) based on the geomorphology-
based hydrological model (Yang et al., 1998 and 2002; Cong et al., 2009). As shown in
Figure 2, the GBEHM used a 1-km grid system to discretize the study catchment, and
the study catchment was divided into 251 sub-catchments. A sub-catchment was further
divided into flow-intervals along its main stream. To capture the sub-grid topography,
each 1-km grid was represented by a number of hillslopes with an average length and
gradient, but different aspect, which were estimated from the 90-m DEM. The terrain
properties of a hillslope include the slope length and gradient, slope aspect, soil type
and vegetation type (Yang et al., 2015).

The hillslope is the basic unit for the hydrological simulation, upon which the water

and heat transfers (both conduction and convection) in the vegetation canopy,
snow/glacier, and soil layers are simulated. The canopy interception, radiation transfer
in the canopy and the energy balance of the land surface are described using the
methods used in SIB2 (Sellers et al., 1985, 1996). The surface runoff on the hillslope is
solved using the kinematic wave equation. The groundwater aquifer is considered as
individual storage units corresponding to each grid. Exchange between the groundwater
and the river water is calculated using Darcy's law (Yang et al., 1998, 2002).

The model runs with a time step of 1 hour. Runoff generated from the grid is the

lateral inflow into the river at the same flow interval in the corresponding sub-
catchment. Flow routing in the river network is calculated using the kinematic wave
equation following the sequence determined by the Horton-Strahler scheme (Strahler,



1957).

### 3.2 Simulation of cryospheric processes

The simulation of cryospheric processes in GBEHM includes glacier ablation, snow
melt, and soil freezing and thawing.
(1) Glacier ablation
Glacier ablation is simulated using an energy balance model (Oerlemans, 2001) as:

$$Q_M = SW(1-\alpha) + LW_{in} - LW_{out} - Q_H - Q_L - Q_G + Q_R \qquad (1)$$

where $Q_M$ is the net energy absorbed by the surface of the glacier (W/m$^2$); $SW$ is the
incoming shortwave radiation (W/m$^2$); $\alpha$ is the surface albedo; $LW_{in}$ is the incoming
longwave radiation (W/m$^2$); $LW_{out}$ is the outgoing longwave radiation (W/m$^2$); $Q_H$ is
the sensible heat flux (W/m$^2$); $Q_L$ is the latent heat flux (W/m$^2$); $Q_R$ is the energy from
rainfall (W/m$^2$); and $Q_G$ is the penetrating shortwave radiation (W/m$^2$). The surface
albedo is calculated as (Oerlemans and Knap, 1998):

$$\alpha = \alpha_{snow} + (\alpha_{ice} - \alpha_{snow})e^{-h/d^*} \qquad (2)$$

where $\alpha_{snow}$ is the albedo of snow on the glacier surface; $\alpha_{ice}$ is the albedo of the ice
surface; $h$ is the snow depth on the glacier surface (m); $d^*$ is a parameter of the snow
depth effect on the albedo (m).
The amount of melt water is calculated as (Oerlemans, 2001):

$$M = \frac{Q_M}{L_f} dt \qquad (3)$$

where $dt$ is the time step used in the model (s) and $L_f$ is the latent heat of fusion (J/kg).
(2) Snow melt
A multi-layer snow cover model is used to describe the mass and energy balance of

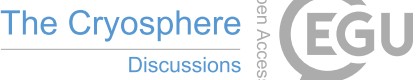



snow cover. The parametrization of snow is based on Jordan (1991), and each snow
layer is described by two constituents, namely, ice and liquid water. For each snow layer,
temperature is solved using an energy balance approach (Bartelt and Lehnin, 2002):
$$C_s \frac{\partial T_s}{\partial t} - L_f \frac{\partial \rho_i \theta_i}{\partial t} = \frac{\partial}{\partial z}(K_s \frac{\partial T}{\partial z}) + \frac{\partial I_R}{\partial z} + Q_R \qquad (4)$$

where $C_s$ is the heat capacity of snow (J·m$^{-3}$·K$^{-1}$); $T_s$ is the temperature of the snow
layer (K); $\rho_i$ is the density of the ice (kg/m$^3$); $\theta_i$ is the volumetric ice content;
$K_s$ is the thermal conductivity of snow (W·m$^{-1}$·K$^{-1}$); $L_f$ is the latent heat of ice fusion
(J/kg) ; $I_R$ is the radiation transferred into the snow layer (W/m$^2$) and $Q_R$ is the energy
brought by rainfall (W/m$^2$) which is only considered for the top snow layer. The solar
radiation transfer in the snow layers and the snow albedo are simulated using the
SNICAR model which is solved using the method developed by Toon et al. (1989). Eq.
(4) is solved using an implicit centered finite difference method, and a Crank-Nicholson
scheme is employed.
The mass balance of the snow layer is described as (Bartelt and Lehnin, 2002):
$$\frac{\partial \rho_i \theta_i}{\partial t} + M_{iv} + M_{il} = 0 \qquad (5)$$

$$\frac{\partial \rho_l \theta_l}{\partial t} + \frac{\partial U_l}{\partial z} + M_{lv} - M_{il} = 0 \qquad (6)$$

where $\rho_l$ is the density of the liquid water (kg/m$^3$); $\theta_l$ is the volumetric liquid water
content; $U_l$ is the liquid water flux (kg·m$^{-2}$·s$^{-1}$); $M_{iv}$ is the mass of ice that is changed
into vapour within a time step (kg·m$^{-3}$·s$^{-1}$); $M_{il}$ is the mass of ice that is changed into
liquid water within a time step (kg·m$^{-3}$·s$^{-1}$); and $M_{lv}$ is the mass of liquid water that is
changed into vapour within a time step (kg·m$^{-3}$·s$^{-1}$). The liquid water flux of the snow
layer is calculated as (Jordan, 1991):

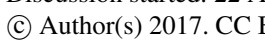



$$U_l = -\frac{k}{\mu_l}\rho_l^2 g \qquad (7)$$


where $k$ is the hydraulic permeability (m$^2$), $\mu_l$ is dynamic viscosity of water at 0 ℃
($1.787\times10^{-3}$ N s/m$^2$), $\rho_l$ is the density of liquid water (kg/m$^3$) and $g$ is gravitational
acceleration (m/s$^2$). The water flux of the bottom snow layer is considered snowmelt
runoff.
(3) Soil freezing and thawing
The energy balance of the soil layer is solved as (Flerchinger and Saxton, 1989):
$$C_s \frac{\partial T}{\partial t} - \rho_i L_f \frac{\partial \theta_i}{\partial t} - \frac{\partial}{\partial z}(\lambda_s \frac{\partial T}{\partial z}) + \rho_l c_l \frac{\partial q_l T}{\partial z} = 0 \qquad (8)$$

where $C_s$ is the volumetric soil heat capacity (J·m$^{-3}$·K$^{-1}$); $T$ is the temperature (K) of
the soil layers; $z$ is the vertical depth of the soil (m); $\theta_i$ is the volumetric ice content;
$\rho_i$ is the density the ice (kg/m$^3$); $\lambda_s$ is the thermal conductivity (W·m$^{-1}$·K$^{-1}$); $\rho_l$ is
the density of liquid water (kg/m$^3$); and $c_l$ is the specific heat of liquid water
(J·kg$^{-1}$·K$^{-1}$). In addition, $q_l$ is the water flux between different soil layers (m/s) and is
solved using the 1-D vertical Richards equation. The unsaturated soil hydraulic
conductivity is calculated using the modified van Genuchten's equation (Wang et al.,
2010) as:
$$K = f_{ice} K_{sat} (\frac{\theta_l - \theta_r}{\theta_s - \theta_r})^{1/2}[1-(1-(\frac{\theta_l - \theta_r}{\theta_s - \theta_r})^{-1/m})^m]^2 \qquad (9)$$

where $K$ is the unsaturated soil hydraulic conductivity (m/s); $K_{sat}$ is the saturated soil
hydraulic conductivity (m/s); $\theta_l$ is the volumetric liquid water content; $\theta_s$ is the
saturated water content; $\theta_r$ is the residual water content; $m$ is an empirical parameter
in van Genuchten's equation and $f_{ice}$ is an empirical hydraulic conductivity reduction



factor which is calculated using soil temperature as (Wang et al., 2010):
$$f_{ice} = \exp[-10(T_f - T_{soil})], \quad 0.05 \leq fice \leq 1 \qquad (10)$$
where $T_f$ is 273.15 K and $T_{soil}$ is the soil temperature.

Eq. (8) solves the soil temperature with the upper boundary condition as the heat flux

into the top surface soil layer. When the ground is not covered by snow, the heat flux
from the atmosphere into the top soil layer is expressed as (Oleson et al., 2010):
$$h = S_g + L_g - H_g - \lambda E_g + Q_R \qquad (11)$$
where $h$ is the upper boundary heat flux into the soil layer (W m$^{-2}$); $S_g$ is the solar
radiation absorbed by the top soil layer (W m$^{-2}$); $L_g$ is the net long wave radiation
absorbed by the ground (W m$^{-2}$), $H_g$ is the sensible heat flux from the ground (W m$^{-2}$);
$\lambda E_g$ is the latent heat flux from the ground (W m$^{-2}$); and $Q_R$ is the energy brought by
rainfall (W/m$^2$). When the ground is covered by snow, the heat flux into the top soil
layer is calculated as:
$$h = I_p + G \qquad (12)$$
where $I_p$ is the radiation that penetrates the snow cover, and $G$ is the heat conduction
from the bottom snow layer to the top soil layer. Eq (8) is solved using a finite difference
scheme with an hourly time step which is similar with the solutions of Eq (4).

To simulate the permafrost we consider an underground depth of 50 m and assume

the bottom boundary condition as zero heat flux exchange. The vertical soil column is
divided into 39 layers in the model (see Figure 2). The topsoil of 1.7 m is subdivided
into 9 layers. The first layer is 5 cm, and the soil layer thickness increases with depth
linearly from 5 cm to 30 cm up to the depths of 0.8 m and later decreases linearly with




depth to 10 cm up to the depths of 1.7 m. There are 12 soil layers from 1.7 m to 3.0 m
with a constant thickness of 10 cm. From the depth of 3 m to 50 m, there are 18 layers
with thickness increasing exponentially from 10 cm to 12 m. The liquid soil moisture,
ice content, and soil temperature of each layer is calculated at each time step. The soil
heat capacity and soil thermal conductivity are estimated using the method developed
by Farouki (1981).
**3.3 Model calibration**
We assume that there is a linear relationship between soil temperature and elevation
at the same depth below surface. The relationship between soil temperature at a specific
depth and elevation is estimated from the observed soil temperature at 6 boreholes (see
Figure 1). For spin up run, the initial soil temperatures at different depths for all grids
of the whole study area were interpolated from the borehole observations using this
relationship. Next, the model had a 500 year spin up run to specify the initial values of
the hydrological variables (e.g., soil moisture, soil temperature, soil ice content, and
groundwater table) by repeating the atmospheric forcing data from 1961 to 1970.
The period of 2002 to 2006 was used for model calibration and the period of 2008 to
2012 was for model validation. The daily soil temperature at the Qilian station and the
frozen depths at the Qilian and Yeniugou stations were used to calibrate the soil
reflectance according to vegetation type. The other parameters such as groundwater
hydraulic conductivity were calibrated according to the baseflow discharge in the
winter season. We calibrated the surface retention capacity and surface roughness to
match the observed flood peaks, and calibrated the leaf reflectance, leaf transmittance



and maximum Rubsico capacity of the top leaf based on the remote sensing
evapotranspiration data. Table 1 shows the major parameters used in the model.
**4.   Results**
**4.1 Validation of the hydrological model**

We carried out a comprehensive validation of the GBEHM model using the soil

temperature profiles observed at six boreholes, long-term observations of the soil
temperature and frozen depths at two CMA stations, soil moisture observations at one
HiWATER station, long-term observations of streamflow at three hydrological stations
and monthly actual evapotranspiration estimated from remote sensing data.

Figure 3 shows the comparison of the model-simulated and observed soil

temperature profiles at six boreholes. The model was generally accurate in capturing
the vertical distribution of the soil temperature at T1, T2, T3 and T4 in the permafrost
area, but overestimations were produced above 20 m depth for T1 and T3. Good
agreement between the simulated and observed soil temperature profiles below the
depth of 20 m implies that the temperature in the deep soil is stable, which is confirmed
by the comparison of temperature profiles in different years as shown in Figure S1 in
the supplemental file. Figure S1 also illustrates that temperature above 20 m shows
significant increasing trends in the past 40 years. The errors in simulating the vertical
temperature profile near the surface might be caused by simplification of the 3-D
topography. At T5 located in seasonally frozen ground, the simulated soil temperature
profile did not agree well with that observed at depth of 4-20 m. This error might also
be related to the heterogeneity of soil properties, especially the thermal conductivity

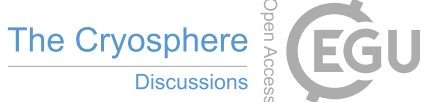

and heat capacity since no such information is available. The model simulation agrees
well with the borehole observation at T7, which is located at the transition zone from
permafrost to seasonally frozen ground. This indicates that the model can identify the
boundary of the permafrost and the seasonally frozen ground.
We also validated model simulation of the freezing/thawing cycles based on long-
term observations of soil temperature and frozen depth. Figure 4 compares the
simulated soil temperature with the observed temperature at the Qilian station, which
is located in the seasonally frozen ground (observed daily soil temperature data are
available from 2004 on). Generally, the model simulations accurately captured the
seasonal changes in soil temperature profile. Validation of the soil temperature at
different depths (5 cm, 10 cm, 20 cm, 40 cm, 80 cm, 160 cm, and 320 cm) showed that
the root mean square error (RMSE) decreases with increasing depth. The RMSE were
approximately 2.5℃ for the top three depths (5 cm, 10 cm and 20 cm). The RMSE
for depths of 40 cm and 80 cm were 1.7℃ and 1.5℃, respectively, and the RMSE was
0.9℃ at a depth of 3.2 m. We compared the model-simulated daily frozen depth with
in situ observations at the Qilian and Yeniugou Stations from 2002 to 2014, as shown
in Figure 5. The model reproduced well the daily variations in frozen depth although
the depth was underestimated by approximately 50 cm at the Yeniugou station. In
general, the validation of soil temperature and frozen depth indicates that the model
captured well the freezing and thawing processes in the upper Heihe basin.
The observed hourly liquid soil moisture at the A'rou Sunny Slope station was used
for an additional independent validation. Figure 6 shows the comparison between the



simulated and observed liquid soil moisture at different depths from January 1 to
December 31 in 2014. By comparing with the observed liquid soil moisture, we can see
that the model simulation is reasonable.
Figure 7 compares the model simulated and the observed daily streamflow discharge
at the Yingluoxia, Qilian and Zhamashike station. The model simulation agreed well
with the observations. The model simulation captured the flood peaks and the
magnitude of base flow in both of the calibration and validation periods. In the
calibration period, the Nash-Sutcliffe efficiency (NSE) coefficients were 0.64, 0.65 and
0.70 for the Yingluoxia, Qilian and Zhamashike stations, respectively; in the validation
period, the NSE values were 0.65, 0.60, and 0.75, respectively. The relative error (RE)
was within 10% for both the calibration and validation periods (see Figure 7). Figure 8
shows the comparison of the model-simulated monthly actual evaporation and remote
sensing-based evaporation data for the entire calibration and validation periods. The
GBEHM simulation showed similar temporal variations in actual evapotranspiration
compared with the remote sensing based estimation, and the RMSE of the simulated
monthly evapotranspiration was 8.0 mm in the calibration period and 6.3 mm in the
validation period.
The model simulated river discharges with and without the frozen soil scheme were
compared. Table S1 in the supplement material shows that model with the frozen soil
scheme achieves better simulation of the daily hydrograph than the model without the
frozen soil scheme. Figure S2 in the supplement material shows that the model without
the frozen soil scheme overestimated the river discharge in the freezing season and



underestimated flood peaks in the warming season.
**4.2 Long-term changes in frozen soils**

In the upper Heihe basin, the ground surface starts freezing in November and thawing

initiates in April (Wang et al., 2015a). From November to March, the ground surface
temperature is below 0℃ in both the permafrost and seasonally frozen ground regions,
and precipitation mainly falls in the period from April to October. Therefore, a year is
subdivided into two seasons, i.e., the freezing season (November to March) and the
thawing season (April to October) to investigate the changes in frozen soils and their
hydrological impact. Increasing precipitation and air temperature in the study area in
both seasons in the past 50 years was reported in a previous study (Wang et al., 2015b).

Figure 9 shows the changes in the basin-averaged soil temperature in the freezing

and thawing seasons. The soil temperature increased in all seasons, especially in the
past 30 years. The increasing trend of soil temperature was larger in the freezing season
than in the thawing season. In the freezing season (Figure 9(a)), the top layer soil
temperature was lower than the deep layer soil temperature. The linear trend of the top
layer (0-0.5 m) soil temperature was 0.48℃/10yr and the trend of the deep layer (2.5-3
m) soil temperature was 0.34℃/10yr. The soil temperature in the deep layer (2.5-3 m)
changed from -1.1℃ in the 1970s to approximately 0℃ in the most recent decade. In
the thawing season (see Figure 9(b)), the increasing trend of the top layer (0-0.5 m) soil
temperature (0.29℃/10yr) was greater than the trend of the deep layer (2.5-3 m) soil
temperature (0.21℃/10yr). The warming trend is larger in shallow soils and this is
because the surface heat flux is impeded by the thermal inertia as it penetrates to greater



depths.

Permafrost is defined as ground with a temperature at or below 0℃ for at least two

consecutive years (Woo, 2012). This study differentiated permafrost from seasonally
frozen ground based on the simulated vertical soil temperature profile in each grid. For
each year in each grid, the frozen ground condition was determined by searching the
soil temperature profile within a four-year window from the previous three years to the
current year. Figure 10 shows the change in permafrost area during 1971-2013. As
shown in Figure 10(a), the permafrost areas decreased by approximately 9.5% (from
6445 km$^2$ in the 1970s to 5831 km$^2$ in the 2000s), indicating evident degradation of the
permafrost in the upper Heihe basin in the past 40 years.

Figure 10 (b) shows the changes in the basin-averaged maximum frozen depth for

the seasonally frozen ground areas and active layer thickness over the permafrost areas.
The basin-averaged annual maximum frozen depth showed a significant decreasing
trend (5.2 cm/10yr). In addition, the maximum frozen depth had a significantly negative
correlation with the annual mean air temperature ($r$ = -0.73). In contrast, an increasing
trend of active layer thickness in the permafrost regions was observed (3.5 cm/10yr),
which had a significantly positive correlation with the annual mean air temperature.

Figure 11 shows the frozen soil distributions in the period of 1971 to 1980 and in the

period of 2001 to 2010. Comparing the frozen soil distributions of the two periods,
major changes in frozen soils were observed on the sunny slopes at elevations between
3500 and 3700 m, especially in the west tributary, where large areas of permafrost
changed into seasonally frozen ground.





415  Figure 12 shows the monthly mean soil temperature over the areas with elevation

416  between 3300 and 3500 m and over areas with elevation between 3500 and 3700 m in

417  the upper Heihe basin. In the areas with elevation between 3300 and 3500 m located in

418  the seasonally frozen ground region, as shown in Figure 12(a), the frozen depth

419  decreased and the soil temperature in the deep layer (with depth greater than 2 m)

420  increased. Figure 12(b) shows that the increase in soil temperature was larger in the

421  area with higher elevation (3500-3700 m). This figure shows that the thickness of the

422  permafrost layer decreased as soil temperature increased, and the permafrost changed

423  into seasonally frozen ground after 2000.

424  **4.3 Changes in the water balance and runoff**

425  Table 2 shows the decadal changes in the annual water balance from 1971 to 2010

426  based on the model simulation. The annual precipitation, annual runoff and annual

427  runoff ratio had the same decadal variation; however the annual evapotranspiration

428  maintained an increasing trend since the 1970s which was consistent with the rising air

429  temperature and soil warming. Although the actual evapotranspiration increased, the

430  runoff ratio remained stable during the 4 decades because of the increased precipitation.

431  The changes in runoff (both simulated and observed) in different seasons are shown

432  in Figure 13 and Table 2. The model-simulated and observed runoff both showed a

433  significant increasing trend in the freezing season and in the thawing season. This

434  indicates that the model simulation accurately reproduced the observed long-term

435  changes. In the freezing season, since there was no glacier melt and snow melt (see

436  Table 2), runoff was mainly the subsurface flow (groundwater flow and lateral flow





from the unsaturated zone). In the thawing season, as shown in Table 2, snowmelt
runoff contributed approximately 16% of the total runoff and glacier runoff contributed
only a small fraction of total runoff (approximately 2.4%). Therefore, rainfall runoff
was the major component of total runoff in the thawing season, and the runoff increase
in the thawing season was mainly due to increased rainfall. As shown in Figure 13, the
actual evapotranspiration increased significantly in both seasons due to increased
precipitation and soil warming. The increasing trend of the actual evapotranspiration
was higher in the thawing season than in the freezing season.
Figure 14 shows the changes in the basin-averaged annual water storage in the top
0-3 m layer and the groundwater storage. The annual liquid water storage of the top 0-
3 m showed a significant increasing trend especially in the most recent 3 decades. This
long-term change in liquid water storage was similar to the runoff change in the freezing
season, as shown in Figure 13 (a), with a correlation coefficient of 0.80. The annual ice
water storage in the top 0-3 m soil showed significant decreasing trend due to frozen
soils changes. Annual groundwater storage showed a significantly increasing trend
especially in the most recent 3 decades, which indicates that the groundwater recharge
increases with the frozen soil degradation.
**5.   Discussion**
**5.1 Impact of frozen soil changes on the soil moisture and runoff**
Based on the model simulated daily soil moisture, long-term changes of the spatially
averaged liquid soil moistures in the region with elevation between 3300 and 3500 m
(covered by the seasonally frozen ground) and in the region with elevation between



3500 and 3700 m (where the permafrost changed into seasonally frozen ground) are
shown in Figure S3 in the supplement material. In the seasonally frozen ground with
elevation of 3300-3500 m, by comparing with the soil temperature shown in Figure 12
(a), we can see that the liquid soil moisture increase was mainly caused by the decrease
in the frozen depth. The liquid soil moisture in the deep soil layer increased significantly
since the 1990s in the area with elevation of 3500-3700 m where the permafrost
changed to seasonally frozen ground. Compared with the soil temperature change
shown in Figure 12 (b), the liquid soil moisture increases in this region were primarily
caused by the change of permafrost to seasonally frozen ground, indicating that the
frozen soil degradation caused a significant increase in liquid soil moisture in both the
freezing and thawing seasons.
In the freezing season, since the surface ground is frozen, runoff is mainly subsurface
flow coming from the seasonally frozen ground. Runoff has the highest correlation
(r=0.82) with the liquid soil moisture in the freezing season, which indicates that the
frozen soils change was the major cause of the increased liquid soil moisture, resulting
in increased runoff in the freezing season. During the past 40 years, parts of the
permafrost changed into seasonally frozen ground, and the thickness of the seasonally
frozen ground decreased, which led to increased liquid soil moisture in the deep layers
during the freezing season. The increase in liquid soil moisture also increased the
hydraulic conductivity which enhanced the subsurface flow.
In the thawing season from April to October, the thickness of the seasonally frozen
ground rapidly decreased to zero and the thaw depth of permafrost reached the



maximum. Runoff in the thawing season was mainly rainfall runoff, as shown in Table
2. The increased runoff mainly came from increased precipitation in the thawing season.
Figure 15 shows the changes in areal mean runoff along the elevation for different
seasons. There was a large difference in runoff variation with the elevation during the
different seasons. In the freezing season, the runoff change from the 1970s to the 2000s
in the areas of seasonally frozen ground (mainly located below 3500 m, see Figure 11)
was relatively small. The areas with elevations of 3500 to 3900 m showed larger
changes in runoff. This is due to the shift from permafrost to seasonally frozen ground
in some areas in the elevation range of 3500 to 3900 m, as simulated by the model,
particularly for the sunny hillslopes (see Figure 11). This finding illustrates that a
change from the permafrost to the seasonally frozen ground has a larger impact on the
runoff than a change in frozen depth in the seasonally frozen ground. In the thawing
season, runoff increased with elevation due to the increase in precipitation with
increasing elevation, and the runoff increase was mainly determined by increased
precipitation (Gao et al., 2016). Precipitation in the region with elevation below 3100
m was low, but air temperature was high. Runoff in this region decreased during 2001-
2010 compared to 1971-1980 because of higher evapotranspiration.
**5.2 Comparison with the previous similar studies**
In this study, the model simulation showed that changes in frozen soils led to
increased freezing season runoff and base flow in the upper Heihe basin. This result is
consistent with previous findings based on the trend analysis of streamflow
observations in high latitude regions (Walvoord and Striegl, 2007; Jacques and Sauchyn,





2009; Ye et al., 2009) and in northeast China (Liu et al., 2003). However, those studies
did not consider spatial variability. This study found that the impact of the change in
frozen soils on runoff had regional characteristics. In the upper Heihe basin (see Figure
15), a change in frozen soils led to increased runoff at higher elevations but led to
decreased runoff at lower elevations during the freezing season. This implies that
change of the freezing season runoff was controlled by the permafrost degradation in
the higher elevation region but by the evaporation increase in the lower elevation region
due to the air temperature rising. However, runoff at the basin scale mainly came from
the higher elevation regions.
This study also showed that the change in frozen soils increased the soil moisture in
the upper Heihe basin, which is consistent with the finding of Subin et al. (2013) using
the CLM model simulation in northern latitude permafrost regions, and the findings of
Cuo et al. (2015) using VIC model simulation at 13 sites on the Tibetan Plateau.
However, Lawrence et al. (2015) found that permafrost thawing caused soil moisture
drying based on CLM model simulations for the global permafrost region. This might
be related to the uncertainties in the soil water parameters and the high spatial
heterogeneity of soil properties, which are difficult to consider in a global-scale model.
Subin et al. (2013) and Lawrence et al. (2015) modelled the soil moisture changes in
the active layer of permafrost in large areas with coarse spatial resolution. This study
revealed the spatio-temporal variability of soil moisture with high spatial resolution and
analyzed the correlations with the change in frozen soils.
Wu and Zhang (2010) focused on the changes in the active layer thickness at 10 sites



in the permafrost region on the Tibetan Plateau and found a significant increasing trend
during the period of 1995-2007, which is consistent with the result of this study. Jin et
al. (2009) found decreased soil moisture and runoff due to the permafrost degradation
based on observations at the plot scale in the source areas in the Yellow River basin.
This result is different from the present study, possibly due to the difference of
hydrogeological structure and the soil hydraulic parameters in the source area of the
Yellow River from those in the upper Heihe basin. Wang et al. (2015a) focused on the
change in the seasonally frozen ground in the Heihe River basin based on plot
observations, and the increasing trend of the maximum frozen depth was estimated as
4.0 cm/10yr during 1972-2006, which is consistent with the GBEHM model simulation
in this study. The increase in groundwater storage illustrated in this study is also
consistent with the finding of Cao et al. (2012) based on the GRACE data which showed
that groundwater storage increased during the period of 2003~2008 in the upper Heihe
basin.
**5.3 Uncertainty in simulation of the frozen soils**
Estimation of the change in permafrost area is a great challenge due to such complex
factors as climatology, vegetation, and geology. Different methods produce large
differences in their estimation results. Jorgenson et al. (2006) found a 4.4% decrease in
the area of permafrost in Arctic Alaska from 1982 to 2001 based on analyses of aerial
photo. Wu et al. (2005) reported that the permafrost area decreased by 12% from 1975
to 2002 in the Xidatan basin, Qinghai-Tibetan Plateau based on a ground penetration
radar survey. Jin et al. (2006) found an area reduction of 35.6% in island permafrost in



Liangdaohe, which is located at the southern Qinghai–Tibet Highway, from 1975 to
1996. Chasmer et al. (2010) found a 30% reduction of the discontinuous permafrost
area in the Northwest Territories, Canada from 1947 to 2008 based on remote sensing.
Compared with the borehole observations by Wang et al. (2013) shown in Figure 2, this
model slightly overestimated the soil temperature in permafrost areas, which might lead
to overestimation of the rate of permafrost area reduction.
There were two major uncertainties in the frozen soils simulation which may lead to
overestimation: uncertainty in the land surface energy balance simulation and
uncertainty in the simulation of the soil heat-water transfer processes (Wu et al., 2016).
Uncertainty in the land surface energy balance simulation might result from the
estimations of radiation and surface albedo due to the complex topography, vegetation
cover and soil moisture distribution, which may introduce uncertainties in the estimated
ground temperature and thermal heat flux into the deep layers. The uncertainty in
simulation of soil heat-water transfer processes might result from the soil water and
heat parameters and the bottom boundary condition of heat flux. Permafrost
degradation is closely related to the thermal properties of rocks and soils, geothermal
flow and initial soil temperature and soil ice conditions. The lack of observed initial
condition data could also cause uncertainty in the permafrost change estimation. For
discontinuous permafrost, lateral heat flux may increase the thawing rate (Kurylyk et
al., 2016; Sjöberg et al., 2016) and this effect is not considered in the present study.
This may lead to underestimation of thawing rates of discontinuous permafrost,
especially in spring. In addition, uncertainties from input data, particularly the solar



radiation which is estimated using interpolated sunshine hour data from limited
observational stations and precipitation which is also interpolated by observations at
these stations, may also influence the results of the model simulation.
**6. Conclusions**
A distributed hydrological model coupled with cryospheric processes was carefully
validated in the upper Heihe River basin using available observations of soil moisture,
soil temperature, frozen depth, actual evaporation and streamflow discharge. Based on
the model simulations from 1971 to 2013 in the upper Heihe River, the long-term
changes in frozen soils were investigated, and the effect of the frozen soils change on
hydrological processes were explored. Based on these analyses, the following
conclusions can be drawn:
(1) The model simulation suggests that 9.5% of permafrost areas degraded into
seasonally frozen grounds in the upper Heihe River basin during the period of 1971 to
2013, which predominantly occurred at the elevations between 3500 m and 3900 m.
The decreasing trend of annual maximum frozen depth is estimated to be 5.2 cm/10yr
for the seasonally frozen grounds, which is consistent with previous observation-based
studies at plot scale. The increasing trend of active layer thickness is estimated to be
3.5 cm/10yr in the permafrost regions.
(2) Model simulated trends in runoff agree with the observed trends. In the freezing
season (November-March), based on the model simulation, runoff was mainly sourced
by subsurface flow which increased significantly in the higher elevation regions where
significant frozen soil changes occurred. This finding implies that runoff increase in the



freezing season is primarily caused by frozen soil changes (permafrost degradation and
decrease of the seasonally frozen depth). In the thawing season (April-October), model
simulation indicates that runoff mainly came from rainfall and showed an increasing
trend at the higher elevations, which can be explained by the increased precipitation. In
both the freezing and thawing seasons, model simulated runoff decreased in the lower
elevation region, which can be explained by increased evaporation due to the rising air
temperature.

(3) Model simulated changes in soil moisture and soil temperature indicates that

annual storage of the liquid water increased especially in the most recent three decades,
due to the change in frozen soils. Annual ice water storage in the top 0-3 m of soil
showed a significant decreasing trend due to soil warming. Model simulated annual
groundwater storage had an increasing trend, which is consistent with the changes
observed by the GRACE satellite. This indicated that groundwater recharge in the upper
Heihe basin was enhanced in recent decades.

(4) Model simulation indicated that regions where the permafrost changed into the

seasonally frozen ground had larger changes in runoff and soil moisture than the areas
covered by seasonally frozen ground.

For a better understanding of changes in frozen soils and their impact on

ecohydrology, the interactions among the soil freezing-thawing processes, vegetation
dynamics and hydrological processes need to be investigated in future studies. There
are uncertainties in simulations of the frozen soils and the hydrological processes that
might be related to the soil properties, the high spatial heterogeneity, and the



parameterization of the lower soil boundary conditions, all of which warrant further
investigation in the future.

**Acknowledgements:** This research was supported by the major plan of "Integrated
Research on the Ecohydrological Processes of the Heihe Basin" (Project Nos.
91225302 and 91425303) funded by the National Natural Science Foundation of China
(NSFC). The authors would like to thank the editor for their constructive comments,
which greatly improved the manuscript.

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



**Figure caption:**
Figure 1. The Study area, hydrological stations, borehole observation and flux tower stations
Figure 2. Model structure and vertical discretization of soil column
Figure 3. Comparison of the simulated and the observed soil temperature at borehole observation
sites, and the observed data is provided by Wang et al. (2013)
Figure 4. Daily soil temperature at the Qilian station: (a) observation; (b) simulation; (c) Simulation-
Observation
Figure 5. Comparison of the simulated and observed daily frozen depths during the period of 2002-
2014 at: (a) the Qilian station, (b) the Yeniugou station
Figure 6. Comparison of the simulated and the observed hourly liquid soil moisture at the A'rou
Sunny Slope station
Figure 7. Comparison of the simulated and the observed daily river discharge at: (a) the Yingluoxia
Gauge, (b) the Qilian Gauge, and (c) the Zhamashike Gauge.
Figure 8. Comparison of the simulated and the remote sensing estimated actual evapotranspiration
provided by Wu (2013) in the period of 2002~2012
Figure 9. Changes of the mean soil temperature in different seasons: (a) the freezing season (from
November to March) (b) the thawing season (from April to October)
Figure 10. Change of the frozen soils in the upper Heihe basin: (a) areas of permafrost and basin
averaged annual air temperature; (b) the basin averaged annual maximum frozen depth of the
seasonally frozen ground and the annual maximum thaw depth of the permafrost
Figure 11. Distribution of permafrost and seasonally frozen ground: (a) distribution in the period of
1971-1980; (b) distribution in the period of 2001-2010; (c) Areas where where permafrost changed





into seasonally frozen ground (d) percentage of areas of permafrost and seasonally frozen ground
on sunny slope; (e) percentage of areas of permafrost and seasonally frozen ground on shaded slope
(the same legend as (d))
Figure 12. Spatial averaged monthly soil temperature during the period of 1971-2013 in different
elevation intervals: (a) the seasonally frozen ground with elevation between 3300-3500 m; (b) the
areas where permafrost changed to seasonally frozen ground with elevation between 3500-3700 m
Figure 13. Changes of the runoff and actual evapotranspiration: (a) in the freezing season; (b) in the
thawing season
Figure 14. Changes of the annual water storage (equivalent water depth) during the period of 1971-
2013: (a) the liquid soil water storage of the top 0-3 m layer; (b) the ice water storage of the top 0-
3 m layer; (c) the groundwater storage
Figure 15. Model simulated changes of runoff: (a) in the freezing season, (b) in the thawing season



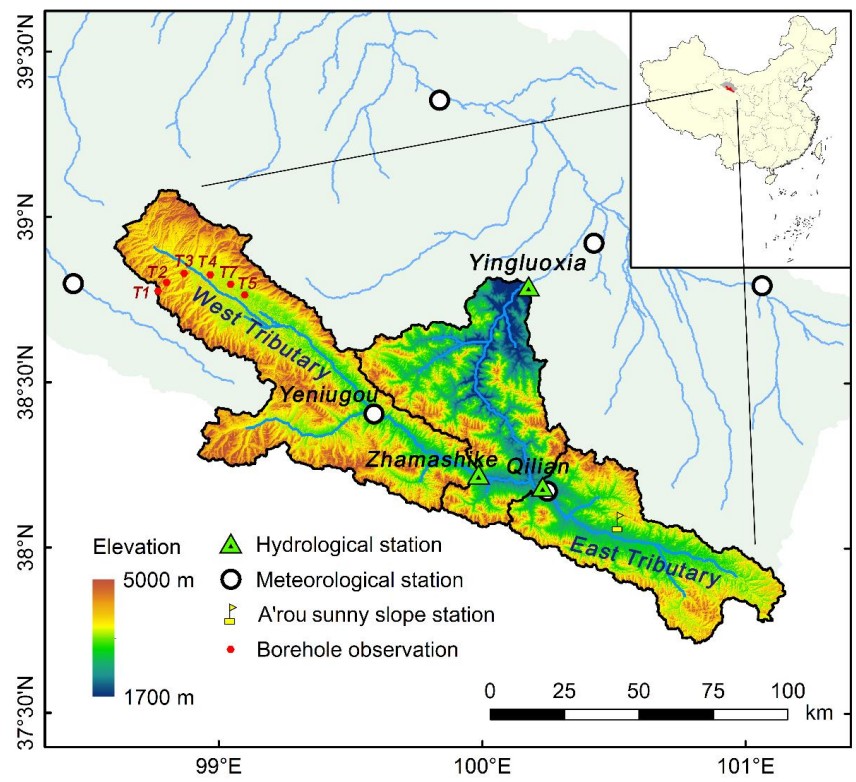


Figure 1. The Study area, hydrological stations, borehole observation and flux tower

stations






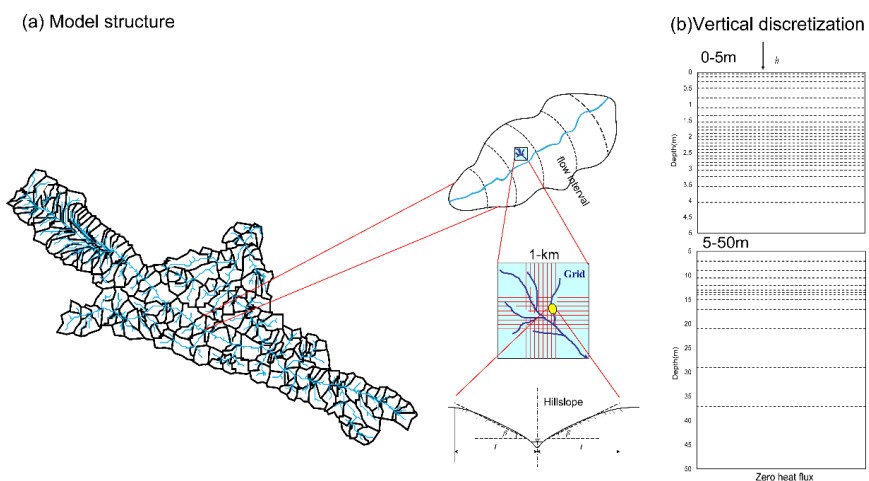


Figure 2. Model structure and vertical discretization of soil column



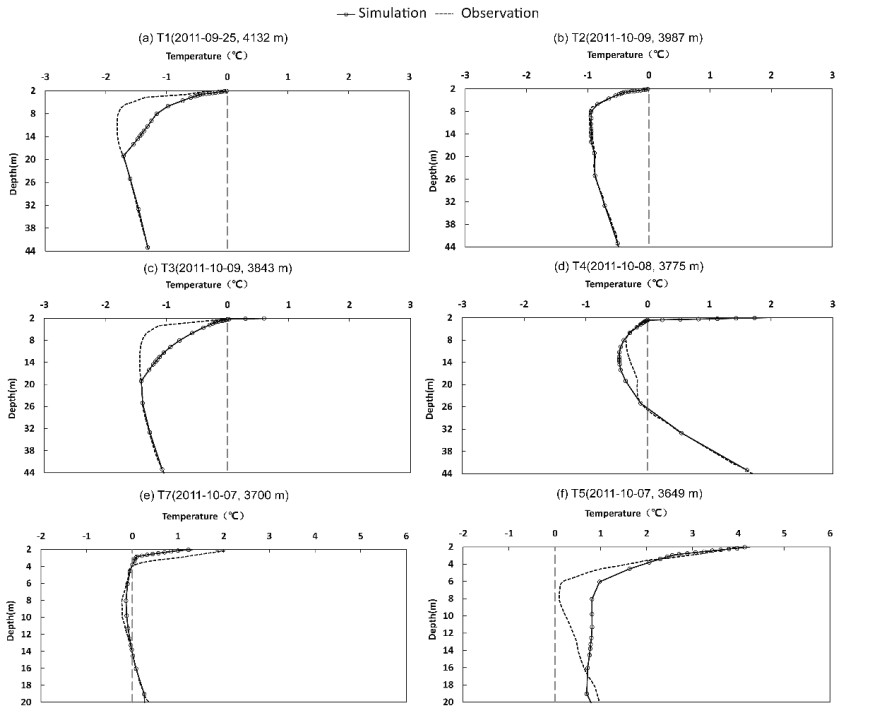


Figure 3. Comparison of the simulated and the observed soil temperature at borehole



observation sites, and the observed data is provided by Wang et al. (2013)

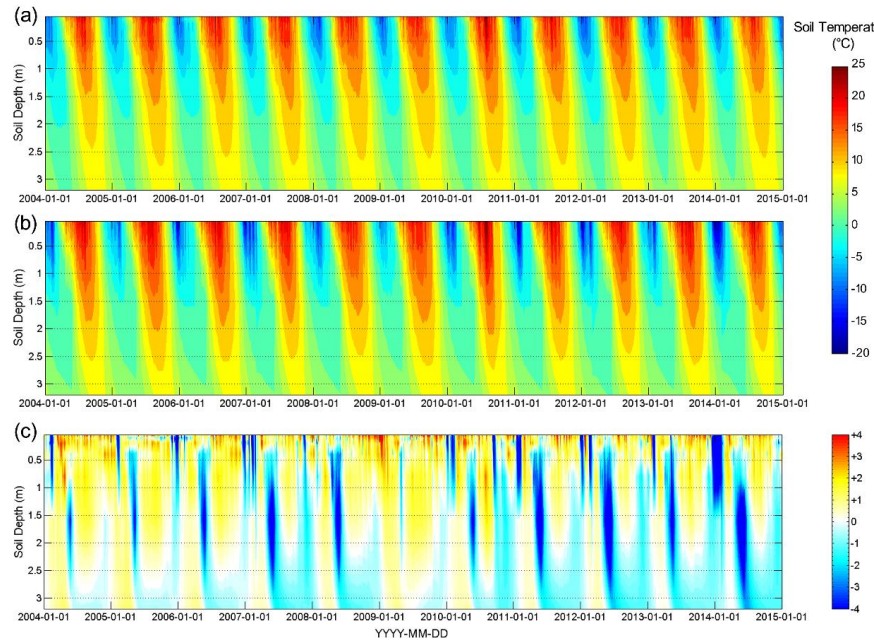


Figure 4 Daily soil temperature at the Qilian station: (a) observation; (b) simulation;

(c) Simulation-Observation






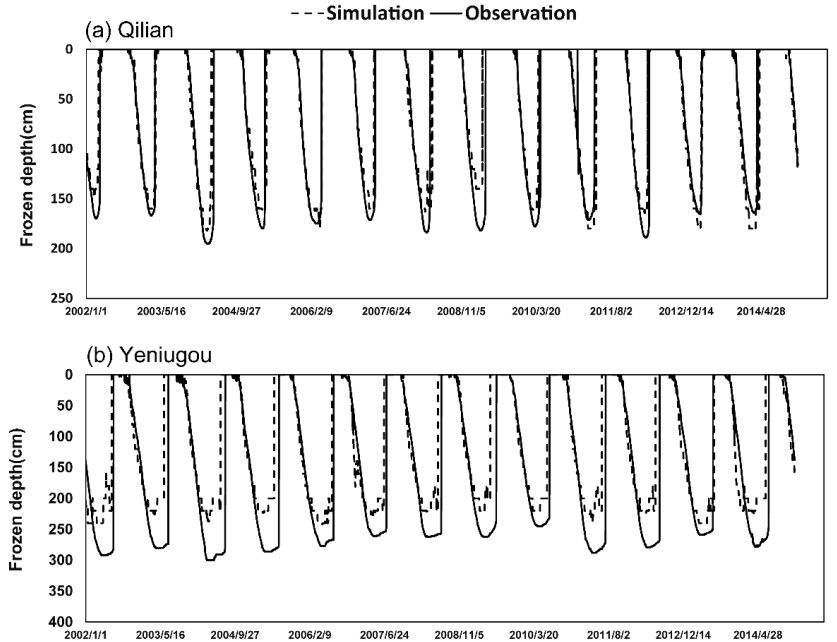

Figure 5. Comparison of the simulated and observed daily frozen depths during the

period of 2002-2014 at: (a) the Qilian station, (b) the Yeniugou station

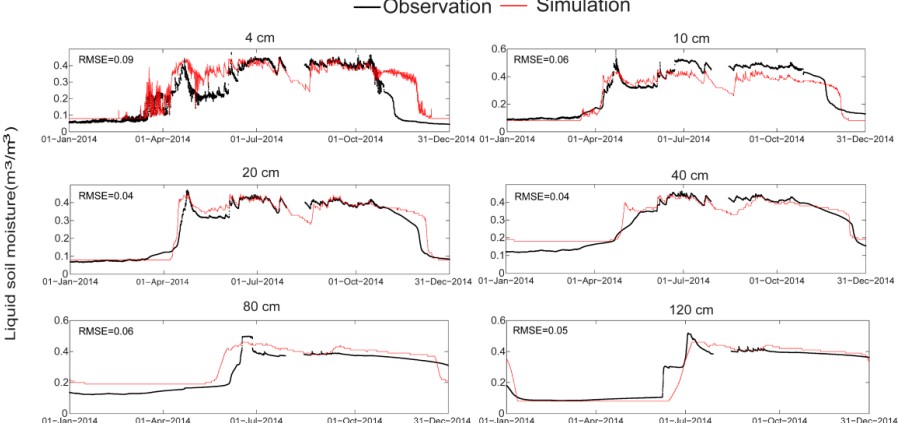

Figure 6. Comparison of the simulated and the observed hourly liquid soil moisture at

the A'rou Sunny Slope station



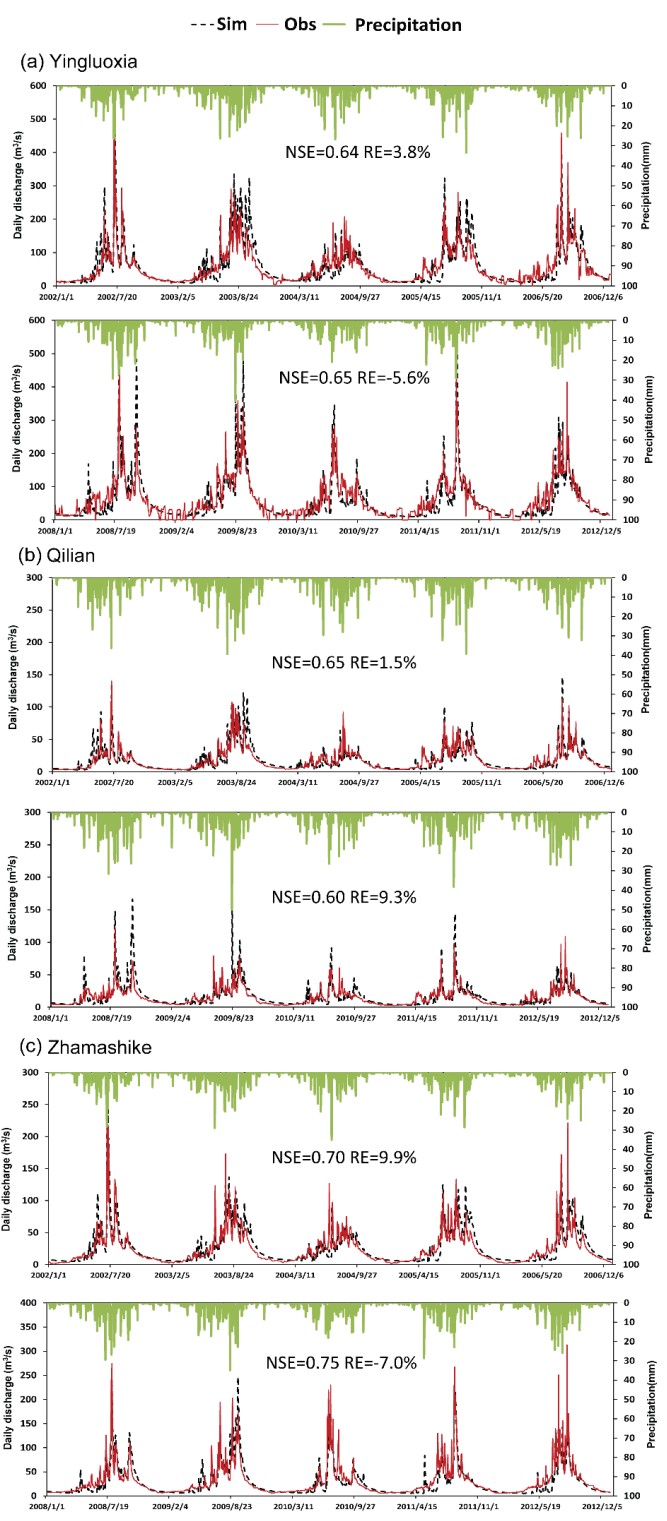





Figure 7. Comparison of the simulated and the observed daily river discharge at: (a)
the Yingluoxia Gauge, (b) the Qilian Gauge, and (c) the Zhamashike Gauge.

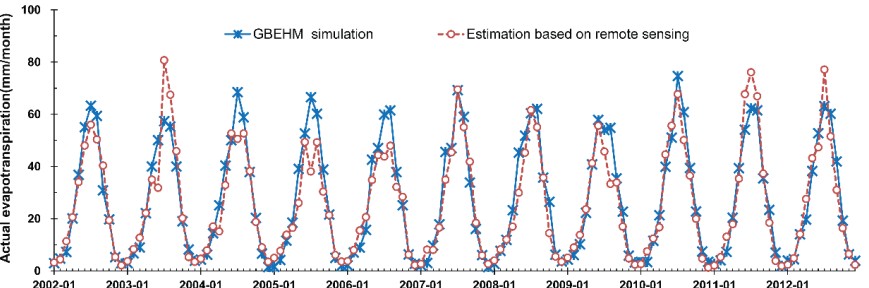


Figure 8. Comparison of the simulated and the remote sensing estimated actual
evapotranspiration provided by Wu (2013) in the period of 2002~2012






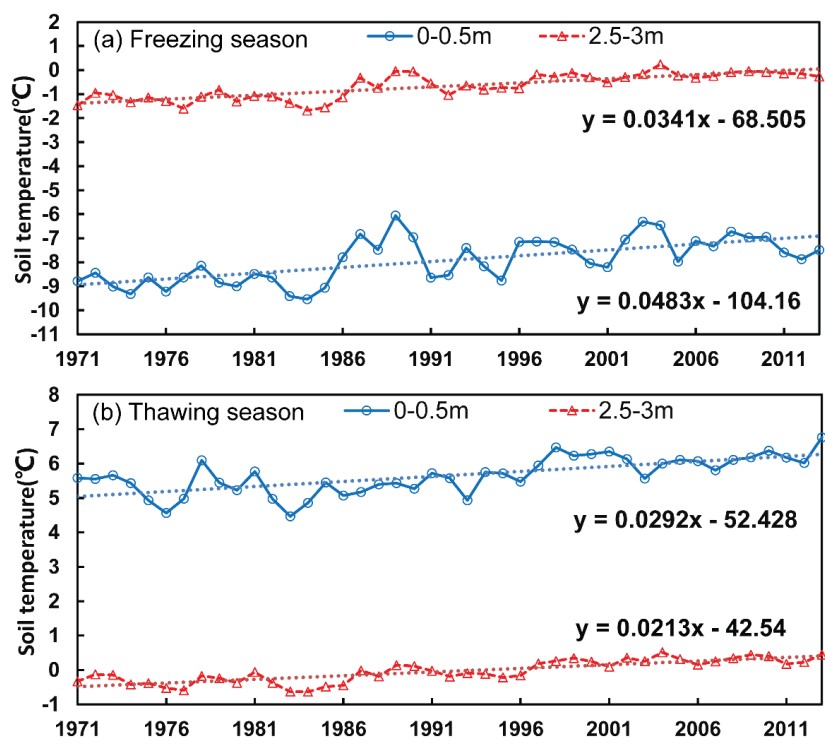

Figure 9. Changes of the mean soil temperature in different seasons: (a) the freezing

season (from November to March) (b) the thawing season (from April to October)




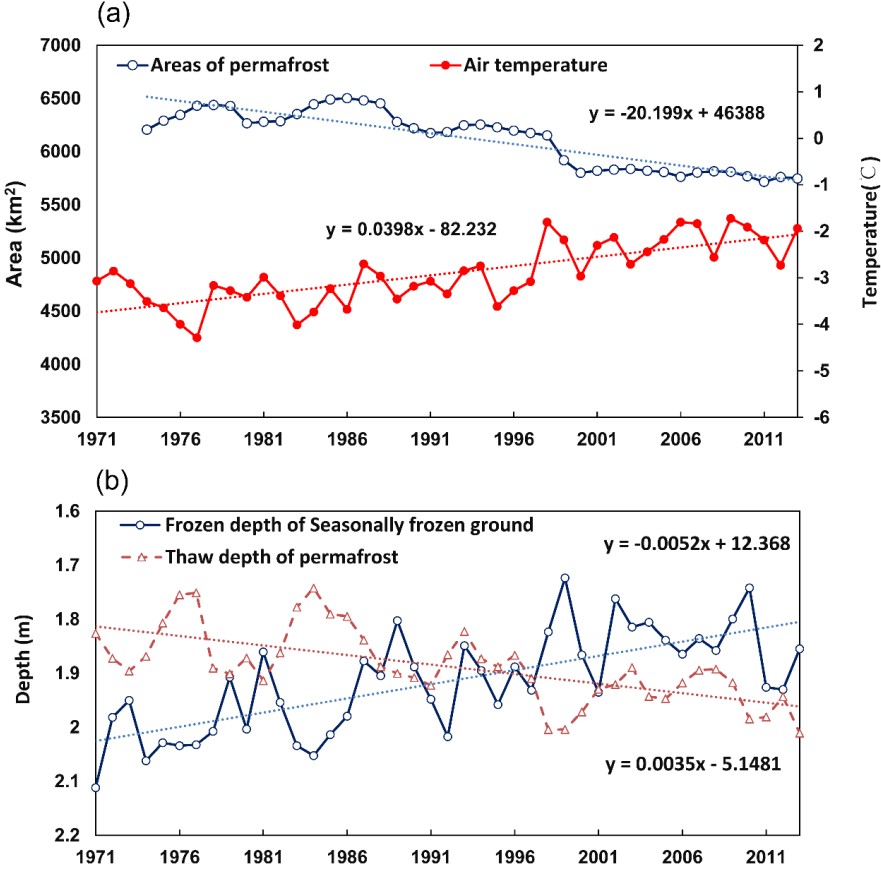

Figure 10. Change of the frozen soils in the upper Heihe basin: (a) areas of permafrost
and basin averaged annual air temperature; (b) the basin averaged annual maximum
frozen depth of the seasonally frozen ground and the annual maximum thaw depth of

the permafrost


886



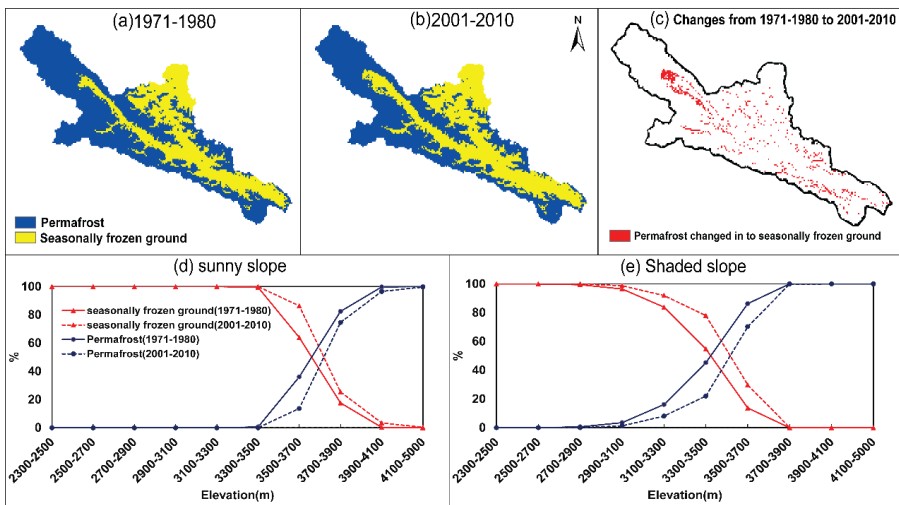

Figure 11. Distribution of permafrost and seasonally frozen ground: (a) distribution in the period of 1971-1980; (b) distribution in the period of 2001-2010; (c) Areas where where permafrost changed into seasonally frozen ground (d) percentage of areas of permafrost and seasonally frozen ground on sunny slope; (e) percentage of areas of permafrost and seasonally frozen ground on shaded slope (the same legend as (d))





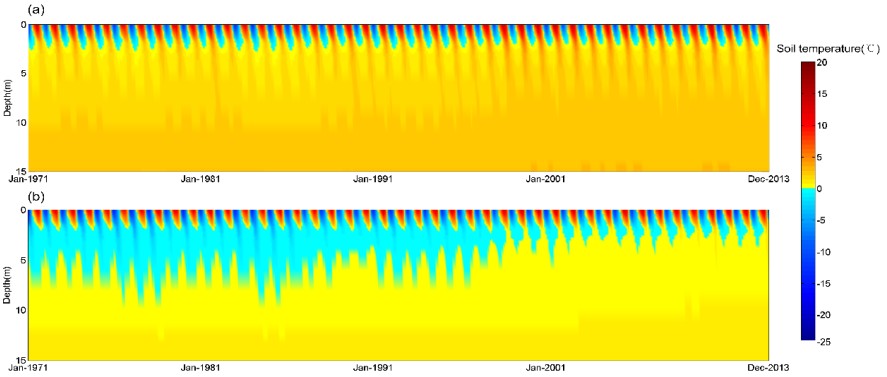


Figure 12. Spatial averaged monthly soil temperature during the period of 1971-2013

in different elevation intervals: (a) the seasonally frozen ground with elevation

between 3300-3500 m; (b) the areas where permafrost changed to seasonally frozen

ground with elevation between 3500-3700 m



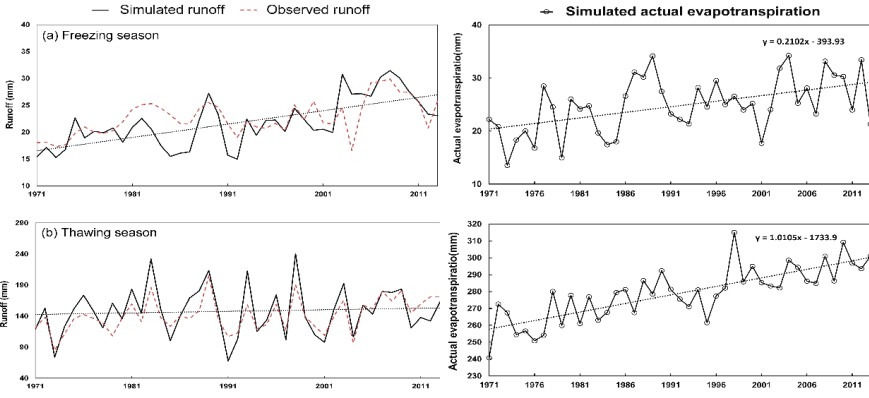


Figure 13. Changes of the runoff and actual evapotranspiration: (a) in the freezing

season; (b) in the thawing season





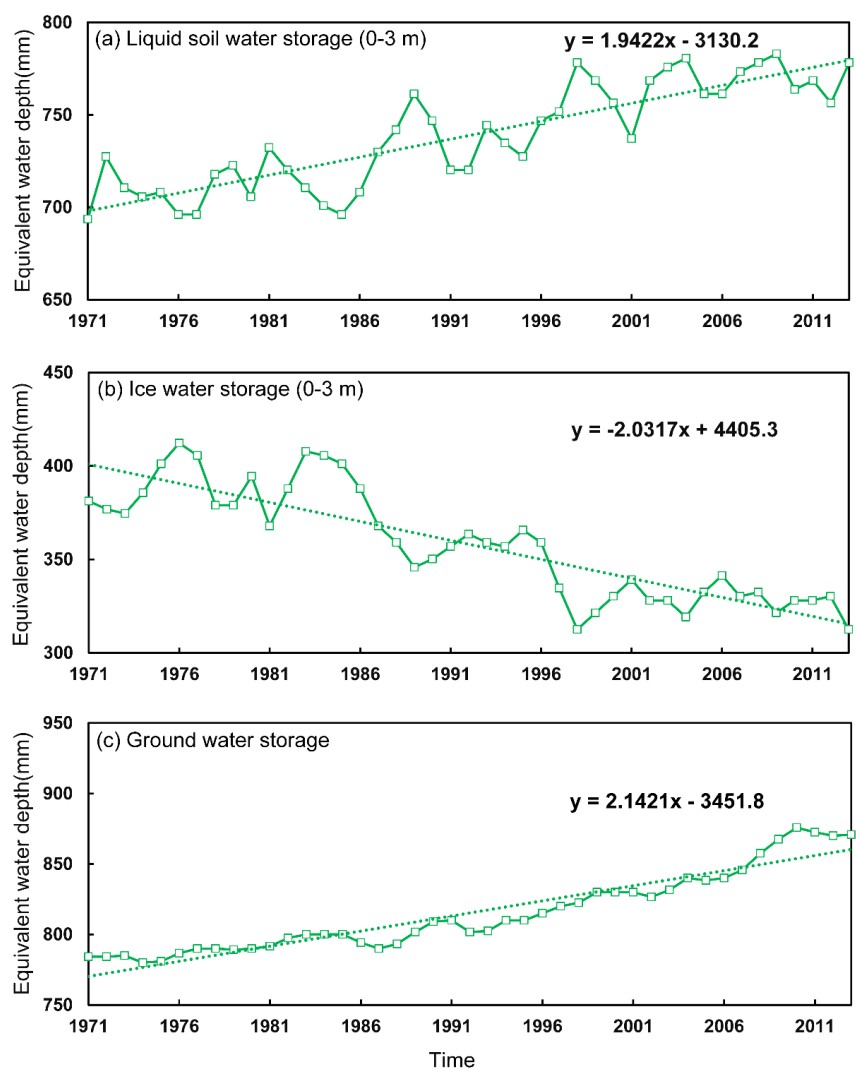


Figure 14. Changes of the annual water storage (equivalent water depth) during the
period of 1971-2013: (a) the liquid soil water storage of the top 0-3 m layer; (b) the ice
water storage of the top 0-3 m layer; (c) the groundwater storage





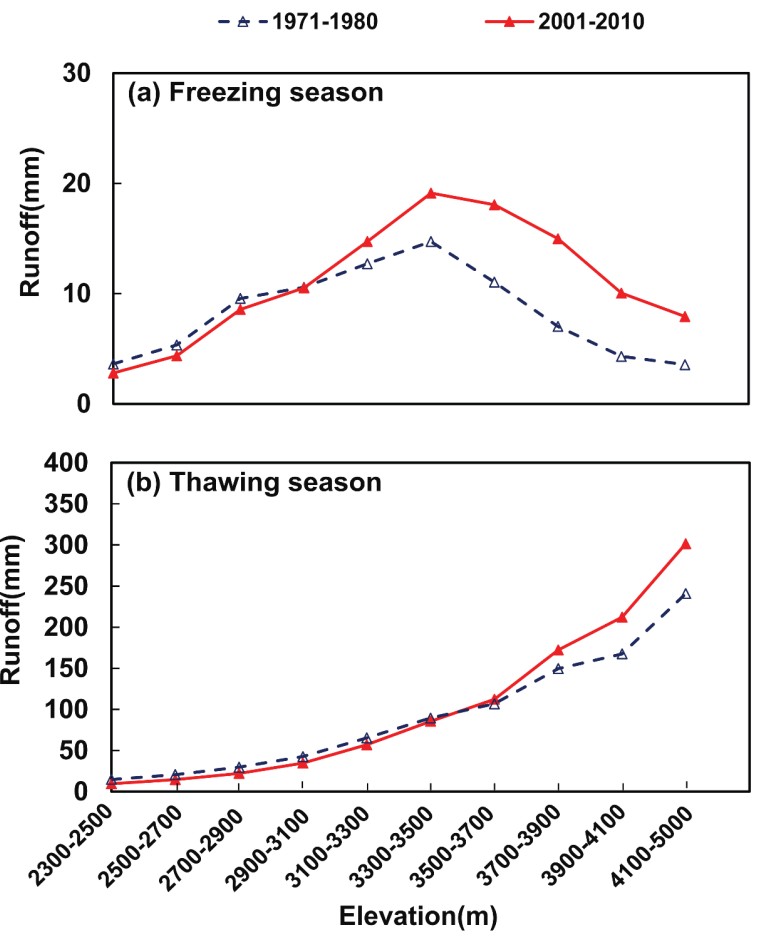


Figure 15. Model simulated changes of runoff: (a) in the freezing season, (b) in the

thawing season












**Table list:**
Table 1 Major parameters of the GBEHM model
Table 2 Changes in annual basin water balance and runoff components in different seasons




Table 1 Major parameters of the GBEHM model

| Parameters | Coniferous Forest | Shrub | Steppe | Alpine Meadow | Alpine Sparse Vegetation | Desert |
|---|---|---|---|---|---|---|
| Surface retention capacity (mm) | 30.0 | 25.0 | 10.0 | 15.0 | 15.0 | 5.0 |
| Surface roughness (Manning coefficient) | 0.5 | 0.3 | 0.1 | 0.1 | 0.1 | 1.0 |
| Soil reflectance to visible light | 0.20 | 0.20 | 0.20 | 0.28 | 0.14 | 0.11 |
| Soil reflectance to near-infrared radiation | 0.225 | 0.225 | 0.225 | 0.28 | 0.225 | 0.225 |
| Leaf reflectance to visible light | 0.105 | 0.105 | 0.105 | 0.105 | 0.105 | — |
| Leaf reflectance to near-infrared radiation | 0.35 | 0.58 | 0.58 | 0.58 | 0.58 | — |
| Leaf transmittance to visible light | 0.05 | 0.07 | 0.07 | 0.07 | 0.07 | — |
| Leaf transmittance to near-infrared radiation | 0.10 | 0.25 | 0.25 | 0.25 | 0.25 | — |
| Maximum Rubsico capacity of top leaf ($10^{-5}$ mol·m$^{-2}$·s$^{-1}$) | 6.0 | 6.0 | 3.3 | 3.3 | 3.0 | — |
| Plant root depth (m) | 2.0 | 1.0 | 0.40 | 0.40 | 0.1 | 0.0 |
| Intrinsic quantum efficiency (mol·mol$^{-1}$) | 0.08 | 0.08 | 0.05 | 0.05 | 0.05 | — |
| Canopy top height (m) | 9.0 | 1.9 | 0.3 | 0.3 | 0.2 | — |
| Leaf length (m) | 0.055 | 0.055 | 0.3 | 0.3 | 0.04 | — |
| Leaf width (m) | 0.001 | 0.001 | 0.005 | 0.005 | 0.001 | — |
| Stem area index | 0.08 | 0.08 | 0.05 | 0.05 | 0.08 | — |


Table 2 Changes in annual basin water balance and runoff components in different seasons

| Decade | Precipitation (mm/yr) | Actual evaporation (mm/yr) | Simulated runoff (mm/yr) | Observed runoff (mm/yr) | Runoff ratio (observed) | Runoff ratio (simulated) | Runoff components (mm/yr) | | | | | |
|---|---|---|---|---|---|---|---|---|---|---|---|---|
| | | | | | | | Freezing season (from November to March) | | | Thawing season (from April to October) | | |
| | | | | | | | T | G | S | T | G | S |
| 1971-1980 | 439.1 | 280.8 | 154.5 | 143.8 | 0.33 | 0.35 | 18.5 | 0.0 | 0.0 | 136.0 | 3.5 | 13.5 |
| 1981-1990 | 492.8 | 300.0 | 186.2 | 174.1 | 0.35 | 0.38 | 20.2 | 0.0 | 0.0 | 166.1 | 3.1 | 28.2 |
| 1991-2000 | 471.0 | 306.1 | 160.1 | 157.4 | 0.33 | 0.34 | 20.4 | 0.0 | 0.0 | 139.7 | 3.8 | 19.2 |
| 2001-2010 | 504.3 | 317.4 | 177.9 | 174.3 | 0.35 | 0.35 | 27.2 | 0.0 | 0.0 | 150.7 | 3.7 | 25.8 |

Note: T means total runoff, G means glacier runoff and S means snowmelt runoff.