# Peer review of "Change in Frozen Soils and Its Effect on Regional Hydrology in the Upper Heihe Basin, on the Northeastern Qinghai-Tibetan Plateau"

_The Cryosphere, 2017_

## Referee Comment (RC1) · Anonymous Referee #1 · 25 Aug 2017

The paper is improved. I have only minor specific comments other than the general comment below.

In general, in this paper I would like the authors to address if there is a horizontal talik above the permafrost or not. Surely there would be somewhere in the catchment especially where the permafrost is discontinuous. Horizontal taliks exist where the maximum frost depth does not extend as deep as the maximum seasonal thaw depth. So there is a perennially unfrozen zone above the permafrost but below the active layer. Could this be significant like the conversion from permafrost to seasonally frozen ground? Would the model account for this?

L51, delete 'the'

L56, delete 'the'

L57,58, the authors should be careful with the phrase 'field experiments' here. If they are talking about looking at field data, some of the studies (e.g., Jacques and Sauchyn 2009) have been done on large spatial scales. If they are talking about intensively monitoring a catchment, that is different and likely only accomplished on fine scales.

L62, delete 'the'

L31 and many other places in the manuscript. If I were the authors I would define runoff up front. They use it in the normal hydrological way, but many hydrogeologists or civil engineers could be interested in this paper, and runoff often means different things to different people. Given that The Cryosphere is a bit of a general cold regions journal, I'd suggest a clear precise definition early on in the paper for runoff.

L70, delete 'the' before 'vertical'

L72, 'overly simple ways' is vague. Explain or remove

L83, delete 'the' before 'downstream'

L88, 'thickness' can be removed (besides it makes no sense to say that the thickness is thin and the thickness is warm)

L91, delete 'the' after 'air'

L103-107 and 172-175. This seems to be almost branding for the research grant and, in my mind, is only suitable for the acknowledgements not the main body of a research paper.

L122, 'mean' should be before 'annual'

I'm surprised by the high RMSE in the soil temperature results, especially after calibration'

Section 5.3, the uncertainty 'analysis' is highly qualitative

Figure 1, The authors could show the plateau on the map of China. OR be clearer what the grey zone refers to in the inset.

Figure 7, the caption should be clearer why there are groups of 2 figure panels for each location.

[Figure]

---

## Referee Comment (RC2) · Anonymous Referee #2 · 14 Sep 2017

Comments on **"Change in Frozen Soils and Its Effect on Regional Hydrology in the Upper Heihe Basin, on the Northeastern Qinghai-Tibetan Plateau"** by Gao et al. submitted to *The Cryospshere*

**General comments**

This paper reports a modelling study about the impacts of climate warming on frozen ground and hydrological processes for a large mountainous area containing permafrost and non-permafrost areas. The model reasonably captured the thermal and hydrological processes, especially the seasonal and long-term variations of river discharge and runoff. The results show changes in permafrost extent and thawing/freezing deaths, and associated changes in hydrological processes in this large area. The spatially distributed modelling approach is novel and efficient for such a large and cold region as well. This work is valuable to demonstrate the progress in high-resolution thermal-hydrological integrated spatial modelling for large cold regions and to understand the impacts of climate change on frozen ground and associated hydrological processes. Although I agree with some of the concerns indicated by the previous reviewer (see details below), I feel it is worthy to be published after a revision.

**Specific comments**

I agree with the concern of the previous reviewer that the almost exact simulation of ground temperatures at deep layers at the test sites (Figure 3, and Figure S1) probably is the results of calibration, i.e., setting the initial values. The paper should indicate that and probably needs to revise the phase "generally accurate" to a looser term. In addition, if there is no geothermal heat flux at the bottom, ground temperature profile in lower ground should not vary much with depth under equilibrium conditions (unless the 10-year climate force used to spin-up varied significantly from year to year). Most simulated and observed soil temperature increased with depth, indicating the existence of a geothermal gradient. The assumption of zero geothermal heat flux at the lower boundary (Line 279) seems not right.

The causal relationship between changes in frozen ground and runoff is an important issue and the paper tried to answer it. The high correlation between liquid soil moisture and runoff in freezing season is not enough to establish that causal relationship (Line 471-474). The modelling exercise of without frozen soil and Figure S2 are a direct way to show the effects of frozen ground and its thaw. More detailed explanation of the modelling exercise needs to be provided (e.g., how the model was modified to do that? is this a run for a grid or for the entire basin?). Figure 15 is interesting but not so clear for

me. An analysis from typical grids (permafrost and non-permafrost grids) and seasonal patters (e.g., Figure S2) might be helpful to understand it.

**Minor points**

Line 30: "active layer depth". Using "active layer thickness" for consistency in the paper

Line 34: "large changes in runoff", can you specify "change" as "increase"?

Line 38-39: "due to the degradation of permafrost in the study area". Increase in precipitation probably also contributed to that change".

Line 47: revise "regime" to "regimes"

Line 51: delete "the".

Line 59: "…the frozen soil, and the long-term…", separate it into two sentences.

Line 71: "…processes especially …", add a comma after "processes".

Lines 85-86. Unpack it into two sentences. "The Qinhai-Tibetan Plateau is characterized by… . Cryospheric processes have great impacts on its hydrological processes".

Line 88: "permafrost thickness", delete "thickness"

Line 91: delete "the"

Line 103-110: Agree with the previous reviewer, delete it.

Section 2: probably no needs for the sub-titles. Just describe the study area, field observations and spatial data. Try to focus on data description rather than how they are used (leave that in the next section).

Line 135-136: moving "provided by ..(CMA)" to the end of the sentence.

Line 141: revise "build" to "run"

Line 173-175: "in an integrated … 2016)" delete it.

Line 176: The approach of the model is very interesting. It is not a fully distributed model with lateral interactions but efficient in computation and handling water flows. I feel that is a important progress of spatial modelling. You may add some sentences about the feature of the model (not branding the project or funding) before "As…".

Line 188 revise "used in" to "of"

Line281-285: You used thinner layers around 0.8m and from 1.7-3m, probably to capture the details of maximum thawing/freezing depths. You mad add some explanations about that.

Section 3.3: you may begin the section by "to initialize the model, we first estimated the soil temperature profiles based on the assumption that ….". You may delete the sentence "For spin up run, the initial … this relationship".

Line 318-319: the good agreement probably is due to calibration of initial values. A 500 year spin up run should change to a near constant ground temperature with depth. You need to check the model or whether the climate data from 1961-1970 vary significantly from year to year that prevent the establishment of equilibrium conditions.

Line 338-339. The value of RMSE and variation with depth is comparable with the study of Ou et al. (2016).

Line 353: revise "station" to "stations"

Line 366: "without the frozen soil scheme". How the model was modified? and is the Figure S2 for the entire basin or just a grid? This is an important part of directly show the effects frozen ground on runoff. More description is needed, probably in the method section.

Section 4.2: It would be useful include the trends of air temperature (annual, thawing and freezing seasons) in the analysis.

Line 407: "In contrast". Not a real contrast. It is expected. Delete it.

Line 434 "accurately reproduced" may be replace by "well reproduced"

Line 453: is increasing in precipitation a factor?

Line 455-469. It is easy to understand that the volume of soil liquid water increases with the increase in the volume of unfrozen soils. The sentences in these lines are long and complicated. You may simplify them.

 Lines 471-474: You need more evidence to support the causal relations. The correlation is only one evidence. See the specific comments.

Line 482: From Table 2, the increase probably is mainly due to increase in snow run off in thaw season.

Line 484-485: revise "during the different seasons" to "between the two seasons".

Line 499, 506, 512: "change in frozen soil". You may specify it as "thaw of frozen soil"

Line 508: "was controlled by" the word probably is too strong. You may use "strongly affected by"

Line 512 revise "soil moisture" to "soil liquid moisture"

Line 540-522: "Different methods produce large differences in their estimates". The following citations do not support such a statement since they are mainly for different areas. Actually, some of the estimates in Qinghai-Tibetan Plateau is comparable with your estimate, which is a support of your estimate.

Line 568: "especially in spring", not clear for me.

The sediment thickness (depth from surface to bedrock), top organic layer thickness, and fraction of rock in soil are important to ground temperature and permafrost. You may add some sentences about them in sections of data, discussion or uncertainty. Active layer is thinner in valleys than in high slopes and on top of mountains due to differences in organic layer and vegetation conditions (Zhang et al., 2013). Temperature inversion and shading by surroundings may also keep the valley cooler than top of the mountains (O'Neil et al., 2015). You may comments on this in the discussions.

Figures: The font of words or numbers are too small in most figures.

Figure 3, S1: It is better to use a line with dots to represent the observations (so readers know the depths of observations). If you have annual averages, it is better to use annual averages rather than a specific date or month.

Figure 10b: revise "thaw depth" to "active layer thickness"

Figure 11d,e, Red curves are not necessary. For easy understanding, you may put elevation as Y axis, and percentage of permafrost to x axis.

References mentioned

Ou, C., B. Leblon, Y. Zhang, A. LaRocque, K. Webster, and J. McLaughlin (2016), Modelling and mapping permafrost at high spatial resolution using Landsat and RADARSAT images in northern Ontario, Canada: Part 1 - Model calibration, International Journal of Remote Sensing, doi: 10.1080/01431161.2016.1157642.

Zhang, Y., X. Wang, R. Fraser, I. Olthof, W. Chen, D. Mclennan, S. Ponomarenko, and W. Wu (2013), Modelling and mapping climate change impacts on permafrost at high spatial resolution for an Arctic region with complex terrain, The Cryosphere, 7, 1121–1137, doi:10.5194/tc-7-1121-2013. www.the-cryosphere.net/7/1121/2013/

O'Neill, H. B., Burn, C. R., Kokelj, S. V. & Lantz, T. C. 'Warm' tundra: atmospheric and near-surface ground temperature inversions across an alpine treeline in continuous permafrost, western arctic, Canada. Permafrost and Periglac. Process. 26, 103–118 (2015). doi: 10.1002/ppp.1838.

---

## Author Comment (AC1) · 1 Oct 2017

First we want to thank Anonymous Referee #1 for his constructive comment and his good suggestions. We are answering his comments in the following, for clarity we repeat the original comment (C) and answer (A) afterwards:

Specific comments

C:The paper is improved. I have only minor specific comments other than the general comment below. In general, in this paper I would like the authors to address if there is a horizontal talik above the permafrost or not. Surely there would be somewhere in

the catchment especially where the permafrost is discontinuous. Horizontal taliks exist where the maximum frost depth does not extend as deep as the maximum seasonal thaw depth. So there is a perennially unfrozen zone above the permafrost but below the active layer. Could this be significant like the conversion from permafrost to seasonally frozen ground? Would the model account for this?

A: Thanks for this comment. We have checked the results of soil temperature of each layer for all grids. We have found some taliks in permafrost region. But the horizontal taliks are not significant as shown in fig. 1.

Minor comments

C: L51, delete 'the'.

A: We have revised as suggested.

C: L56, delete 'the'

A: We have revised as suggested.

C: L57,58, the authors should be careful with the phrase 'field experiments' here. If they are talking about looking at field data, some of the studies (e.g., Jacques and Sauchyn 2009) have been done on large spatial scales. If they are talking about intensively monitoring a catchment, that is different and likely only accomplished on fine scales.

A: We have modified "field experiments" as "intensive field observations"

C: L62, delete 'the'.

A: We have revised as suggested.

C: L31 and many other places in the manuscript. If I were the authors I would define runoff up front. They use it in the normal hydrological way, but many hydrogeologists or civil engineers could be interested in this paper, and runoff often means different things to different people. Given that The Cryosphere is a bit of a general cold regions

journal, I'd suggest a clear precise definition early on in the paper for runoff.

A: Thank you for this suggestion, we added the runoff definition when it first appeared in this paper. The related sentence has been revised as "A few studies reported that permafrost thawing might reduce river runoff (This paper defines the runoff as all liquid water flowing out of the study area.)".

C: L70, delete 'the' before 'vertical'.

A: We have revised as suggested.

C: L72, 'overly simple ways' is vague. Explain or remove?

A: We modified "in overly simple ways" as "by simplified ways".

C: L83, delete 'the' before 'downstream'

A: We have revised as suggested.

C: L88, 'thickness' can be removed (besides it makes no sense to say that the thickness is thin and the thickness is warm).

A: We have revised as suggested.

C: L91, delete 'the' after 'air'.

A: We have revised as suggested.

C: L103-107 and 172-175. This seems to be almost branding for the research grant and, in my mind, is only suitable for the acknowledgements not the main body of a research paper.

A: we have delete the sentences about the Heihe Research plan in the main body.

C: L122, 'mean' should be before 'annual'.

A: We have revised.

C: I'm surprised by the high RMSE in the soil temperature results, especially after calibration'

A: We calibrated the soil reflectance according to vegetation type, and we do not calibrate the soil heat capacity and soil thermal conductivity. The soil heat capacity and soil thermal conductivity are estimated using the method developed by Farouki (1981). This may lead uncertainties in simulation of the soil temperature.

C: Section 5.3, the uncertainty 'analysis' is highly qualitative.

A: Yes, we discussed the uncertainty qualitatively in this paper. Due to the complexity of the distributed model and large number of model parameters, it is challenge to quantify simulation uncertainty. This work will be done in the future study.

C: Figure 1, The authors could show the plateau on the map of China. OR be clearer what the grey zone refers to in the inset.?

A: We have modified Figure 1 to show the plateau on the map of China.

C: Figure 7, the caption should be clearer why there are groups of 2 figure panels for each location.

A: We have changed the caption of Figure 7 as "Comparison of the simulated and the observed daily river discharge at: (a) the Yingluoxia Gauge, (b) the Qilian Gauge, and (c) the Zhamashike Gauge (The upper panel is the calibration period, and the bottom panel is the validation period for each gauge)"

─────────────────────────────

**Legend**

- ▢ Seasonally frozen ground
- ▢ Permafrost
- ▢ Talik

**Fig. 1.**

---

## Author Comment (AC2) · 1 Oct 2017

First we want to thank Anonymous Referee #2 for his constructive comment and his good suggestions. We are answering his comments in the following, for clarity we repeat the original comment (C) and answer (A) afterwards.

General comments:

C: This paper reports a modelling study about the impacts of climate warming on frozen ground and hydrological processes for a large mountainous area containing permafrost and non-permafrost areas. The model reasonably captured the thermal and hydrologi-

cal processes, especially the seasonal and long-term variations of river discharge and runoff. The results show changes in permafrost extent and thawing/freezing deaths, and associated changes in hydrological processes in this large area. The spatially distributed modelling approach is novel and efficient for such a large and cold region as well. This work is valuable to demonstrate the progress in high-resolution thermal-hydrological integrated spatial modelling for large cold regions and to understand the impacts of climate change on frozen ground and associated hydrological processes. Although I agree with some of the concerns indicated by the previous reviewer (see details below), I feel it is worthy to be published after a revision.

A: Thanks for this positive comment.

Specific comments:

C:I agree with the concern of the previous reviewer that the almost exact simulation of ground temperatures at deep layers at the test sites (Figure 3, and Figure S1) probably is the results of calibration, i.e., setting the initial values. The paper should indicate that and probably needs to revise the phase "generally accurate" to a looser term. In addition, if there is no geothermal heat flux at the bottom, ground temperature profile in lower ground should not vary much with depth under equilibrium conditions (unless the 10-year climate force used to spin-up varied significantly from year to year). Most simulated and observed soil temperature increased with depth, indicating the existence of a geothermal gradient. The assumption of zero geothermal heat flux at the lower boundary (Line 279) seems not right.

A: We have deleted "accurate", and changed the expression as "The model generally captured the vertical distribution of the soil temperature at T1, T2, T3 and T4 in the permafrost area. Good agreement between the simulated and observed soil temperature profiles below the depth of 20 m is probably due to fitting of initial values." We also recognized the fact of soil temperature increased with depth, indicating the existence of a geothermal gradient, which may cause uncertainty in our simulation. We will estimate

the geothermal heat flux at the lower boundary and run the model by considering the heat flux at the lower boundary and compare the results.

C: The causal relationship between changes in frozen ground and runoff is an important issue and the paper tried to answer it. The high correlation between liquid soil moisture and runoff in freezing season is not enough to establish that causal relationship (Line 471-474). The modelling exercise of without frozen soil and Figure S2 are a direct way to show the effects of frozen ground and its thaw. More detailed explanation of the modelling exercise needs to be provided (e.g., how the model was modified to do that? is this a run for a grid or for the entire basin?). Figure 15 is interesting but not so clear for me. An analysis from typical grids (permafrost and non-permafrost grids) and seasonal patters (e.g., Figure S2) might be helpful to understand it.

A: For Figure S2, the result is obtained by a run for entire basin. We have modified the figure caption to indicate this. We have added explanations about how the model is modified to run without frozen soil in section 3.4. Figure S4 has been added in supplement files to show runoff changes in typical regions where the permafrost changed into seasonally frozen ground. We have added discussions on the runoff changes in section 5.1 as "Runoff changes of typical grids in areas (with elevation between 3500-3700 m) covered by permafrost in the period of 1971 to 1980 and by seasonally frozen ground in the period of 2001 to 2010 are shown in Figure S4. It illustrates that thaw of permafrost increased the runoff in the freezing season and slowed recession processes in autumn. The increase in freezing season runoff and shift in the seasonal flow pattern are also illustrated by the model simulation without frozen soil scheme as shown in Figure S2." We will also show seasonal pattern of the runoff in the permafrost areas and the seasonally frozen ground areas in Figure 15.

Minor comments:

C: Line 30: "active layer depth". Using "active layer thickness" for consistency in the paper

[Figure]

A: We have revised as suggested.

C: Line 34: "large changes in runoff", can you specify "change" as "increase"?

A: Yes, we have use "increase" instead of "change".

C: Line 38-39: "due to the degradation of permafrost in the study area". Increase in precipitation probably also contributed to that change".

A: Increase in precipitation may also contribute to this change, but degradation of permafrost is the major reason. We modified this sentence as "mainly due to the degradation of permafrost in the study area"

C: Line 47: revise "regime" to "regimes"

A: We have revised as suggested.

C: Line 51: delete "the".

A: We have revised as suggested.

C: Line 59: "...the frozen soil, and the long-term...", separate it into two sentences.

A: We have revised as suggested.

C: Line 71: "...processes especially ...", add a comma after "processes".

A: We have modified this sentence as "but they represent the flow routing at the catchment scale by simplified ways" according to the comment of reviewer 1.

C: Lines 85-86. Unpack it into two sentences. "The Qinhai-Tibetan Plateau is characterized by.... Cryospheric processes have great impacts on its hydrological processes".

A: We have revised as suggested.

C: Line 88: "permafrost thickness", delete "thickness"

A: We have revised as suggested.

C: Line 91: delete "the"

A: We have revised as suggested.

C: Line 103-110: Agree with the previous reviewer, delete it.

A: We have delete the sentences about the Heihe Research plan as suggested.

C: Section 2: probably no needs for the sub-titles. Just describe the study area, field observations and spatial data. Try to focus on data description rather than how they are used (leave that in the next section).

A: We have revised as suggested.

C: Line 135-136: moving "provided by ..(CMA)" to the end of the sentence.

A: We have revised as suggested.

C: Line 141: revise "build" to "run"

A: We have revised as suggested.

C: Line 173-175: "in an integrated . . . 2016)" delete it.

A: We have revised as suggested.

C: Line 176: The approach of the model is very interesting. It is not a fully distributed model with lateral interactions but efficient in computation and handling water flows. I feel that is a important progress of spatial modelling. You may add some sentences about the feature of the model (not branding the project or funding) before "As. . .".

A: We have added some sentences as "GBEHM is a spatial distributed model for large-scale river basin. It employs the geomorphologic properties to reduce the lateral two-dimensions into one-dimension for flow routing calculation within a sub-catchment, which greatly improves the computation efficiency while retaining the spatial heterogeneity in water flow paths at basin scale."

C: Line 188 revise "used in" to "of"

A: We have revised as suggested.

C: Line281-285: You used thinner layers around 0.8m and from 1.7-3m, probably to capture the details of maximum thawing/freezing depths. You mad add some explanations about that.

A: We have added a sentence as "As shown in Figure 2, thinner layers are used at the depth from 1.7 to 3 m for better capturing the maximum frozen depth according to the field observations."

C: Section 3.3: you may begin the section by "to initialize the model, we first estimated the soil temperature profiles based on the assumption that . . ..". You may delete the sentence "For spin up run, the initial . . . this relationship".

A: We have revised as suggested.

C: Line 318-319: the good agreement probably is due to calibration of initial values. A 500 year spin up run should change to a near constant ground temperature with depth. You need to check the model or whether the climate data from 1961-1970 vary significantly from year to year that prevent the establishment of equilibrium conditions.

A: We have checked the climate data, air temperature rising started from 1980s. We have changed this sentence as "Good agreement between the simulated and observed soil temperature profiles below the depth of 20 m is probably due to fitting of initial values".

C: Line 338-339. The value of RMSE and variation with depth is comparable with the study of Ou et al. (2016).

A: We have added a sentence as "This result is similar with the findings by Ou et al. (2016) using the Northern Ecosystem Soil Temperature (NEST) model."
C: Line 353: revise "station" to "stations"

A: We have revised as suggested.

C: Line 366: "without the frozen soil scheme". How the model was modified? and is the Figure S2 for the entire basin or just a grid? This is an important part of directly show the effects frozen ground on runoff. More description is needed, probably in the method section.

A: We added some explanations about the model modification in section 3.4. Figure S2 is for entire basin.

C: Section 4.2: It would be useful include the trends of air temperature (annual, thawing and freezing seasons) in the analysis.

A: We have added a table in supplement file (Table S2) and discussions in the manuscript as "Table S2 shows that annual mean air temperature increased by approximately 1.2°C in the period of 2001 to 2010 comparing with the period of 1971 to 1980. And air temperature in the freezing season shows larger increase (approximately 1.4°C) than in the thawing season (1.1°C) between the two periods."

C: Line 407: "In contrast". Not a real contrast. It is expected. Delete it.

A: We have revised as suggested.

C: Line 434 "accurately reproduced" may be replace by "well reproduced"

A: We have revised as suggested.

C: Line 453: is increasing in precipitation a factor?

A: Increasing precipitation may be a factor, but it is not the major factor.

C: Line 455-469. It is easy to understand that the volume of soil liquid water increases with the increase in the volume of unfrozen soils. The sentences in these lines are long and complicated. You may simplify them.

A: We have simplified this part as suggested.

C: Lines 471-474: You need more evidence to support the causal relations. The correlation is only one evidence. See the specific comments.

A: We have added new figures and analysis to support the causal relations as mentioned above.

C: Line 482: From Table 2, the increase probably is mainly due to increase in snow run off in thaw season.

A: We have revised this sentence as "The increased runoff mainly came from increased precipitation and snowmelt in the thawing season."

C: Line 484-485: revise "during the different seasons" to "between the two seasons".

A: We have revised as suggested.

C: Line 499, 506, 512: "change in frozen soil". You may specify it as "thaw of frozen soil"

A: We have revised as suggested.

C: Line 508: "was controlled by" the word probably is too strong. You may use "strongly affected by"

A: We have revised as suggested.

C: Line 512 revise "soil moisture" to "soil liquid moisture"

A: We have revised as suggested.

C: Line 540-522: "Different methods produce large differences in their estimates". The following citations do not support such a statement since they are mainly for different areas. Actually, some of the estimates in Qinghai-Tibetan Plateau is comparable with your estimate, which is a support of your estimate.

A: We have deleted the citations (Jorgenson et al., 2006 and Chasmer et al., 2010) for other areas and only compared our results with estimates in the Qinghai-Tibetan Plateau. A new citation of Guo et al. (2013) for the change of permafrost area in the Qinghai-Tibetan Plateau is added. We have deleted the sentence "Different methods produce large differences in their estimates"

C: Line 568: "especially in spring", not clear for me.

A: Here this means high groundwater flow rate events such as spring freshet. To make it more clear. We modified this as "especially when high groundwater flow rate events occur"

C: The sediment thickness (depth from surface to bedrock), top organic layer thickness, and fraction of rock in soil are important to ground temperature and permafrost. You may add some sentences about them in sections of data, discussion or uncertainty. Active layer is thinner in valleys than in high slopes and on top of mountains due to differences in organic layer and vegetation conditions (Zhang et al., 2013). Temperature inversion and shading by surroundings may also keep the valley cooler than top of the mountains (O'Neil et al., 2015). You may comments on this in the discussions.

A: We have added some discussions about this in section 5.3 as "Sub-grid topography may also affect the frozen soil simulation. For example, active layer thickness is different in the low valleys and high slopes due to different vegetation conditions, soil organic layers and shading by surroundings. These factors are not well considered in this study."

C: Figures: The font of words or numbers are too small in most figures.

A: We will change the font of words or numbers in the figures to make it clear.

C: Figure 3, S1: It is better to use a line with dots to represent the observations (so readers know the depths of observations). If you have annual averages, it is better to use annual averages rather than a specific date or month.

A: We have modified the figure as suggested. We do not have annual averages.

C: Figure 10b: revise "thaw depth" to "active layer thickness"

A: We have modified the figure as suggested.

C: Figure 11d,e, Red curves are not necessary. For easy understanding, you may put elevation as Y axis, and percentage of permafrost to x axis.

A: We have modified the figure as suggested.

---

## Author Response (AR1)

**1. Comments from Referees**

**Comment by Anonymous Referee #1**

**General comments**

The paper is improved. I have only minor specific comments other than the general comment below.

In general, in this paper I would like the authors to address if there is a horizontal talik above the permafrost or not. Surely there would be somewhere in the catchment especially where the permafrost is discontinuous. Horizontal taliks exist where the maximum frost depth does not extend as deep as the maximum seasonal thaw depth. So there is a perennially unfrozen zone above the permafrost but below the active layer. Could this be significant like the conversion from permafrost to seasonally frozen ground? Would the model account for this?

**Minor comments**

**1.** L51, delete 'the'.
**2.** L56, delete 'the'
**3.** L57,58, the authors should be careful with the phrase 'field experiments' here. If they are talking about looking at field data, some of the studies (e.g., Jacques and Sauchyn 2009) have been done on large spatial scales. If they are talking about intensively monitoring a catchment, that is different and likely only accomplished on fine scales.
**4.** L62, delete 'the'.
**5.** L31 and many other places in the manuscript. If I were the authors I would define runoff up front. They use it in the normal hydrological way, but many hydrogeologists or civil engineers could be interested in this paper, and runoff often means different things to different people. Given that The Cryosphere is a bit of a general cold regions journal, I'd suggest a clear precise definition early on in the paper for runoff.
**6.** L70, delete 'the' before 'vertical'.
**7.** L72, 'overly simple ways' is vague. Explain or remove?
**[8.** L83, delete 'the' before 'downstream'
**9.** L88, 'thickness' can be removed (besides it makes no sense to say that the thickness is thin and the thickness is warm).
**10.** L91, delete 'the' after 'air'.
**11.** L103-107 and 172-175. This seems to be almost branding for the research grant and, in my mind, is only suitable for the acknowledgements not the main body of a research paper.
**12.** L122, 'mean' should be before 'annual'.
**13.** I'm surprised by the high RMSE in the soil temperature results, especially after calibration'
**14.** Section 5.3, the uncertainty 'analysis' is highly qualitative.

**15.** Figure 1, The authors could show the plateau on the map of China. OR be clearer what the grey zone refers to in the inset.?

**16.** Figure 7, the caption should be clearer why there are groups of 2 figure panels for each location.

**Comment by Anonymous Referee #2**

**General comments**

This paper reports a modelling study about the impacts of climate warming on frozen ground and hydrological processes for a large mountainous area containing permafrost and non-permafrost areas. The model reasonably captured the thermal and hydrological processes, especially the seasonal and long-term variations of river discharge and runoff. The results show changes in permafrost extent and thawing/freezing deaths, and associated changes in hydrological processes in this large area. The spatially distributed modelling approach is novel and efficient for such a large and cold region as well. This work is valuable to demonstrate the progress in high-resolution thermal-hydrological integrated spatial modelling for large cold regions and to understand the impacts of climate change on frozen ground and associated hydrological processes. Although I agree with some of the concerns indicated by the previous reviewer (see details below), I feel it is worthy to be published after a revision.

**Specific comments**

**1.** I agree with the concern of the previous reviewer that the almost exact simulation of ground temperatures at deep layers at the test sites (Figure 3, and Figure S1) probably is the results of calibration, i.e., setting the initial values. The paper should indicate that and probably needs to revise the phase "generally accurate" to a looser term. In addition, if there is no geothermal heat flux at the bottom, ground temperature profile in lower ground should not vary much with depth under equilibrium conditions (unless the 10-year climate force used to spin-up varied significantly from year to year). Most simulated and observed soil temperature increased with depth, indicating the existence of a geothermal gradient. The assumption of zero geothermal heat flux at the lower boundary (Line 279) seems not right.

**2.** The causal relationship between changes in frozen ground and runoff is an important issue and the paper tried to answer it. The high correlation between liquid soil moisture and runoff in freezing season is not enough to establish that causal relationship (Line 471-474). The modelling exercise of without frozen soil and Figure S2 are a direct way to show the effects of frozen ground and its thaw. More detailed explanation of the modelling exercise needs to be provided (e.g., how the model was modified to do that? is this a run for a grid or for the entire basin?). Figure 15 is interesting but not so clear for me. An analysis from typical grids (permafrost and non-permafrost grids) and seasonal patters (e.g., Figure S2) might be helpful to understand it.

**Minor comments**

**1.** Line 30: "active layer depth". Using "active layer thickness" for consistency in the paper

**2.** Line 34: "large changes in runoff", can you specify "change" as "increase"?

**3.** Line 38-39: "due to the degradation of permafrost in the study area". Increase in precipitation probably also contributed to that change".

**4.** Line 47: revise "regime" to "regimes"

**5.** Line 51: delete "the".

**6.** Line 59: "…the frozen soil, and the long-term…", separate it into two sentences.

**7.** Line 71: "…processes especially …", add a comma after "processes".

**8.** Lines 85-86. Unpack it into two sentences. "The Qinhai-Tibetan Plateau is characterized by… . Cryospheric processes have great impacts on its hydrological processes".

**9.** Line 88: "permafrost thickness", delete "thickness"

**10.** Line 91: delete "the"

**11.** Line 103-110: Agree with the previous reviewer, delete it.

**12.** Section 2: probably no needs for the sub-titles. Just describe the study area, field observations and spatial data. Try to focus on data description rather than how they are used (leave that in the next section).

**13.** Line 135-136: moving "provided by ..(CMA)" to the end of the sentence.

**14.** Line 141: revise "build" to "run"

**15.** Line 173-175: "in an integrated … 2016)" delete it.

**16.** Line 176: The approach of the model is very interesting. It is not a fully distributed model with lateral interactions but efficient in computation and handling water flows. I feel that is a important progress of spatial modelling. You may add some sentences about the feature of the model (not branding the project or funding) before "As…".

**17.** Line 188 revise "used in" to "of"

**18.** Line281-285: You used thinner layers around 0.8m and from 1.7-3m, probably to capture the details of maximum thawing/freezing depths. You mad add some explanations about that.

**19.** Section 3.3: you may begin the section by "to initialize the model, we first estimated the soil temperature profiles based on the assumption that ….". You may delete the sentence "For spin up run, the initial … this relationship".

**20.** Line 318-319: the good agreement probably is due to calibration of initial values. A 500 year spin up run should change to a near constant ground temperature with depth. You need to check the model or whether the climate data from 1961-1970 vary significantly from year to year that prevent the establishment of equilibrium conditions.

**21.** Line 338-339. The value of RMSE and variation with depth is comparable with the study of Ou et al. (2016).

**22.** Line 353: revise "station" to "stations"

**23.** Line 366: "without the frozen soil scheme". How the model was modified? and is the Figure S2 for the entire basin or just a grid? This is an important part of directly show the effects frozen ground on runoff. More description is needed, probably in the method section.

**24.** Section 4.2: It would be useful include the trends of air temperature (annual, thawing and freezing seasons) in the analysis.

**25.** Line 407: "In contrast". Not a real contrast. It is expected. Delete it.

**26.** Line 434 "accurately reproduced" may be replace by "well reproduced"

**27.** Line 453: is increasing in precipitation a factor?

**28.** Line 455-469. It is easy to understand that the volume of soil liquid water increases with the increase in the volume of unfrozen soils. The sentences in these lines are long and complicated. You may simplify them.

**29.** Lines 471-474: You need more evidence to support the causal relations. The correlation is only one evidence. See the specific comments.

**30.** Line 482: From Table 2, the increase probably is mainly due to increase in snow run off in thaw season.

**31.** Line 484-485: revise "during the different seasons" to "between the two seasons".

**32.** Line 499, 506, 512: "change in frozen soil". You may specify it as "thaw of frozen soil"

**33.** Line 508: "was controlled by" the word probably is too strong. You may use "strongly affected by"

**34.** Line 512 revise "soil moisture" to "soil liquid moisture"

**35.** Line 540-522: "Different methods produce large differences in their estimates". The following citations do not support such a statement since they are mainly for different areas. Actually, some of the estimates in Qinghai-Tibetan Plateau is comparable with your estimate, which is a support of your estimate.

**36.** Line 568: "especially in spring", not clear for me.

**37.** The sediment thickness (depth from surface to bedrock), top organic layer thickness, and fraction of rock in soil are important to ground temperature and permafrost. You may add some sentences about them in sections of data, discussion or uncertainty. Active layer is thinner in valleys than in high slopes and on top of mountains due to differences in organic layer and vegetation conditions (Zhang et al., 2013). Temperature inversion and shading by surroundings may also keep the valley cooler than top of the mountains (O'Neil et al., 2015). You may comments on this in the discussions.

**38.** Figures: The font of words or numbers are too small in most figures.

**39.** Figure 3, S1: It is better to use a line with dots to represent the observations (so readers know the depths of observations). If you have annual averages, it is better to use annual averages rather than a specific date or month.

**40.** Figure 10b: revise "thaw depth" to "active layer thickness"

**41.** Figure 11d,e, Red curves are not necessary. For easy understanding, you may put elevation as Y axis, and percentage of permafrost to x axis.

**2. Author's response**

**Reply to the Referee Comment by Anonymous Referee #1**

**General comments:**

The paper is improved. I have only minor specific comments other than the general comment below.
In general, in this paper I would like the authors to address if there is a horizontal talik above the permafrost or not. Surely there would be somewhere in the catchment especially where the permafrost is discontinuous. Horizontal taliks exist where the maximum frost depth does not extend as deep as the maximum seasonal thaw depth. So there is a perennially unfrozen zone above the permafrost but below the active layer. Could this be significant like the conversion from permafrost to seasonally frozen ground? Would the model account for this?

**Reply**: Thanks for this comment. We have checked the results of soil temperature of each layer for all grids. We have found some taliks in permafrost region. But the horizontal taliks are not significant as shown in the following figure.

[Figure]

**Minor comments:**

**Q[1]**: L51, delete 'the'.
**Reply**: We have revised as suggested (Please see line 51 in the revised clean version manuscript ).

**Q[2]**: L56, delete 'the'
**Reply**: We have revised as suggested (Please see line 56 in the revised clean version manuscript).

**Q[3]**: L57,58, the authors should be careful with the phrase 'field experiments' here. If they are talking about looking at field data, some of the studies (e.g., Jacques and Sauchyn 2009) have been done on large spatial scales. If they are talking about intensively monitoring a catchment, that is different and likely only accomplished on fine scales.
**Reply**: We have modified "field experiments" as "intensive field observations" (Please see line 58-59 in the revised clean version manuscript).

**Q[4]:** L62, delete 'the'.
**Reply**: We have revised as suggested (Please see line 63 in the revised clean version manuscript).

**Q[5]:** L31 and many other places in the manuscript. If I were the authors I would define runoff up front. They use it in the normal hydrological way, but many hydrogeologists or civil engineers could be interested in this paper, and runoff often means different things to different people. Given that The Cryosphere is a bit of a general cold regions journal, I'd suggest a clear precise definition early on in the paper for runoff.
**Reply**: Thank you for this suggestion, we added the runoff definition when it first appeared in this paper. The related sentence has been revised as "A few studies reported that permafrost thawing might reduce river runoff (This paper defines the runoff as all liquid water flowing out of the study area.)". Please see line 57 in the revised clean version manuscript.

**Q[6]:** L70, delete 'the' before 'vertical'.
**Reply**: We have revised as suggested (Please see line 71 in the revised clean version manuscript).

**Q[7]:** L72, 'overly simple ways' is vague. Explain or remove?
**Reply**: We modified "in overly simple ways" as "by simplified ways" (Please see line 72 in the revised clean version manuscript).

**Q[8]:** L83, delete 'the' before 'downstream'
**Reply**: We have revised as suggested (Please see line 83 in the revised clean version manuscript).

**Q[9]:** L88, 'thickness' can be removed (besides it makes no sense to say that the thickness is thin and the thickness is warm).
**Reply**: We have revised as suggested (Please see line 88 in the revised clean version manuscript).

**Q[10]:** L91, delete 'the' after 'air'.
**Reply**: We have revised as suggested (Please see line 90 in the revised clean version manuscript).

**Q[11]:** L103-107 and 172-175. This seems to be almost branding for the research grant and, in my mind, is only suitable for the acknowledgements not the main body of a research paper.
**Reply**: we have delete the sentences about the Heihe Research plan in the main body (Please see line 103-106 and line 161-164 in the revised clean version manuscript).

**Q[12]:** L122, 'mean' should be before 'annual'.
**Reply:** We have revised (Please see line 119 in the revised clean version manuscript).

**Q[13]:** I'm surprised by the high RMSE in the soil temperature results, especially after calibration'
**Reply:** We calibrated the soil reflectance according to vegetation type, and we do not calibrate the soil heat capacity and soil thermal conductivity. The soil heat capacity and soil thermal conductivity are estimated using the method developed by Farouki (1981). This may lead uncertainties in simulation of the soil temperature.

**Q[14]:** Section 5.3, the uncertainty 'analysis' is highly qualitative.
**Reply:** We have quantified the uncertainties of soil temperature simulation induced by geothermal flux (Please see line 576-580 in the revised clean version manuscript). However, due to the complexity of the distributed model and large number of model parameters, it is challenge to quantify overall simulation uncertainty. This work will be done in the future study.

**Q[15]:** Figure 1, The authors could show the plateau on the map of China. OR be clearer what the grey zone refers to in the inset.?
**Reply:** We have modified Figure 1 to show the plateau on the map of China (Please see Figure 1 in the revised clean version manuscript).

**Q[16]:** Figure 7, the caption should be clearer why there are groups of 2 figure panels for each location.
**Reply:** We have changed the caption of Figure 7 as "Comparison of the simulated and the observed daily river discharge at: (a) the Yingluoxia Gauge, (b) the Qilian Gauge, and (c) the Zhamashike Gauge (The upper panel is the calibration period, and the bottom panel is the validation period for each gauge)". Please see Figure 7 in the revised clean version manuscript.

**Reply to the Referee Comment by Anonymous Referee #2**

**General comments:**

This paper reports a modelling study about the impacts of climate warming on frozen ground and hydrological processes for a large mountainous area containing permafrost and non-permafrost areas. The model reasonably captured the thermal and hydrological processes, especially the seasonal and long-term variations of river discharge and runoff. The results show changes in permafrost extent and thawing/freezing deaths, and associated changes in hydrological processes in this large area. The spatially distributed modelling approach is novel and efficient for such a large and cold region as well. This work is valuable to demonstrate the progress in high-resolution thermal-hydrological integrated spatial modelling for large cold regions and to understand the impacts of climate change on frozen ground and associated hydrological processes. Although I agree with some of the concerns indicated by the previous reviewer (see details below), I feel it is worthy to be published after a revision.

**Reply**: Thanks for this positive comment. We have revised the manuscript according to your suggestions.

**Specific comments:**

**Q[1]**: I agree with the concern of the previous reviewer that the almost exact simulation of ground temperatures at deep layers at the test sites (Figure 3, and Figure S1) probably is the results of calibration, i.e., setting the initial values. The paper should indicate that and probably needs to revise the phase "generally accurate" to a looser term. In addition, if there is no geothermal heat flux at the bottom, ground temperature profile in lower ground should not vary much with depth under equilibrium conditions (unless the 10-year climate force used to spin-up varied significantly from year to year). Most simulated and observed soil temperature increased with depth, indicating the existence of a geothermal gradient. The assumption of zero geothermal heat flux at the lower boundary (Line 279) seems not right.

**Reply:** We have deleted "accurate", and changed the expression as "The model generally captured the vertical distribution of the soil temperature at T1, T2, T3 and T4 in the permafrost area. Good agreement between the simulated and observed soil temperature profiles below the depth of 20 m is probably due to fitting of initial values." (Please see line 316-320 in the revised clean version manuscript).
We also recognized the fact of soil temperature increased with depth, indicating the existence of a geothermal gradient, which may cause uncertainty in our simulation. We have estimated the geothermal heat flux at the lower boundary and run the model using a heat flux of 0.2 W/m$^2$ at the lower boundary and compared the results with simulation with zero heat flux. The geothermal heat flux of 0.2 W/m$^2$ is estimated by geothermal gradient of T4 shown in Figure 3. And the geothermal gradient of T4 is larger than the other boreholes. The results are shown in Figure S5 in the supplement file. It can be seen from Figure S5 that geothermal heat flux only causes slightly increase in soil temperature below 30 m. We have added discussions about the uncertainties due to geothermal heat flux in section 5.3 as "Figure S5 in the supplement material compares the results of simulation with zero thermal flux at the lower boundary and the results of simulation with thermal flux of 0.2 W/m2 (Estimated by geothermal gradient at T4 in Figure 3). It can be seen that the geothermal heat flux at the lower boundary causes slight increase in soil temperature below the depth of 30 m." (Please see line 576-580 in the revised clean version manuscript).

**Q[2]**: The causal relationship between changes in frozen ground and runoff is an important issue and the paper tried to answer it. The high correlation between liquid soil moisture and runoff in freezing season is not enough to establish that causal relationship (Line 471-474). The modelling exercise of without frozen soil and Figure S2 are a direct way to show the effects of frozen ground and its thaw. More detailed explanation of the modelling exercise needs to be provided (e.g., how the model was modified to do that? is this a run for a grid or for the entire basin?). Figure 15 is interesting but not so clear for me. An analysis from typical grids (permafrost and non-permafrost grids) and seasonal patters (e.g., Figure S2) might be helpful to understand it.

**Reply:** For Figure S2, the result is obtained by a run for entire basin. We have modified the figure caption to indicate this. We have added explanations about how the model is modified to run without frozen soil in section 3.4 (Please see line 302-306 in the revised clean version manuscript). Figure S4 has been added in supplement files to show runoff changes in typical regions where the permafrost changed into seasonally frozen ground. We have also checked seasonal pattern of the runoff in the permafrost areas and the seasonally frozen soils, see Figure 15(c).

We discussed the runoff changes in section 5.1 as "Figure 15(c) shows the seasonal pattern of runoff in the permafrost area and seasonally frozen soils. From April to October (the thawing season), runoff in the permafrost area is much larger than in the seasonally frozen soils, but in the freezing season runoff in the permafrost area is lower than in the seasonally frozen soils. Figure S4 in the supplement material shows runoff changes from typical area (with elevation between 3500-3700 m) where covered by the permafrost in the period of 1971 to 1980 and changed into the seasonally frozen ground in the period of 2001 to 2010. This illustrates that thaw of permafrost increased the runoff in the freezing season and slowed recession processes in autumn. The increase in freezing season runoff and shift in the seasonal flow pattern are also illustrated by the model simulation without frozen soil scheme as shown in Figure S2." (Please see line 481-491 in the revised clean version manuscript)

**Minor comments:**

**Q[1]**: Line 30: "active layer depth". Using "active layer thickness" for consistency in the paper
**Reply**: We have revised as suggested (Please see line 30 in the revised clean version manuscript).

**Q[2]**: Line 34: "large changes in runoff", can you specify "change" as "increase"?
**Reply**: Yes, we have use "increase" instead of "change" (Please see line 34 in the revised clean version manuscript).

**Q[3]**: Line 38-39: "due to the degradation of permafrost in the study area". Increase in precipitation probably also contributed to that change".
**Reply**: Increase in precipitation may also contribute to this change, but degradation of permafrost is the major reason. We modified this sentence as "mainly due to the degradation of permafrost in the study area" (Please see line 38-39 in the revised clean version manuscript).

**Q[4]**: Line 47: revise "regime" to "regimes"

**Reply**: We have revised as suggested (Please see line 48 in the revised clean version manuscript).

**Q[5]**: Line 51: delete "the".
**Reply**: We have revised as suggested (Please see line 51 in the revised clean version manuscript ).

**Q[6]**: Line 59: "…the frozen soil, and the long-term…", separate it into two sentences.
**Reply**: We have revised as suggested (Please see line 60 in the revised clean version manuscript).

**Q[7]**: Line 71: "…processes especially …", add a comma after "processes".
**Reply**: We have modified this sentence as "but they represent the flow routing at the catchment scale by simplified ways" according to the comment of reviewer 1 (Please see line 72 in the revised clean version manuscript).

**Q[8]**: Lines 85-86. Unpack it into two sentences. "The Qinhai-Tibetan Plateau is characterized by… . Cryospheric processes have great impacts on its hydrological processes".
**Reply**: We have revised as suggested (Please see line 85-87 in the revised clean version manuscript).

**Q[9]**: Line 88: "permafrost thickness", delete "thickness"
**Reply**: We have revised as suggested (Please see line 88 in the revised clean version manuscript).

**Q[10]**: Line 91: delete "the"
**Reply**: We have revised as suggested (Please see line 90 in the revised clean version manuscript).

**Q[11]**: Line 103-110: Agree with the previous reviewer, delete it.
**Reply**: We have delete the sentences about the Heihe Research plan as suggested (Please see line 103-106 in the revised clean version manuscript).

**Q[12]**: Section 2: probably no needs for the sub-titles. Just describe the study area, field observations and spatial data. Try to focus on data description rather than how they are used (leave that in the next section).
**Reply**: We have revised as suggested (Please see line 114, line 127-130, line 148-149 in the revised clean version manuscript).

**Q[13]**: Line 135-136: moving "provided by ..(CMA)" to the end of the sentence.
**Reply**: We have revised as suggested (Please see line 132-133 in the revised clean version manuscript).

**Q[14]**: Line 141: revise "build" to "run"
**Reply**: We have revised as suggested (Please see line 138 in the revised clean version manuscript).

**Q[15]**: Line 173-175: "in an integrated … 2016)" delete it.
**Reply**: We have revised as suggested (Please see line 161-163 in the revised clean version manuscript).

**Q[16]**: Line 176: The approach of the model is very interesting. It is not a fully distributed model with lateral interactions but efficient in computation and handling water flows. I feel that is a important progress of spatial modelling. You may add some sentences about the feature of the model (not branding the project or funding) before "As…".
**Reply**: We have added some sentences as "GBEHM is a spatial distributed model for large-scale river basin. It employs the geomorphologic properties to reduce the lateral two-dimensions into one-dimension for flow routing calculation within a sub-catchment, which greatly improves the computation efficiency while retaining the spatial heterogeneity in water flow paths at basin scale." (Please see line 164-168 in the revised clean version manuscript)

**Q[17]**: Line 188 revise "used in" to "of"

**Reply**: We have revised as suggested (Please see line 179 in the revised clean version manuscript).

**Q[18]**: Line281-285: You used thinner layers around 0.8m and from 1.7-3m, probably to capture the details of maximum thawing/freezing depths. You mad add some explanations about that.
**Reply**: We have added a sentence as "As shown in Figure 2, thinner layers are used at the depth from 1.7 to 3 m for better capturing the maximum frozen depth according to the field observations." (Please see line 274-275 in the revised clean version manuscript).

**Q[19]**: Section 3.3: you may begin the section by "to initialize the model, we first estimated the soil temperature profiles based on the assumption that ....". You may delete the sentence "For spin up run, the initial … this relationship".
**Reply**: We have revised as suggested (Please see line 285-289 in the revised clean version manuscript).

**Q[20]**: Line 318-319: the good agreement probably is due to calibration of initial values. A 500 year spin up run should change to a near constant ground temperature with depth. You need to check the model or whether the climate data from 1961-1970 vary significantly from year to year that prevent the establishment of equilibrium conditions.
**Reply**: We have checked the climate data, air temperature rising started from 1980s. We have changed this sentence as "Good agreement between the simulated and observed soil temperature profiles below the depth of 20 m is probably due to fitting of initial values" (Please see line 318-320 in the revised clean version manuscript).

**Q[21]**: Line 338-339. The value of RMSE and variation with depth is comparable with the study of Ou et al. (2016).
**Reply**: We have added a sentence as "This result is similar with the findings by Ou et al. (2016) using the Northern Ecosystem Soil Temperature (NEST) model." (Please see line 342-343 in the revised clean version manuscript).

**Q[22]**: Line 353: revise "station" to "stations"
**Reply**: We have revised as suggested (Please see line 356 in the revised clean version manuscript).

**Q[23]**: Line 366: "without the frozen soil scheme". How the model was modified? and is the Figure S2 for the entire basin or just a grid? This is an important part of directly show the effects frozen ground on runoff. More description is needed, probably in the method section.
**Reply**: We added some explanations about the model modification in section 3.4. Figure S2 is for entire basin (Please see line 302-306 in the revised clean version manuscript).

**Q[24]**: Section 4.2: It would be useful include the trends of air temperature (annual, thawing and freezing seasons) in the analysis.

**Reply**: We have added a table in supplement file (Table S2) and discussions in the manuscript as "Table S2 in the supplement material shows that annual mean air temperature increased by approximately 1.2℃ in the period of 2001 to 2010 comparing with the period of 1971 to 1980. And air temperature in the freezing season shows larger increase (approximately 1.4℃) than in the thawing season (1.1℃) between the two periods." (Please see line 384-387 in the revised clean version manuscript).

**Q[25]**: Line 407: "In contrast". Not a real contrast. It is expected. Delete it.
**Reply**: We have revised as suggested (Please see line 414 in the revised clean version manuscript).

**Q[26]**: Line 434 "accurately reproduced" may be replace by "well reproduced"
**Reply**: We have revised as suggested (Please see line 441 in the revised clean version manuscript).

**Q[27]**: Line 453: is increasing in precipitation a factor?
**Reply**: Increasing precipitation may be a factor, but it is not the major factor.

**Q[28]**: Line 455-469. It is easy to understand that the volume of soil liquid water increases with the increase in the volume of unfrozen soils. The sentences in these lines are long and complicated. You may simplify them.
**Reply**: We have simplified this part as suggested (Please see line 462-472 in the revised clean version manuscript).

**Q[29]**: Lines 471-474: You need more evidence to support the causal relations. The correlation is only one evidence. See the specific comments.
**Reply**: We have added new figures and analysis to support the causal relations as mentioned above (Please see line 481-491 in the revised clean version manuscript).

**Q[30]**: Line 482: From Table 2, the increase probably is mainly due to increase in snow run off in thaw season.
**Reply**: We have revised this sentence as "The increased runoff mainly came from increased precipitation and snowmelt in the thawing season." (Please see line 495-496 in the revised clean version manuscript)

**Q[31]**: Line 484-485: revise "during the different seasons" to "between the two seasons".
**Reply**: We have revised as suggested (Please see line 498-499 in the revised clean version manuscript).

**Q[32]**: Line 499, 506, 512: "change in frozen soil". You may specify it as "thaw of frozen soil"

**Reply**: We have revised as suggested (Please see line 513, 519, 526 in the revised clean version manuscript).

**Q[33]**: Line 508: "was controlled by" the word probably is too strong. You may use "strongly affected by"

**Reply**: We have revised as suggested (Please see line 522 in the revised clean version manuscript).

**Q[34]**: Line 512 revise "soil moisture" to "soil liquid moisture"

**Reply**: We have revised as suggested (Please see line 526 in the revised clean version manuscript).

**Q[35]**: Line 540-522: "Different methods produce large differences in their estimates". The following citations do not support such a statement since they are mainly for different areas. Actually, some of the estimates in Qinghai-Tibetan Plateau is comparable with your estimate, which is a support of your estimate.

**Reply**: We have deleted the citations (Jorgenson et al., 2006 and Chasmer et al., 2010) for other areas and only compared our results with estimates in the Qinghai-Tibetan Plateau. A new citation of Guo et al. (2013) for the change of permafrost area in the Qinghai-Tibetan Plateau is added. We have deleted the sentence "Different methods produce large differences in their estimates" (Please see line 555-561 in the revised clean version manuscript).

**Q[36]**: Line 568: "especially in spring", not clear for me.

**Reply**: Here this means high groundwater flow rate events such as spring freshet. To make it more clear. We modified this as "especially when high groundwater flow rate events occur" (Please see line 589 in the revised clean version manuscript).

**Q[37]**: The sediment thickness (depth from surface to bedrock), top organic layer thickness, and fraction of rock in soil are important to ground temperature and permafrost. You may add some sentences about them in sections of data, discussion or uncertainty. Active layer is thinner in valleys than in high slopes and on top of mountains due to differences in organic layer and vegetation conditions (Zhang et al., 2013). Temperature inversion and shading by surroundings may also keep the valley cooler than top of the mountains (O'Neil et al., 2015). You may comments on this in the discussions.

**Reply**: We have added some discussions about this in section 5.3 as "The uncertainty in simulation of soil heat-water transfer processes might result from the soil water and heat parameters and the bottom boundary condition of heat flux. For example, soil depth and fraction of rock in soil may greatly affect soil temperature simulation." and "Sub-grid topography may also affect the frozen soil simulation. For example, active layer thickness is different in the low valleys and high slopes due to different vegetation conditions, soil organic layers and shading by surroundings (Zhang et al., 2013; O'Neill et al., 2015). These factors are not well considered in this study." (Please see line 571-574 and line 582-585 in the revised clean version manuscript).

**Q[38]**: Figures: The font of words or numbers are too small in most figures.
**Reply**: We will change the font of words or numbers in the figures to make it clear (Please see Figure 5, Figure 6, Figure 7, Figure 8, Figure 10, Figure 11, Figure 12 and Figure 13 in the revised clean version manuscript).

**Q[39]**: Figure 3, S1: It is better to use a line with dots to represent the observations (so readers know the depths of observations). If you have annual averages, it is better to use annual averages rather than a specific date or month.
**Reply**: We have modified the figure as suggested. We do not have annual averages. Please see Figure 3 and Figure S1 in the revised clean version manuscript.

**Q[40]**: Figure 10b: revise "thaw depth" to "active layer thickness"

**Reply**: We have modified the figure as suggested. Please see Figure 10 in the revised clean version manuscript.

**Q[41]:** Figure 11d,e, Red curves are not necessary. For easy understanding, you may put elevation as Y axis, and percentage of permafrost to x axis.
**Reply**: We have modified the figure as suggested. Please see Figure 11 in the revised clean version manuscript.

**3. Author's changes in manuscript**

1. Add figure S4 in the supplement file to show the runoff changes in typical regions.

2. Add Table S2 in the supplement file to show the changes in air temperature.

3. Add Figure S5 to show the effect of geothermal flux at the lower boundary.

4. Modified Figure 3 and Figure S1 to use a line with dots to represent the observations.

5. Modified Figure 5, Figure 6, Figure 7, Figure 8 and Figure 10, Figure 12, Figure 13 to use larger fonts.

6. Add Seasonal patterns of runoff in permafrost area and seasonally frozen ground in Figure 15.

7. Add introductions about the model without frozen soil scheme in section 3.4.

8. Add analysis about the runoff changes in typical regions and seasonal pattern changes in section 5.1.

9. Add discussions about uncertainty caused by geothermal flux at the lower boundary in section 5.3.

10. Delete sub-titles in Section 2 and focus on data description.

11. Modified Figure 1 to show the Plateau on the map

12. Modified Figure 11d,e and put elevation as Y axis, and percentage of permafrost to x axis

13. Add some discussions in section 5.3 about the factors which influences the soil temperature simulation.

14. Some other changes according to the minor comments of two reviewers

[revised manuscript text omitted]

Note: T means total runoff, G means glacier runoff and S means snowmelt runoff.

---

## Editor Decision (ED1)

Thank you again for your revised manuscript entitled "Change in Frozen Soils and Its Effect on Regional Hydrology in the Upper Heihe Basin, on the Northeastern Qinghai-Tibetan Plateau" (tc-2017-158).

The revised material adequately incorporates most of the reviewer's comments, and should be suitable for publication in TC with further revision. Most revisions stem from points raised by the referee's comments, but there is still a question that remains in relation to tc-2016-289 with respect to code availability, data used, and scripts used to perform the modelling. Most figures also need some work to generate a common look and feel. The revisions are for the most part minor, but will take some time.

**General comments**

With respect to Referee #1

Referee #1 makes the general comment that taliks are to be expected, and you responded that talik development is not substantial and include a figure. However, I do not see that you addressed the question in the paper. Please incorporate the finding into your results, and discuss the implications in your discussion. I agree with the reviewer that taliks are often expected when permafrost changes to seasonally frozen ground. I wonder if the lack of taliks is a relict of modelling and therefore an underestimate, or if there is a likely explanation that is physically based. The relevant figure is probably best included in your supplementary material.

Referee #1 [13]: Your reply is adequate, but you need a line of explanation in the manuscript to reflect your answer.

With respect to Referee #2

Referee #2 [Q1]: The important issue is raised here that an assumption of zero heat flux is simply not intuitive, and I expect that this assumption will continue to raise questions. Indeed, Fig. 12b seems to suggest a bottom-up degradation of permafrost as thaw depths are comparatively invariant. Top-down thaw would lead to more widespread talik development as expected by Referee #1. Is this pattern of permafrost degradation highlighted in Fig. 12b a function of the model calibration and spin up, or is it related to actual increases in freezing season air temperatures whereas summer thaw season air temperatures are relatively stable. This is an issue that needs to be addressed clearly.

We need a paper that is strong, and without distractions so that the important points shine through. I strongly suggest that you revise the manuscript with the general assumption that there **is** a geothermal heat flux, and abandon any comparison with a model scenario that does not include such a flux. This is a major revision that will affect figures and text. Are there no deep boreholes in the region, or heat flux models for the region, from which to obtain an estimate? If so, please look into using them. Your estimate of 0.2 W/m$^2$ seems reasonable, but how does this compare with published values for QTP (e.g. Wu et al., 2010, Global and Planetary Change: 72: 32-38)?

Referee #2 [Q37]: Please re-visit the text with respect to this question and include discussion about temperature inversion. The effect is not related to vegetation or soil conditions, but relates to accumulation of cold, dense air in valleys. Bonnaventure et al. (2012), Permafrost and Periglacial

Processes, 23: 52-68) incorporated inverted surface lapse rates in their model of Yukon Territory permafrost distribution, and it may be a useful reference for you.

Referee #2 [Q39]: Regarding annual averages, please indicate in the text why you did not use annual averages, but instead had to rely on measurements from specific dates. Please discuss any implications due to this choice.

With respect to code, data and scripts

Your novel approach was of interest to the reviewers, and will be to other readers who will want to apply the approach to new areas, or test model-to-model results, or examine the reproducibility of experiments, uncertainties, and goodness of fit. I suggest that you indicate where the model code, data, and scripts used are publicly available.

With respect to figures and tables

The figures require a common appearance so the work does not look like the figures were drafted by different co-authors. This includes figures in the supplementary material.

Use similar colours to show similar things. For example, simulation and observation should keep the same color coding in all figures. See Fig 5 versus Figs. 4 and 6.

Some figures have boxes around panels, while others do not. Please be consistent.

Graph axes: tic marks inside or outside? Some figure panels are only enclosed on 2 sides, while the majority have 4 sides.

Font sizes are often too small: Fig. 2; Fig. 3; Fig. 4; Fig. 6; Fig. 7; Fig. 11, panels c, d, and e; Fig. 12; Fig. 13.

Fig. 15 fonts and overall scale is much larger than the rest.

Axes labels, panel titles, and tables: Label/title and text within brackets need to be separated by a space. E.g., "Depth(m)" becomes "Depth (m)" or "(a)T1 (2011-09-25,4132 m)" becomes "(a) T1 (2011-09-25,4132 m)".  Carefully check Fig.2b , Fig. 3; Fig. 5; Fig. 6; Fig. 7 (Precipitation); Fig. 9; Fig. 10a; Fig. 11; Fig. 12; Fig. 13 (Actual evapotranspiration); Fig. 14; Fig. 15.

A period "." Is required at the end of the last sentence of most figure captions.

Specific comments

Throughout: change passive tense to active tense. E.g. line 415, change "an increasing trend of active layer thickness in the permafrost regions was observed (3.5 cm/10yr),which had a significantly positive correlation with annual mean air temperature." to " Simulated active layer thickness in permafrost

regions increased (3.5 cm·decade), and correlates positively with annual mean air temperature (p=XXXX)." Indicate the level of significance.

Throughout: Please carefully reduce the word count. This is a long manuscript that can be written more succinctly.

Throughout: Please refer to "supplementary material" rather than "supplemental file" or supplement material".

Throughout: please convert cm values to either mm or m.

Throughout the text, figures, and tables: please be consistent in how units are related to each other. E.g., "mol·m$^{-2}$·s$^{-1}$" versus "mol/m2/s". The former is preferred.

Throughout: "Soil temperature" is used throughout, but you really mean "ground temperatures". Soil implies weathering, etc., that is unlikely at great depths. This change likely affects figures, captions, and the main text.

Line 1. Suggest changing title to "entitled "Change in Frozen Soils and the Effects on Regional Hydrology, Upper Heihe Basin, Northeastern Qinghai-Tibetan Plateau"

Line 38: Change "degradation" to reduction in permafrost extent". Existing text could imply that ground ice in permafrost is contributing to groundwater recharge.

Lines 58 to 63: Sentence are still not clear. Perhaps change to: "Intensive field observations on frozen soils were typically carried out a small spatial scales over short periods. Consequently, regional patterns and long-term trends are not captured. Long-term meteorological and hydrological observations are available, but they do not provide information on soil freezing and thawing processes …"

Lines 69 to 72. Both reviewers took issue with this sentence. It is still too vague. Please delete "by simplified ways" and provide some explanation of the simplifications.

Lines 86. Change to "Consequently, cryospheric …"

Line 88. What is meant by "thin and warm"? Report thicknesses and temperature ranges published in the literature.

Line 163.  Change "based on the" to "from a"

Lines 274-275. Delete sentence and work idea into text on Line 279.

Line 279. Change to "… with a constant thickness of 10 cm to try to replicate the maximum freezing depths according to field observations."

Line 342. Uncertainties in the simulations may relate to the estimates of ground heat capacity and thermal conductivity derived according to Farouki (1981), but the results are similar to the findings of Ou et al. (2016) …"

Lines 384 to 387. Change to "Compared to the decadal mean for 1971 to 1980, mean air temperature for the 2001 to 2010 period increased by approximately 1.2 °C, with a larger increase in the freezing season (1.4 °C) than in the thawing season (1.1 °C) (Figure 9 and Table S2).

Lines 465 to 472. These sentence are not well written and do not read easily. Please revise.

Line 478. Delete comma after frozen ground.

Lines 477 to 480. Change "decreased, which led" to "decreased, leading"

Line 482. Change "in the permafrost area and seasonally frozen soils" to "from the entire basin".

Lines 494 to 496. Re-write and combine sentences so that it reads more easily.

Lines497 to 499. Delete the first 2 sentences and change to "Figure 15 shows the large difference in runoff variation with elevation between the freezing and thawing seasons."

Line 526. Change order of words: "soil liquid" to "liquid soil'.

Lines 555 to 557. Indicate the year the decrease was observed.

Lines 582 to 585. Include potential for temperature inversion in this discussion.

Line 586: change "lateral heat" to laterally advected heat"

Line 589: Change "when high groundwater flow rate events occur" to "where groundwater flow rates are high."

Figure 1.White background conveys no information/context for meteorological stations. If you show the colorized DEM (elevation) for the whole panel, the study area will remain obvious due to the encircling black polygon.

Figure 3. Too much wasted space. Try to reduce figure size. Move panel titles inside of the panels. Keep temperature scales the same; really only need 4 degrees of freedom in each figure, or keep a uniform temperature range of -2 to 4 °C. Depth scale range in e and f are half of a-d. for comparative purposes it would be helpful if all depth scales were the same range, 2-44 m. Panel e, "°C" is offset below the axis title.

Figure 4. Figure labels: second and subsequent words are not to be capitalized. E.g., "Soil Depth (m)" becomes "Soil depth (m)". Dates shown on x-axis are annual. Simplify labels to show only the year. Axis title can be changed to "Year".  Change color scale in panels a and b so that the 0 °C isotherm is clear.

Color scale used in Fig. 12 is good. Plotting the isotherm as a black line would also help. In caption change "Simulation-Observation" to "difference (simulation – observation).".

Figure 5. Use annual increments on x-axis, label every 2nd or 5th year, and title "Year".

Figure 6. Panels are all too small and time series lines too thin. Does not reproduce well as a result. Perhaps move panel titles inside the panel to give more room. Show monthly tic marks, but label every second one, or label "J F M A M J J A S O N D". Figure caption: change "… Sunny slope station." To "…Sunny Slope station (2014 calendar year). Root mean square errors are indicated."

Figure 7. Indicate within every panel if it is a Calibration or Validation period, and perhaps enclose each pair in a box. Change caption to "…the Yingluoxia gauge, (b) the Qilian gauge, and (c) the Zhamashike gauge. For each gauge, the upper and lower panels show the calibration and validation periods, respectively. Nash-Sutcliffe efficiency and relative error coefficients are indicated."

Figure 8. Plot tic marks for each year. No need to indicate "-01" for month. X-axis title "Year". Change caption to "comparison of simulated monthly evapotranspiration with a remote-sensing-derived estimate (Wu, 2013) for the period of 2002 to 2012."

Figure 9. Y-axes in both panels should share the same scaling ratio so that the figure highlights the fact that freezing season temperatures are increasing at greater rates than thawing season temperatures. Time series labels: Space between depth interval and unit. Change caption to "Simulated ground temperature changes in: (a) … and (b) …". Include a line about the linear regressions. What is the statistical significance of the slopes?

Figure 10. Panel b time series labels: change "Frozen depth of Seasonally frozen ground" to "Seasonally frozen depth"". Change "Active layer thickness of permafrost" to "Thaw depth". Change caption text to "… annual maximum depths of seasonally frozen ground and thaw above permafrost." Include a line about the linear regressions. What is the statistical significance of the slopes?

Figure 11. Tic marks on panels d and e are not visible. Panel d: Capitalize "Sunny". Change caption to "Distribution of permafrost and seasonally frozen ground for two periods: (a) 1971-1980 and (b) 2001-2010. (c) Area where permafrost degraded to seasonally frozen ground between the two periods. Percentage of permafrost area for the two periods with respect to elevation on slopes that are (d) sunny or (e) shaded. Note that (d) and (e) share a legend."

Figure 12. Change caption to "Spatially averaged monthly ground temperatures simulated from 1971 to 2013 for two elevation intervals: (a) seasonally frozen ground between 3300 and 3500 m; (b) permafrost that degraded to seasonally frozen ground between 3500 and 3700 m." Show annual tic marks on x-axis, but perhaps label every 2nd or 5th year.

Figure 13. This figure needs work. Caption says actual evapotranspiration but the data are for simulated evapotranspiration. It is not clear which two panels are paired together. Labels are missing. Tic mark intervals and labels are different though time scale ranges are the same. Change caption to "Runoff and simulated evapotranspiration in (a) the freezing season and (b) the thawing season." Either report trend

lines and significance in caption or in the figure, or remove the trend lines. Are trend lines in the left-hand side panels for the simulated or observed data? This needs to be clear.

Figure 14. These are simulation results. Are these basin-averaged? Change caption to "(Basin averaged?) Annual water storage (equivalent water depth) changes simulated over the period of 1971 to 2013 for: (a) liquid water in the top layer of the ground (0-3 m); (b) ice in the top layer of the ground (0-3 m); (c) and ground water." Indicate if trend lines are significant.

Figure 15. Needs work. Look and feel is quite different than other figures. Panel a is missing a properly scaled and labeled x-axis. There is a typo in panel c. Change caption to "Model simulated runoff changes from the 1971-1980 period to the 2001-2010 period with elevation for (a) the freezing season and (b) the thawing season, and (c) monthly averaged seasonal runoff in permafrost and seasonally frozen ground for the period of 2001 to 2010.

Table 2. Several column headings show words that are split across lines.

Figure S1. Capitalize "simulation" in legend. Add line to caption "Legend in (a) applies to all panels.

Figure S2. Change "Obs" to "Observations". X-axis time scale should be adjust to even spacing by months.

Figure S3. Tic marks on x-axis should indicate years, with every 2$^{nd}$ or 5$^{th}$ labelled. Figure caption should be re-worded in a similar manner as Figure 12.

---

## Author Response (AR2)

**1. Comments from Editor**

The revised material adequately incorporates most of the reviewer's comments, and should be suitable for publication in TC with further revision. Most revisions stem from points raised by the referee's comments, but there is still a question that remains in relation to tc-2016-289 with respect to code availability, data used, and scripts used to perform the modelling. Most figures also need some work to generate a common look and feel. The revisions are for the most part minor, but will take some time.

**General comments**

**With respect to Referee #1**

**1.** Referee #1 makes the general comment that taliks are to be expected, and you responded that talik development is not substantial and include a figure. However, I do not see that you addressed the question in the paper. Please incorporate the finding into your results, and discuss the implications in your discussion. I agree with the reviewer that taliks are often expected when permafrost changes to seasonally frozen ground. I wonder if the lack of taliks is a relict of modelling and therefore an underestimate, or if there is a likely explanation that is physically based. The relevant figure is probably best included in your supplementary material.

**2.** Referee #1 [13]: Your reply is adequate, but you need a line of explanation in the manuscript to reflect your answer.

**With respect to Referee #2**

**1.** Referee #2 [Q1]: The important issue is raised here that an assumption of zero heat flux is simply not intuitive, and I expect that this assumption will continue to raise questions. Indeed, Fig. 12b seems to suggest a bottom-up degradation of permafrost as thaw depths are comparatively invariant. Top-down thaw would lead to more widespread talik development as expected by Referee #1. Is this pattern of permafrost degradation highlighted in Fig. 12b a function of the model calibration and spin up, or is it related to actual increases in freezing season air temperatures whereas summer thaw season air temperatures are relatively stable. This is an issue that needs to be addressed clearly.

**2.** We need a paper that is strong, and without distractions so that the important points shine through. I strongly suggest that you revise the manuscript with the general assumption that there is a geothermal heat flux, and abandon any comparison with a model scenario that does not include such a flux. This is a major revision that will affect figures and text. Are there no deep boreholes in the region, or heat flux models for the region, from which to obtain an estimate? If so, please look into using them. Your estimate of 0.2 W/m2 seems reasonable, but how does this compare with published values for QTP (e.g. Wu et al., 2010, Global and Planetary Change: 72: 32-38)?

**3.**  Referee #2 [Q37]: Please re-visit the text with respect to this question and include discussion about temperature inversion. The effect is not related to vegetation or soil conditions, but relates to accumulation of cold, dense air in valleys. Bonnaventure et al. (2012), Permafrost and Periglacial Processes, 23: 52-68) incorporated inverted surface lapse rates in their model of Yukon Territory permafrost distribution, and it may be a useful reference for you.

**4.**  Referee #2 [Q39]: Regarding annual averages, please indicate in the text why you did not use annual averages, but instead had to rely on measurements from specific dates. Please discuss any implications due to this choice.

**With respect to code, data and scripts**

Your novel approach was of interest to the reviewers, and will be to other readers who will want to apply the approach to new areas, or test model-to-model results, or examine the reproducibility of experiments, uncertainties, and goodness of fit. I suggest that you indicate where the model code, data, and scripts used are publicly available.

**With respect to figures and tables**

**1.**  The figures require a common appearance so the work does not look like the figures were drafted by different co-authors. This includes figures in the supplementary material.

**2.**  Use similar colours to show similar things. For example, simulation and observation should keep the same color coding in all figures. See Fig 5 versus Figs. 4 and 6.

**3.**  Some figures have boxes around panels, while others do not. Please be consistent.

**4.**  Graph axes: tic marks inside or outside? Some figure panels are only enclosed on 2 sides, while the majority have 4 sides.

**5.**  Font sizes are often too small: Fig. 2; Fig. 3; Fig. 4; Fig. 6; Fig. 7; Fig. 11, panels c, d, and e; Fig. 12; Fig. 13.

**6.**  Fig. 15 fonts and overall scale is much larger than the rest.

**7.**  Axes labels, panel titles, and tables: Label/title and text within brackets need to be separated by a space. E.g., "Depth(m)" becomes "Depth (m)" or "(a)T1 (2011-09-25,4132 m)" becomes "(a) T1 (2011-09-25,4132 m)". Carefully check Fig.2b , Fig. 3; Fig. 5; Fig. 6; Fig. 7 (Precipitation); Fig. 9; Fig. 10a; Fig. 11; Fig. 12; Fig. 13 (Actual evapotranspiration); Fig. 14; Fig. 15.

**8.**  A period "." Is required at the end of the last sentence of most figure captions.

**Specific comments**

Throughout: change passive tense to active tense. E.g. line 415, change "an increasing trend of active layer thickness in the permafrost regions was observed (3.5 cm/10yr),which had a significantly positive correlation with annual mean air temperature." to " Simulated active layer thickness in permafrost regions increased (3.5 cm·decade), and correlates positively with annual mean air temperature (p=XXXX)." Indicate the level of significance.

Throughout: Please carefully reduce the word count. This is a long manuscript that can be written more succinctly.

Throughout: Please refer to "supplementary material" rather than "supplemental file" or supplement material".

Throughout: please convert cm values to either mm or m.

Throughout the text, figures, and tables: please be consistent in how units are related to each other. E.g., "mol·m-2·s-1" versus "mol/m2/s". The former is preferred.

Throughout: "Soil temperature" is used throughout, but you really mean "ground temperatures". Soil implies weathering, etc., that is unlikely at great depths. This change likely affects figures, captions, and the main text.

Line 1. Suggest changing title to "entitled "Change in Frozen Soils and the Effects on Regional Hydrology, Upper Heihe Basin, Northeastern Qinghai-Tibetan Plateau"

Line 38: Change "degradation" to reduction in permafrost extent". Existing text could imply that ground ice in permafrost is contributing to groundwater recharge.

Lines 58 to 63: Sentence are still not clear. Perhaps change to: "Intensive field observations on frozen soils were typically carried out a small spatial scales over short periods. Consequently, regional patterns and long-term trends are not captured. Long-term meteorological and hydrological observations are available, but they do not provide information on soil freezing and thawing processes …"

Lines 69 to 72. Both reviewers took issue with this sentence. It is still too vague. Please delete "by simplified ways" and provide some explanation of the simplifications.

Lines 86. Change to "Consequently, cryospheric …"

Line 88. What is meant by "thin and warm"? Report thicknesses and temperature ranges published in the literature.

Line 163. Change "based on the" to "from a"

Lines 274-275. Delete sentence and work idea into text on Line 279.

Line 279. Change to "… with a constant thickness of 10 cm to try to replicate the maximum freezing depths according to field observations."

Line 342. Uncertainties in the simulations may relate to the estimates of ground heat capacity and thermal conductivity derived according to Farouki (1981), but the results are similar to the findings of Ou et al. (2016) …"

Lines 384 to 387. Change to "Compared to the decadal mean for 1971 to 1980, mean air temperature for the 2001 to 2010 period increased by approximately 1.2 °C, with a larger increase in the freezing season (1.4 °C) than in the thawing season (1.1 °C) (Figure 9 and Table S2).

Lines 465 to 472. These sentence are not well written and do not read easily. Please revise.

Line 478. Delete comma after frozen ground.

Lines 477 to 480. Change "decreased, which led" to "decreased, leading"

Line 482. Change "in the permafrost area and seasonally frozen soils" to "from the entire basin".

Lines 494 to 496. Re-write and combine sentences so that it reads more easily.

Lines497 to 499. Delete the first 2 sentences and change to "Figure 15 shows the large difference in runoff variation with elevation between the freezing and thawing seasons."

Line 526. Change order of words: "soil liquid" to "liquid soil'.

Lines 555 to 557. Indicate the year the decrease was observed.

Lines 582 to 585. Include potential for temperature inversion in this discussion.
Line 586: change "lateral heat" to laterally advected heat"

Line 589: Change "when high groundwater flow rate events occur" to "where groundwater flow rates are high."

Figure 1.White background conveys no information/context for meteorological stations. If you show the colorized DEM (elevation) for the whole panel, the study area will remain obvious due to the encircling black polygon.

Figure 3. Too much wasted space. Try to reduce figure size. Move panel titles inside of the panels. Keep temperature scales the same; really only need 4 degrees of freedom in each figure, or keep a uniform temperature range of -2 to 4 °C. Depth scale range in e and f are half of a-d. for comparative purposes it would be helpful if all depth scales were the same range, 2-44 m. Panel e, "°C" is offset below the axis title.

Figure 4. Figure labels: second and subsequent words are not to be capitalized. E.g., "Soil Depth (m)" becomes "Soil depth (m)". Dates shown on x-axis are annual. Simplify labels to show only the year. Axis title can be changed to "Year". Change color scale in panels a and b so that the 0 °C isotherm is clear. Color scale used in Fig. 12 is good. Plotting the isotherm as a black line would also help. In caption change "Simulation-Observation" to "difference (simulation – observation).".

Figure 5. Use annual increments on x-axis, label every 2nd or 5th year, and title "Year".

Figure 6. Panels are all too small and time series lines too thin. Does not reproduce well as a result. Perhaps move panel titles inside the panel to give more room. Show monthly tic marks, but label every second one, or label "J F M A M J J A S O N D". Figure caption: change "… Sunny slope station." To "…Sunny Slope station (2014 calendar year). Root mean square errors are indicated."

Figure 7. Indicate within every panel if it is a Calibration or Validation period, and perhaps enclose each pair in a box. Change caption to "…the Yingluoxia gauge, (b) the Qilian gauge, and (c) the Zhamashike gauge. For each gauge, the upper and lower panels show the calibration and validation periods, respectively. Nash-Sutcliffe efficiency and relative error coefficients are indicated."

Figure 8. Plot tic marks for each year. No need to indicate "-01" for month. X-axis title "Year". Change caption to "comparison of simulated monthly evapotranspiration with a remote-sensing-derived estimate (Wu, 2013) for the period of 2002 to 2012."

Figure 9. Y-axes in both panels should share the same scaling ratio so that the figure highlights the fact that freezing season temperatures are increasing at greater rates than thawing season temperatures. Time series labels: Space between depth interval and unit. Change caption to "Simulated ground temperature changes in: (a) … and (b) …". Include a line about the linear regressions. What is the statistical significance of the slopes?

Figure 10. Panel b time series labels: change "Frozen depth of Seasonally frozen ground" to "Seasonally frozen depth"". Change "Active layer thickness of permafrost" to "Thaw depth". Change caption text to "… annual maximum depths of seasonally frozen ground and thaw above permafrost." Include a line about the linear regressions. What is the statistical significance of the slopes?

Figure 11. Tic marks on panels d and e are not visible. Panel d: Capitalize "Sunny". Change caption to "Distribution of permafrost and seasonally frozen ground for two periods: (a) 1971-1980 and (b) 2001-2010. (c) Area where permafrost degraded to seasonally frozen ground between the two periods. Percentage of permafrost area for the two periods with respect to elevation on slopes that are (d) sunny or (e) shaded. Note that (d) and (e) share a legend."

Figure 12. Change caption to "Spatially averaged monthly ground temperatures simulated from 1971 to 2013 for two elevation intervals: (a) seasonally frozen ground between 3300 and 3500 m; (b) permafrost that degraded to seasonally frozen ground between 3500 and 3700 m." Show annual tic marks on x-axis, but perhaps label every 2nd or 5th year.

Figure 13. This figure needs work. Caption says actual evapotranspiration but the data are for simulated evapotranspiration. It is not clear which two panels are paired together. Labels are missing. Tic mark intervals and labels are different though time scale ranges are the same. Change caption to "Runoff and simulated evapotranspiration in (a) the freezing season and (b) the thawing season." Either report trend lines and significance in caption or in the figure, or remove the trend lines. Are trend lines in the left-hand side panels for the simulated or observed data? This needs to be clear.

Figure 14. These are simulation results. Are these basin-averaged? Change caption to "(Basin averaged?) Annual water storage (equivalent water depth) changes simulated over the period of 1971 to 2013 for: (a) liquid water in the top layer of the ground (0-3 m); (b) ice in the top layer of the ground (0-3 m); (c) and ground water." Indicate if trend lines are significant.

Figure 15. Needs work. Look and feel is quite different than other figures. Panel a is missing a properly scaled and labeled x-axis. There is a typo in panel c. Change caption to "Model simulated runoff changes from the 1971-1980 period to the 2001-2010 period with elevation for (a) the freezing season and (b) the thawing season, and (c) monthly averaged seasonal runoff in permafrost and seasonally frozen ground for the period of 2001 to 2010.

Table 2. Several column headings show words that are split across lines.

Figure S1. Capitalize "simulation" in legend. Add line to caption "Legend in (a) applies to all panels.

Figure S2. Change "Obs" to "Observations". X-axis time scale should be adjust to even spacing by months.

Figure S3. Tic marks on x-axis should indicate years, with every 2nd or 5th labelled. Figure caption should be re-worded in a similar manner as Figure 12.

**2. Author's responses**

The revised material adequately incorporates most of the reviewer's comments, and should be suitable for publication in TC with further revision. Most revisions stem from points raised by the referee's comments, but there is still a question that remains in relation to tc-2016-289 with respect to code availability, data used, and scripts used to perform the modelling. Most figures also need some work to generate a common look and feel. The revisions are for the most part minor, but will take some time.

**Reply:** Thanks for handling our manuscript and the suggestions to revise the manuscript. Following comments from the editor, we have substantially revised our manuscript. The details are given bellow.

**General comments**

**With respect to Referee #1**

**1.** Referee #1 makes the general comment that taliks are to be expected, and you responded that talik development is not substantial and include a figure. However, I do not see that you addressed the question in the paper. Please incorporate the finding into your results, and discuss the implications in your discussion. I agree with the reviewer that taliks are often expected when permafrost changes to seasonally frozen ground. I wonder if the lack of taliks is a relict of modelling and therefore an underestimate, or if there is a likely explanation that is physically based. The relevant figure is probably best included in your supplementary material.

**Reply:** Thanks for your suggestion. We have added the relevant result in the supplementary material as Figure S5.

We added a sentence in section 4.2 as "Figure S5, illustrating the taliks simulated in the period of 2001-2010, shows that the taliks were mainly located on the edge of the permafrost area and the development of taliks was not significant." (Please see line 430-432 in the revised clean version manuscript).

We also discussed the talik development in the manuscript as "The laterally advected heat flux may increase the thawing of permafrost, especially in areas with high groundwater flow rates (Kurylyk et al., 2016; Sjöberg et al., 2016). Not considering the lateral heat flux may lead to an underestimation of talik development and thawing rates of permafrost." (Please see line 583-586 in the revised clean version manuscript)

**2.** Referee #1 [13]: Your reply is adequate, but you need a line of explanation in the manuscript to reflect your answer.

**Reply:** Thank you for this comment. We have added a sentence as "Uncertainties in the simulations may be related to the ground heat capacity and thermal conductivity estimated according to Farouki (1981), and the results are similar to the findings by Ou et al. (2016) using the Northern Ecosystem Soil Temperature (NEST) model." (Please see line 349-352 in the revised clean version manuscript)

**With respect to Referee #2**

**1.** Referee #2 [Q1]: The important issue is raised here that an assumption of zero heat flux is simply not intuitive, and I expect that this assumption will continue to raise questions. Indeed, Fig. 12b seems to suggest a bottom-up degradation of permafrost as thaw depths are comparatively invariant. Top-down thaw would lead to more widespread talik development as expected by Referee #1. Is this pattern of permafrost degradation highlighted in Fig. 12b a function of the model calibration and spin up, or is it related to actual increases in freezing season air temperatures whereas summer thaw season air temperatures are relatively stable. This is an issue that needs to be addressed clearly.

**Reply:** Thank you for this comment. We considered a geothermal heat flux and re-run the model. The related figures and texts are updated.

We realized that the thaw depths changed slowly comparing with the frozen depths. The main reason may be the effect of geothermal heat flux. Air temperature increase in the freezing season ( $0.41\,℃$ decadal$^{-1}$) is much larger than the thawing season ( $0.26\,℃$ decade$^{-1}$), which may be another reason. We have updated the related results in the revised manuscript and explained as "The thaw depths changed slowly compared with the frozen depths as shown in Figure 10, which may be primarily due to the geothermal heat flux. Additionally, the faster increase in the air temperature in the freezing season (0.41 ℃ decade-1) than in the thawing season (0.26 ℃ decade-1) may be another reason." (Please see line 441-445 in the revised clean version manuscript).

**2.** We need a paper that is strong, and without distractions so that the important points shine through. I strongly suggest that you revise the manuscript with the general assumption that there is a geothermal heat flux, and abandon any comparison with a model scenario that does not include such a flux. This is a major revision that will affect figures and text. Are there no deep boreholes in the region, or heat flux models for the region, from which to obtain an estimate? If so, please look into using them. Your estimate of 0.2 W/m2 seems reasonable, but how does this compare with published values for QTP (e.g. Wu et al., 2010, Global and Planetary Change: 72: 32-38)?

**Reply:** As mentioned above, we considered a geothermal heat flux and re-run the model. There are no deep boreholes in the study area. We estimated upward geothermal heat flux as 0.14 W m$^{-2}$ at a depth of 50 m by the average geothermal gradient at 4 boreholes (T1-T4) shown in Figure 3, which is reasonable comparing with the observations along Qinghai-Tibet Highway/Railway in the interior QTP (vary from 0.02 W m$^{-2}$ to 0.16 W m$^{-2}$) from the published literature (Wu et al., 2010), Please see line 275-280 in the revised clean version manuscript.

We have deleted comparison with a model scenario that does not include such a flux and revised the related part in the manuscript.

**3.** Referee #2 [Q37]: Please re-visit the text with respect to this question and include discussion about temperature inversion. The effect is not related to vegetation or soil conditions, but relates to accumulation of cold, dense air in valleys. Bonnaventure et al. (2012), Permafrost and Periglacial Processes, 23: 52-68) incorporated inverted surface lapse rates in their model of Yukon Territory permafrost distribution, and it may be a useful reference for you.

**Reply:** We have modified the text as "Sub-grid topography may also affect the frozen soil simulation. For example, active layer thickness is different between the low-elevation valleys and higher-elevation slopes due to the temperature inversion caused by the accumulation of cold air in valleys (Bonnaventure et al., 2012; Zhang et al., 2013; O'Neill et al., 2015)." (Please see line 579-583 in the revised clean version manuscript).

**4.** Referee #2 [Q39]: Regarding annual averages, please indicate in the text why you did not use annual averages, but instead had to rely on measurements from specific dates. Please discuss any implications due to this choice.

**Reply:** We have no data to estimate the annual average soil temperature profiles due to lack of continuous measurement. We have added a sentence to explain as "We used the observations at specific dates instead of annual averages due to lack of continuous measurement." (Please see line 154-155 in the revised clean version manuscript).

**With respect to code, data and scripts**

Your novel approach was of interest to the reviewers, and will be to other readers who will want to apply the approach to new areas, or test model-to-model results, or examine the reproducibility of experiments, uncertainties, and goodness of fit. I suggest that you indicate where the model code, data, and scripts used are publicly available.

**Reply:** We have added a sentence in the Acknowledgements as "All data for this paper are properly cited and referred in the reference list. The model code with a working example is freely available from our website (https://github.com/gb03/GBEHM) or upon request from the corresponding author (yangdw@tsinghua.edu.cn)" (Please see line 638-641 in the revised clean version manuscript). And we will continue to work on the website in the future.

**With respect to figures and tables**

**1.** The figures require a common appearance so the work does not look like the figures were drafted by different co-authors. This includes figures in the supplementary material.

**Reply:** Thank you for this comment. We have revised all the figures.

**2.** Use similar colours to show similar things. For example, simulation and observation should keep the same color coding in all figures. See Fig 5 versus Figs. 4 and 6.

**Reply:** We have modified the figures to using red colors for simulation and black color for observation.

**3.** Some figures have boxes around panels, while others do not. Please be consistent.

**Reply:** We have changed all the figures to keep them consistent.

**4.** Graph axes: tic marks inside or outside? Some figure panels are only enclosed on 2 sides, while the majority have 4 sides.
**Reply:** We have modified the figures and make all tic marks inside.

**5.** Font sizes are often too small: Fig. 2; Fig. 3; Fig. 4; Fig. 6; Fig. 7; Fig. 11, panels c, d, and e; Fig. 12; Fig. 13.
**Reply:** We have modified the figures to use larger fonts.

**6.** Fig. 15 fonts and overall scale is much larger than the rest.
**Reply:** We have modified the figure and used the same font size as other figures.

**7.** Axes labels, panel titles, and tables: Label/title and text within brackets need to be separated by a space. E.g., "Depth(m)" becomes "Depth (m)" or "(a)T1 (2011-09-25,4132 m)" becomes "(a) T1 (2011-09-25,4132 m)". Carefully check Fig.2b , Fig. 3; Fig. 5; Fig. 6; Fig. 7 (Precipitation); Fig. 9; Fig. 10a; Fig. 11; Fig. 12; Fig. 13 (Actual evapotranspiration); Fig. 14; Fig. 15.
**Reply:** We have revised as suggested.

**8.** A period "." Is required at the end of the last sentence of most figure captions.
**Reply:** We have revised as suggested.

**Specific comments**

**1.** Throughout: change passive tense to active tense. E.g. line 415, change "an increasing trend of active layer thickness in the permafrost regions was observed (3.5 cm/10yr),which had a significantly positive correlation with annual mean air temperature." to " Simulated active layer thickness in permafrost regions increased (3.5 cm·decade), and correlates positively with annual mean air temperature (p=XXXX)." Indicate the level of significance.
**Reply:** We have changed passive tense to active tense in the whole manuscript.

**2.** Throughout: Please carefully reduce the word count. This is a long manuscript that can be written more succinctly.
**Reply:** Thank you for this comment. We have tried our best to reduce the length of the manuscript.

**3.** Throughout: Please refer to "supplementary material" rather than "supplemental file" or supplement material".
**Reply:** We have revised as suggested.

**4.** Throughout: please convert cm values to either mm or m.
**Reply:** We have converted cm values to m.

**5.** Throughout the text, figures, and tables: please be consistent in how units are related to each other. E.g., "mol·m-2·s-1" versus "mol/m2/s". The former is preferred.
**Reply:** We have revised as suggested.

**6.** Throughout: "Soil temperature" is used throughout, but you really mean "ground temperatures". Soil implies weathering, etc., that is unlikely at great depths. This change likely affects figures, captions, and the main text.
**Reply:** We have used ground temperature instead of soil temperature as suggested.

**7.** Line 1. Suggest changing title to "entitled "Change in Frozen Soils and the Effects on Regional Hydrology, Upper Heihe Basin, Northeastern Qinghai-Tibetan Plateau"
**Reply:** We have changed the title as suggested.

**8.** Line 38: Change "degradation" to reduction in permafrost extent". Existing text could imply that ground ice in permafrost is contributing to groundwater recharge.
**Reply:** We have revised as suggested (Please see line 38-39 in the revised clean version manuscript).

**9.** Lines 58 to 63: Sentence are still not clear. Perhaps change to: "Intensive field observations on frozen soils were typically carried out a small spatial scales over short periods. Consequently, regional patterns and long-term trends are not captured. Long-term meteorological and hydrological observations are available, but they do not provide information on soil freezing and thawing processes …"
**Reply:** Thanks for this suggestion. We have revised as suggested (Please see line 58-63 in the revised clean version manuscript)

**10.** Lines 69 to 72. Both reviewers took issue with this sentence. It is still too vague. Please delete "by simplified ways" and provide some explanation of the simplifications.
**Reply:** We have changed this sentence as "but they simplify the flow routing using linear scheme." (Please see 71-72 in in the revised clean version manuscript).

**11.** Lines 86. Change to "Consequently, cryospheric …"
**Reply:** We have revised as suggested (Please see line 86-87 in the revised clean version manuscript).

**12.** Line 88. What is meant by "thin and warm"? Report thicknesses and temperature ranges published in the literature.
**Reply:** We modified this sentence as **"**the thickness of permafrost on the Qinghai-Tibetan Plateau ranges 1-130 m and the temperature varies from -0.5 to -3.5 °C (Yang et al., 2010)" (Please see line 87-89 in the revised clean version manuscript).

**13.** Line 163. Change "based on the" to "from a"
**Reply:** We have delete this sentence to reduce the length of the manuscript.

**14.** Lines 274-275. Delete sentence and work idea into text on Line 279.

**Reply:** We have revised as suggested (Please see line 284-286 in the revised clean version manuscript)

**15.** Line 279. Change to "… with a constant thickness of 10 cm to try to replicate the maximum freezing depths according to field observations."

**Reply:** We have revised as suggested (Please see line 284-286 in the revised clean version manuscript)

**16.** Line 342. Uncertainties in the simulations may relate to the estimates of ground heat capacity and thermal conductivity derived according to Farouki (1981), but the results are similar to the findings of Ou et al. (2016) …"

**Reply:** We have revised as suggested (Please see line 349-352 in the revised clean version manuscript)

**17.** Lines 384 to 387. Change to "Compared to the decadal mean for 1971 to 1980, mean air temperature for the 2001 to 2010 period increased by approximately 1.2 °C, with a larger increase in the freezing season (1.4 °C) than in the thawing season (1.1 °C) (Figure 9 and Table S2).

**Reply:** We have revised as suggested (Please see line 393-396 in the revised clean version manuscript)

**18.** Lines 465 to 472. These sentence are not well written and do not read easily. Please revise.

**Reply:** We have revised these sentences as suggested (Please see line 481-486 in the revised clean version manuscript).

**19.** Line 478. Delete comma after frozen ground.

**Reply:** We have revised as suggested (Please see line 492 in the revised clean version manuscript)

**20.** Lines 477 to 480. Change "decreased, which led" to "decreased, leading"

**Reply:** We have revised as suggested (Please see line 491-494 in the revised clean version manuscript)

**21.** Line 482. Change "in the permafrost area and seasonally frozen soils" to "from the entire basin".

**Reply:** We have revised as suggested (Please see line 496 in the revised clean version manuscript)

**22.** Lines 494 to 496. Re-write and combine sentences so that it reads more easily.

**Reply:** We have revised as suggested (Please see line 462-464 in the revised clean version manuscript)

**23.** Lines497 to 499. Delete the first 2 sentences and change to "Figure 15 shows the large difference in runoff variation with elevation between the freezing and thawing seasons."

**Reply:** We have revised as suggested (Please see line 506-507 in the revised clean version manuscript)

**24.** Line 526. Change order of words: "soil liquid" to "liquid soil'.

**Reply:** We have revised as suggested (Please see line 533-534 in the revised clean version manuscript)

**25.** Lines 555 to 557. Indicate the year the decrease was observed.

**Reply:** We have indicate the year the decrease was observed in the manuscript (Please see line 560 in the revised clean version manuscript).

**26.** Lines 582 to 585. Include potential for temperature inversion in this discussion.

**Reply:** We have revised as suggested (Please see line 579-583 in the revised clean version manuscript)

**27.** Line 586: change "lateral heat" to laterally advected heat"

**Reply:** We have revised as suggested (Please see line 583 in the revised clean version manuscript)

**28.** Line 589: Change "when high groundwater flow rate events occur" to "where groundwater flow rates are high."

**Reply:** We have revised as suggested (Please see line 584 in the revised clean version manuscript)

**29.** Figure 1. White background conveys no information/context for meteorological stations. If you show the colorized DEM (elevation) for the whole panel, the study area will remain obvious due to the encircling black polygon.

**Reply:** We have revised this figure as suggested (Please see Figure 1 in the revised clean version manuscript).

**30.** Figure 3. Too much wasted space. Try to reduce figure size. Move panel titles inside of the panels. Keep temperature scales the same; really only need 4 degrees of freedom in each figure, or keep a uniform temperature range of -2 to 4 °C. Depth scale range in e and f are half of a-d. for comparative purposes it would be helpful if all depth scales were the same range, 2-44 m. Panel e, "°C" is offset below the axis title.

**Reply:** We have revised this figure as suggested (Please see Figure 3 in the revised clean version manuscript).

**31.** Figure 4. Figure labels: second and subsequent words are not to be capitalized. E.g., "Soil Depth (m)" becomes "Soil depth (m)". Dates shown on x-axis are annual. Simplify labels to show only the year. Axis title can be changed to "Year". Change color scale in panels a and b so that the 0 °C isotherm is clear. Color scale used in Fig. 12 is good. Plotting the isotherm as a black line would also help. In caption change "Simulation-Observation" to "difference (simulation – observation).".

**Reply:** We have revised this figure as suggested (Please see Figure 4 in the revised clean version manuscript).

**32.** Figure 5. Use annual increments on x-axis, label every 2nd or 5th year, and title "Year".

**Reply:** We have revised this figure as suggested (Please see Figure 5 in the revised clean version manuscript).

**33.** Figure 6. Panels are all too small and time series lines too thin. Does not reproduce well as a result. Perhaps move panel titles inside the panel to give more room. Show monthly tic marks, but label every second one, or label "J F M A M J J A S O N D". Figure caption: change "… Sunny slope station." To "…Sunny Slope station (2014 calendar year). Root mean square errors are indicated."

**Reply:** We have revised this figure as suggested (Please see Figure S2 in the revised supplement materials).

**34.** Figure 7. Indicate within every panel if it is a Calibration or Validation period, and perhaps enclose each pair in a box. Change caption to "…the Yingluoxia gauge, (b) the Qilian gauge, and (c) the Zhamashike gauge. For each gauge, the upper and lower panels show the calibration and validation periods, respectively. Nash-Sutcliffe efficiency and relative error coefficients are indicated."

**Reply:** We have revised this figure as suggested (Please see Figure 6 in the revised clean version manuscript).

**35.** Figure 8. Plot tic marks for each year. No need to indicate "-01" for month. X-axis title "Year". Change caption to "comparison of simulated monthly evapotranspiration with a remote-sensing-derived estimate (Wu, 2013) for the period of 2002 to 2012."

**Reply:** We have revised this figure as suggested (Please see Figure S3 in the revised supplement materials).

**36.** Figure 9. Y-axes in both panels should share the same scaling ratio so that the figure highlights the fact that freezing season temperatures are increasing at greater rates than thawing season temperatures. Time series labels: Space between depth interval and unit. Change caption to "Simulated ground temperature changes in: (a) … and (b) …". Include a line about the linear regressions. What is the statistical significance of the slopes?

**Reply:** We have revised this figure as suggested and the statistical significance is shown in the figure (Please see Figure 7 in the revised clean version manuscript).

**37.** Figure 10. Panel b time series labels: change "Frozen depth of Seasonally frozen ground" to "Seasonally frozen depth"". Change "Active layer thickness of permafrost"

to "Thaw depth". Change caption text to "… annual maximum depths of seasonally frozen ground and thaw above permafrost." Include a line about the linear regressions. What is the statistical significance of the slopes?

**Reply:** We have revised this figure as suggested and the statistical significance is shown in the figure (Please see Figure 8 in the revised clean version manuscript).

**38.** Figure 11. Tic marks on panels d and e are not visible. Panel d: Capitalize "Sunny". Change caption to "Distribution of permafrost and seasonally frozen ground for two periods: (a) 1971-1980 and (b) 2001-2010. (c) Area where permafrost degraded to seasonally frozen ground between the two periods. Percentage of permafrost area for the two periods with respect to elevation on slopes that are (d) sunny or (e) shaded. Note that (d) and (e) share a legend."

**Reply:** We have revised this figure as suggested (Please see Figure 9 in the revised clean version manuscript).

**39.** Figure 12. Change caption to "Spatially averaged monthly ground temperatures simulated from 1971 to 2013 for two elevation intervals: (a) seasonally frozen ground between 3300 and 3500 m; (b) permafrost that degraded to seasonally frozen ground between 3500 and 3700 m." Show annual tic marks on x-axis, but perhaps label every 2nd or 5th year.

**Reply:** We have revised this figure as suggested (Please see Figure 10 in the revised clean version manuscript).

**40.** Figure 13. This figure needs work. Caption says actual evapotranspiration but the data are for simulated evapotranspiration. It is not clear which two panels are paired together. Labels are missing. Tic mark intervals and labels are different though time scale ranges are the same. Change caption to "Runoff and simulated evapotranspiration in (a) the freezing season and (b) the thawing season." Either report trend lines and significance in caption or in the figure, or remove the trend lines. Are trend lines in the left-hand side panels for the simulated or observed data? This needs to be clear.

**Reply:** We have revised this figure as suggested and the statistical significance is shown in the figure (Please see Figure 11 in the revised clean version manuscript).

**41.** Figure 14. These are simulation results. Are these basin-averaged? Change caption to "(Basin averaged?) Annual water storage (equivalent water depth) changes simulated over the period of 1971 to 2013 for: (a) liquid water in the top layer of the ground (0-3 m); (b) ice in the top layer of the ground (0-3 m); (c) and ground water." Indicate if trend lines are significant.

**Reply:** We have revised this figure as suggested the statistical significance is shown in the figure (Please see Figure 12 in the revised clean version manuscript).

**42.** Figure 15. Needs work. Look and feel is quite different than other figures. Panel a is missing a properly scaled and labeled x-axis. There is a typo in panel c. Change caption to "Model simulated runoff changes from the 1971-1980 period to the 2001- period with elevation for (a) the freezing season and (b) the thawing season, and (c) monthly averaged seasonal runoff in permafrost and seasonally frozen ground for the period of 2001 to 2010.

**Reply:** We have revised this figure as suggested (Please see Figure 13 in the revised clean version manuscript).

**43.** Table 2. Several column headings show words that are split across lines.

**Reply:** We have revised this table as suggested (Please see Table 2 in the revised clean version manuscript).

**44.** Figure S1. Capitalize "simulation" in legend. Add line to caption "Legend in (a) applies to all panels.

**Reply:** We have revised this figure as suggested (Please see Figure S1 in the revised supplement materials).

**45.** Figure S2. Change "Obs" to "Observations". X-axis time scale should be adjust to even spacing by months.

**Reply:** We have revised this figure as suggested (Please see Figure S4 in the revised supplement materials).

**46.** Figure S3. Tic marks on x-axis should indicate years, with every 2nd or 5th labelled. Figure caption should be re-worded in a similar manner as Figure 12.

**Reply:** We have revised this figure as suggested (Please see Figure S6 in the revised supplement materials).

**3. Author's changes in manuscript**

1. Re-run the model with consideration of the geothermal heat flux in the bottom boundary and update the related texts and figures.

2. Move Figure 6 and Figure 8 in the previous version to the supplement materials in order to reduce the length of the manuscript.

3. Add figure in the supplement materials to show the taliks.

4. Modified all the figures according to the comments.

5. Modified the manuscript according to the suggestions of a native speaker and reduce the manuscript length.

6. Add introductions about model code availability and data used in the Acknowledgement.

[revised manuscript text omitted]
 of ice ($\text{kg} \cdot /\text{m}^{3 \cdot 3}$); $\lambda_s$ is the thermal conductivity ($\text{W} \cdot \text{m}^{-1} \cdot \text{K}^{-1}$); $\rho_l$

is the density of liquid water ($\text{kg} \cdot /\text{m}^3 \text{m}^{-3}$); and $c_l$ is the specific heat of liquid water ($\text{J} \cdot \text{kg}^{-1} \cdot \text{K}^{-1}$). In addition, $q_l$ is the water flux between different soil layers ($\text{m} \cdot /\text{s}^{-1}$) and is solved using the 1-D vertical Richards equation. The unsaturated soil hydraulic conductivity is calculated using the modified van Genuchten's equation (Wang et al.,

2010), as follows:

$$K = f_{ice} K_{sat} (\frac{\theta_l - \theta_r}{\theta_s - \theta_r})^{1/2}[1-(1-(\frac{\theta_l - \theta_r}{\theta_s - \theta_r})^{-1/m})^m]^2 \qquad (9)$$

where $K$ is the unsaturated soil hydraulic conductivity ($\text{m} \cdot /\text{s}^{-1}$); $K_{sat}$ is the saturated soil hydraulic conductivity ($\text{m} \cdot /\text{s}^{-1}$); $\theta_l$ is the volumetric liquid water content; $\theta_s$ is the saturated water content; $\theta_r$ is the residual water content; $m$ is an empirical parameter in van Genuchten's equation and $f_{ice}$ is an empirical hydraulic conductivity reduction factor which that is calculated using soil temperature as follows (Wang et al.,

2010):

$$f_{ice} = \exp[-10(T_f - T_{soil})], \quad 0.05 \le fice \le 1 \qquad (10)$$

where $T_f$ is 273.15 K and $T_{soil}$ is the soil temperature.

Eq. (8) solves the soil temperature with the upper boundary condition as the heat flux into the uppermost soil layer. When the ground is not covered by snow, the heat flux from the atmosphere into the uppermost soil layer is expressed as follows (Oleson et al., 2010):

$$h = S_g + L_g - H_g - \lambda E_g + Q_R \tag{11}$$

where $h$ is the upper boundary heat flux into the soil layer (W·m$^{-2}$); $S_g$ is the solar radiation absorbed by the uppermost soil layer (W·m$^{-2}$); $L_g$ is the net long wave radiation absorbed by the ground (W·m$^{-2}$), $H_g$ is the sensible heat flux from the ground (W·m$^{-2}$); $\lambda E_g$ is the latent heat flux from the ground (W·m$^{-2}$); and $Q_R$ is the energy delivered by rainfall (W·/m$^{2}$). When the ground is covered by snow, the heat flux into the uppermost soil layer is calculated as follows:

$$h = I_p + G \tag{12}$$

where $I_p$ is the radiation that penetrates the snow cover, and $G$ is the heat conduction from the bottom snow layer to the uppermost soil layer. Eq (8) is solved using a finite difference scheme with an hourly time step, similar to the solution of Eq (4).

There are The values of geothermal heat flux obtained from the observations along

Qinghai-Tibet Highway/Railway in the interior QTP vary from 0.02 W m$^{-2}$ to 0.16 W

m$^{-2}$ in the published literature (Wu et al., 2010), but there is no data available observations of the geothermal heat flux for the northeastern QTP. To simulate the permafrost we consider an underground depth of 50 m. We assume an the bottom boundary condition as upward thermal heat flux at the bottom boundary and estimate its value as to be of 0.16 0.42 W·m$^{-2}$ at a depth of 50 m (Estimated using by the average geothermal gradient from the 4 boreholes (T1-T4) shown in Figure 3, which is reasonable based on a comparison comparing with the observations (0.02 W·m$^{-2}$ to 0.16

W·m$^{-2}$) infrom the interior of the QTP (Wu et al., 2010)).zero heat flux exchange due to the data limitation. This assumption may not be true because the observed soil temperature increased with depth in the deep layer. The vertical soil column is divided into 39 layers in the model (see Figure 2). As shown in Figure 2, thinner layers are used at the depth from 1.7 to 3 m for better capturing the maximum frozen depth according to the field observations. The 1.7 m topsoil layer of 1.7 m is subdivided into 9 layers.

The first layer is 0.05 cm, and the soil layer thickness increases with depth linearly from

0.05 cm to 0.30 cm up to theat a depths of 0.8 m and later then decreases linearly with depth to 0.10 cm up to theat a depths of 1.7 m. There are 12 soil layers with a constant thickness of 0.1 m from 1.7 m to 3.0 m with a constant thickness of 10 cmto try to replicate the maximum freezing depths according to field observations. From the depth of 3 m to 50 m, there are 18 layers with thicknesses increasing exponentially from 0.10

cm to 12 m. The liquid soil moisture, ice content, and soil temperature of each layer is calculated at each time step. The soil heat capacity and soil thermal conductivity are estimated using the method developed by Farouki (1981).

**3.3 Model calibration**

To initialize the model, we first estimated the soil temperature profiles based on the assumption that there is a linear relationship between the groundsoil temperature at a given depth below the surface and elevation at the same depth below surface. This temperature-elevation The relationship between groundsoil temperature at a specific is estimated from the observed ground temperature  
[revised manuscript text omitted]
|---|---|---|---|---|---|---|---|---|---|---|---|---|
| 1971-1980 | 439.1 | 2.81 | 4.51 | 143.8 | 0.33 | 0.35 | 18.5 | 0.0 | 0.0 | 5.06 | 3.5 | 13.8 |
| 1981-1990 | 492.8 | 300.08 | 88.25 | 174.1 | 0.35 | 0.38 | 20.5 | 0.0 | 0.0 | 8.10 | 3.1 | 27.28 |
| 1991-2000 | 471.0 | 7.16 | 1.19 | 157.4 | 0.33 | 0.34 | 20.5 | 0.0 | 0.0 | 1.74 | 3.8 | 18.24 |
| 2001-2010 | 504.3 | 9.40 | 0.96 | 174.3 | 0.35 | 0.36 | 26.2 | 0.0 | 0.0 | .73 | 3.7 | 24.81 |

Note: P means precipitation, E means actual evaporation, R means runoff, T means total runoff, G means glacier runoff and S means snowmelt runoff, Sim means simulation and Obs means observation.

---

## Editor Decision (ED2)

| 1 | Change | in | Frozen | Soils | and | the | Effects on | Region | al H | vdrology | . U | pper      |
|---|--------|----|--------|-------|-----|-----|------------|--------|------|----------|-----|-----------|
| - |        |    |        | 10 0  |     |     |            |        |      | ,        | , - | FF |

**Heihe Basin, Northeastern Qinghai-Tibetan Plateau 2**

Bing Gao1, Dawen Yang2\*, Yue Qin2, Yuhan Wang2, Hongyi Li3, Yanlin Zhang3, and 3

Tingjun Zhang4 4

- 1 School of Water Resources and Environment, China University of Geosciences, 5
- Beijing 100083, China 6
- 2 State Key Laboratory of Hydroscience and Engineering, Department of Hydraulic 7
- Engineering, Tsinghua University, Beijing 100084, China 8
- 3 Cold and Arid Regions Environmental and Engineering Research Institute, Chinese 9
- 10 Academy of Sciences, Lanzhou, Gansu 730000, China
- 4 Key Laboratory of West China's Environmental Systems (MOE), College of Earth 11

1

- and Environmental Sciences, Lanzhou University, Lanzhou, 730000, China 12
- 13
- 14 \* Corresponding author: Dawen Yang (vangdw@tsinghua.edu.cn)
- 15
- To be submitted to: The Cryosphere, August 2017 16

17

19 ABSTRACT:

23

25

20 Frozen ground has an important role in regional hydrological cycles and ecosystems,

21 especially on the Qinghai-Tibetan Plateau (QTP), which is characterized by high

22 elevations and a dry climate. This study modified a distributed physically based

24 changes in frozen ground and the effects on hydrology in the upper Heihe basin, which

hydrological model and applied it to simulate the long-term (from 1971 to 2013)

is located on the northeastern QTP. The model was carefully validated against data

26 obtained from multiple ground-based observations. Based on the model simulations,

27 we analyzed the changes in frozen soils and their effects on the hydrology. The results

28 showed that the permafrost area shrank by 8.8% (approximately 500 km2), especially

in areas with elevations between 3500 m and 3900 m. The maximum frozen depth of

30 seasonally frozen ground decreased at a rate of approximately 0.032 m decade-1, and

31 the active layer thickness over the permafrost increased by approximately 0.043

m decade-1. Runoff increased significantly during the cold season (November-March)
due to the increase in liquid soil moisture caused by rising soil temperatures. Areas in

which permafrost changed into seasonally frozen ground at high elevations showed especially large increases in runoff. Annual runoff increased due to increased precipitation, the base flow increased due to changes in frozen soils, and the actual evapotranspiration increased significantly due to increased precipitation and soil warming. The groundwater storage showed an increasing trend, indicating that a reduction in permafrost extent enhanced the groundwater recharge.

40 KEYWORDS: permafrost; seasonally frozen ground; soil moisture; ground

Comment [PDM1]: "particularly"

Comment [PDM2]: "distributed, physically-based"

Comment [PDM3]: "long-term (197 2013)"

| Comment [PDM4 | ]: Delete text |
|---------------|----------------|
| Comment [PDM5 | ]: Delete      |
| Comment [PDM6 | ]: Delete      |

| 1 | Comment [PDM7]: "spatio-tempora |
|---|---------------------------------|
|   | Comment [PDM8]: Delete          |
| 1 | Comment [PDM9]: "Our"           |
|   | Comment [PDM10]: "show"         |
| ١ | Comment [PDM11]: "area with"    |
| ١ | Comment [PDM12]: Delete         |
| ١ | Comment [PDM13]: "predominantl  |
| ١ | Comment [PDM14]: Delete         |

Comment [PDM15]: "an"

41 temperature; runoff

**42 **1. Introduction**

43 Global warming has led to significant changes in frozen soils, including both permafrost 44 and seasonally frozen ground at high latitudes and high elevations (Hinzman et al., 2013; 45 Cheng and Wu, 2007). Changes in frozen soils can greatly affect land-atmosphere interactions and the energy and water balances of the land surface (Subin et al., 2013; 46 47 Schuur et al., 2015), altering soil moisture, water flow pathways and stream flow 48 regimes (Walvoord and Kurylyk, 2016). Understanding the changes in frozen soils and 49 their impacts on regional hydrology is important for water resources management and ecosystem protection in cold regions. 50

| 51 | Previous studies based on either experimental observations or long-term                  |
|----|------------------------------------------------------------------------------------------|
| 52 | meteorological or hydrological observations have examined changes in frozen soils and    |
| 53 | their impacts on hydrology. Several studies reported that permafrost thawing might       |
| 54 | enhance base flow in the Arctic and the Subarctic (Walvoord and Striegl, 2007; Jacques   |
| 55 | and Sauchyn, 2009; Ye et al., 2009), as well as in northeastern China (Liu et al., 2003; |
| 56 | Duan et al., 2017). A few studies reported that permafrost thawing might reduce river    |
| 57 | runoff (here, runoff is defined as all liquid water flowing out of the study area),      |
| 58 | especially on the Qinghai-Tibetan Plateau (e.g., Qiu, 2012; Jin et al., 2009). Intensive |
| 59 | field observations of frozen soils have typically been performed at small spatial scales |
| 60 | over short periods. Consequently, regional patterns and long-term trends have not been   |
| 61 | captured. Long-term meteorological and hydrological observations are available, but      |
| 62 | they do not provide information on soil freezing and thawing processes (McClelland et    |
| 63 | al., 2004; Liu et al., 2003; Niu et al., 2011). Therefore, previous observation-based    |

Comment [PDM16]: " (QTP) " Comment [PDM17]: "Those studies that include intensive"

Comment [PDM18]: "(e.g., Cite an example or two)" Comment [PDM19]: "are not typica Comment [PDM20]: "available in many areas" 64 studies have not provided a sufficient understanding of the long-term changes in frozen

65 soils and their impact on regional hydrology (Woo et al., 2008).

[revised manuscript text omitted]
_{k} \frac{\partial T_{k}}{\partial \rho_{k}} = \frac{\partial}{\partial r_{k}} (K_{k} \frac{\partial T}{\partial r_{k}}) + \frac{\partial I_{R}}{\partial r_{k}} + Q_{R}$ (4)

$$\partial s \partial t$$
  $\partial t \partial z \partial z \partial z$

217 where  $C_s$  is the heat capacity of snow (J-m-3K-1);  $T_s$  is the temperature of the snow

layer (K);  $\rho_i$  is the density of ice (kg-m-3);  $\theta_i$  is the volumetric ice content;  $K_s$ 218 is the thermal conductivity of snow (W  $\cdot$ m-1K-1); Lf is the latent heat of ice fusion (J  $\cdot$ kg-1 219 1);  $I_R$  is the radiation transferred into the snow layer (W-m-2); and  $Q_R$  is the energy 220 221 delivered by rainfall (W-m-2), which is only considered for the top snow layer. The solar 222 radiation transfer in the snow layers and the snow albedo are simulated using the 223 SNICAR model, which is solved using the method developed by Toon et al. (1989). Eq. 224 (4) is solved using an implicit centered finite difference method, and a Crank-Nicholson 225 scheme is employed.

The mass balance of the snow layer is described as follows (Bartelt and Lehnin, 2002):

227
$$\frac{\partial \rho_i \theta_i}{\partial t} + M_{iv} + M_{il} = 0$$
(5)

228
$$\frac{\partial \rho_{l} \theta_{l}}{\partial t} + \frac{\partial U_{l}}{\partial z} + M_{lv} - M_{il} = 0$$
(6)

where  $\rho_l$  is the density of the liquid water (kg -m-3);  $\theta_l$  is the volumetric liquid water content;  $U_l$  is the liquid water flux (kg -m-2 s-1);  $M_{iv}$  is the mass of ice that changes into vapour within a time step (kg -m-3 s-1);  $M_{il}$  is the mass of ice that changes into liquid water within a time step (kg -m-3 s-1); and  $M_{lv}$  is the mass of liquid water that changes into vapour within a time step (kg -m-3 s-1). The liquid water flux of the snow layer is calculated as follows (Jordan, 1991):

$$U_{l} = \frac{k}{\mu_{l}} \rho_{l}^{2} g \tag{7}$$

where *k* is the hydraulic permeability (m2),  $\mu_l$  is dynamic viscosity of water at 0  $^{\circ}$ (1.787-10-3 N - s -m-2),  $\rho_l$  is the density of liquid water (kg -m-3) and *g* is gravitational acceleration (m -s-2). The water flux of the bottom snow layer is considered snowmelt runoff.

**240 (3) Soil freezing and thawing**

The energy balance of the soil layer is solved as follows (Flerchinger and Saxton,1989):

243
$$C_{s}\frac{\partial T}{\partial t} - \rho_{i}L_{f}\frac{\partial \theta_{i}}{\partial t} - \frac{\partial}{\partial z}(\lambda_{s}\frac{\partial T}{\partial z}) + \rho_{i}c_{l}\frac{\partial q_{l}T}{\partial z} = 0$$
(8)

where  $C_s$  is the volumetric soil heat capacity (J -m-3K-1); T is the temperature (K) of 244 the soil layers; z is the vertical depth of the soil (m);  $\theta_i$  is the volumetric ice content; 245  $\rho_i$  is the density of ice (kg-m-3);  $\lambda_s$  is the thermal conductivity (W-m-1K-1);  $\rho_l$  is 246 the density of liquid water (kg-m-3); and  $c_1$  is the specific heat of liquid water 247  $(J kg^{-1}K^{-1})$ . In addition,  $q_l$  is the water flux between different soil layers (m s-1) and is 248 249 solved using the 1-D vertical Richards equation. The unsaturated soil hydraulic 250 conductivity is calculated using the modified van Genuchten's equation (Wang et al., 251 2010), as follows:

252
$$K = f_{i c e} K_{s a t} \left(\frac{\theta_{l} - \theta_{r}}{\theta_{s} - \theta_{r}}\right)^{1/2} \left[1 - \left(1 - \left(\frac{\theta_{l} - \theta_{r}}{\theta_{s} - \theta_{r}}\right)^{-1/m}\right)^{m}\right]^{2}$$
(9)

where *K* is the unsaturated soil hydraulic conductivity (m s-1);  $K_{sat}$  is the saturated soil hydraulic conductivity (m s-1);  $\theta_l$  is the volumetric liquid water content;  $\theta_s$  is the saturated water content;  $\theta_r$  is the residual water content; *m* is an empirical parameter in van Genuchten's equation and  $f_{ice}$  is an empirical hydraulic conductivity reduction factor that is calculated using soil temperature as follows (Wang et al., 2010):

258
$$f_{ice} = \exp[-10(T_f - T_{soil})], \quad 0.05 \le fice \le 1$$
(10)

where  $T_f$  is 273.15 K and  $T_{soil}$  is the soil temperature.

**260 Eq. (8) solves the soil temperature with the upper boundary condition as the heat flux**

into the uppermost soil layer. When the ground is not covered by snow, the heat flux

Comment [PDM44]: "Equation"

262 from the atmosphere into the uppermost soil layer is expressed as follows (Oleson et263 al., 2010):

$$h = S_g + L_g - H_g - \lambda E_g + Q_R \tag{11}$$

where *h* is the upper boundary heat flux into the soil layer (W ·m-2);  $S_g$  is the solar radiation absorbed by the uppermost soil layer (W ·m-2);  $L_g$  is the net long wave radiation absorbed by the ground (W ·m-2),  $H_g$  is the sensible heat flux from the ground (W ·m-2);  $\lambda E_g$  is the latent heat flux from the ground (W ·m-2); and  $Q_R$  is the energy delivered by rainfall (W ·m-2). When the ground is covered by snow, the heat flux into the uppermost soil layer is calculated as follows:

$$h = I_p + G \tag{12}$$

where  $I_p$  is the radiation that penetrates the snow cover, and *G* is the heat conduction from the bottom snow layer to the uppermost soil layer. Eq (8) is solved using a finite difference scheme with an hourly time step, similar to the solution of Eq (4).

275 There are no available observations of the geothermal heat flux for the northeastern

276 QTP. To simulate the permafrost we consider an underground depth of 50 m. We assume 277 an upward thermal heat flux at the bottom boundary and estimate its value to be 0.14 278 W  $\cdot m^{-2}$  at a depth of 50 m using the average geothermal gradient from the 4 boreholes 279 (T1-T4) shown in Figure 3, which is reasonable based on a comparison with the 280 observations (0.02 W  $\cdot m^{-2}$  to 0.16 W  $\cdot m^{-2}$ ) from the interior of the QTP (Wu et al., 2010). 281 The vertical soil column is divided into 39 layers in the model (see Figure 2). The 1.7 282 m topsoil layer is subdivided into 9 layers. The first layer is 0.05 m, and the soil layer

thickness increases with depth linearly from 0.05 m to 0.3 m at a depth of 0.8 m and

**Comment [PDM45]:** "simulate permafrost we"

Comment [PDM46]: " soil " Comment [PDM47]: "at the surface Comment [PDM48]: delete Comment [PDM49]: "thicknesses increse **284** 
[revised manuscript text omitted]

**Comment [PDM86]:** "challenging, b is part of our ongoing research."

Comment [PDM87]: "ground"

Comment [PDM88]: A trend of

Comment [PDM89]: "of"

increasing

**594 This work carefully validated a distributed hydrological model coupled with 595 cryospheric processes in the upper Heihe River basin using available observations of soil moisture, soil temperature, frozen depth, actual evaporation and streamflow 596 discharge. Based on the model simulations from 1971 to 2013 in the upper Heihe River, 597 598 the long-term changes in frozen soils were investigated, and the effects of the frozen 599 soil changes on the hydrological processes were explored. Based on these analyses, we 600 have reached the following conclusions: 601 (1) The model simulation suggests that 8.8% of the permafrost areas degraded into 602 seasonally frozen grounds in the upper Heihe River basin during 1971-2013, 603 predominantly between elevations of 3500 m and 3900 m. The results indicate that the 604 decreasing trend of the annual maximum frozen depth of the seasonally frozen ground is 0.032 m decade-1, which is consistent with previous observation-based studies at the 605 606 plot scale. Additionally, our work indicates that the increasing trend of active layer thickness in the permafrost regions is 0.043 m-decade-1. 607 608 (2) The model-simulated runoff trends agree with the observed trends. In the freezing 609 season (November-March), based on the model simulation, runoff was mainly sourced 610 from subsurface flow, which increased significantly in the higher elevation regions 611 where significant frozen soil changes occurred. This finding implies that the runoff increase in the freezing season is primarily caused by frozen soil changes (permafrost 612**

613 degradation and reduced seasonally frozen depth). In the thawing season (April-

614 October), the model simulation indicates that runoff was mainly sourced from rainfall 615 and showed an increasing trend at higher elevations, which can be explained by the 616 increase in precipitation. In both the freezing and thawing seasons, the model-simulated runoff decreased in the lower-elevation regions, which can be explained by increased 617 618 evaporation due to rising air temperatures.

619 (3) The model-simulated changes in soil moisture and ground temperature indicate 620 that the annual storage of liquid water increased, especially in the most recent three 621 decades, due to frozen soil changes. The annual ice water storage in the top 0-3 m of 622 soil showed a significant decreasing trend due to soil warming. The model simulated 623 annual groundwater storage had an increasing trend, which is consistent with the 624 changes observed by the GRACE satellite. Therefore, groundwater recharge in the 625 upper Heihe basin has increased in recent decades.

626 (4) The model simulation indicated that regions where permafrost changed into 627 seasonally frozen ground had larger changes in runoff and soil moisture than the areas 628 covered by seasonally frozen ground throughout the study period.

629 For a better understanding of the changes in frozen soils and their impact on 630 ecohydrology, the interactions among soil freezing-thawing processes, vegetation 631 dynamics and hydrological processes need to be investigated in future studies. There 632 are uncertainties in simulations of frozen soils and hydrological processes that also 633 warrant further investigation in the future.

Comment [PDM90]: Delete

634

Acknowledgements: This research was supported by the major plan of "Integrated

Comment [PDM91]: At your discretion, I recommend acknowledging that suggestions from reviewers have substantially improved the paper.

[revised manuscript text omitted]

---

## Author Response (AR3)

**1. Comments from Editor**

**Comments:**

1. Line 21: change "especially" to "particularly".
2. Line 22: change "distributed physically based" to "distributed, physically-based".
3. Line 23: change "the long-term (from 1971 to 2013)" to "long-term (1971-2013)".
4. Line 24-25: Delete "which is located on the".
5. Line 25: Delete "carefully".
6. Line 26: Delete "the".
7. Line 27: change "the" to "spatio-temporal".
8. Line 27: delete "the" before "hydrology".
9. Line 27: change "the results" to "our results".
10. Line 28: change "showed" to "show".
11. Line 28: change "permafrost area" to "area with permafrost".
12. Line 28-29: change "especially" to "predominantly". Delete "frozen".
13. Line 33: change "the" to "an".
14. Line 58: Add "(QTP)".
15. Line 58: change "Intensive" to "Those studies that include intensive".
16. Line 60: "(e.g., Cite an example or two).
17. Line 60-61: change "have not been" to "are not typically". change "available" to "available in many areas".
18. Line 66: "As an alternat strategy, hydrological".
19. Line 69: change "which could" to " "but these cann not".
20. Line 77: change "for" to "to".
21. Line 82: change "Qinghai-Tibetan Plateau (QTP)" to "(QTP)".
22. Line 93: Delete "the".
23. Line 96: Delete "the".
24. Line 105: Delete "solid".
25. Line 107: Change "basin located on the" to "basin,".
26. Line 108: change "On the basis of previous studies" to "Specifically". Also, append to the end of the previous paragraph.
27. Line 115: change "of " to "ranging from".
28. Line 121: delete ",".
29. Line 122: delete ",".
30. Line 122-123: change "," to ";" change "i.e.," to "one at".
31. Line 124-125: change "and" to "and the other at". Delete "see".
32. Line 132: change "by" to "from".
33. Line 137: Add "(LAI)" after "index".
34. Line 140: change "leaf area index (LAI)" to "LAI".
35. Line 155: Add "a" before "lack". Change "measurement" to "measurements"
36. Line 169-170: change "and the study catchment" to "which". change "a" to "each".
37. Line 173-174: change to "Additional hillslope properties include soil and vegetation types".
38. Line 260: change "Eq" to "Equation".

39. Line 276: change "simulate the permafrost we consider an underground depth of 50 m.We" to "simulate permafrost we".

40. Line 282: change "the first layer" to " the first soil layer at the surface". Delete "soil" before "layer".

41. Line 283-284: change "thickness increases" to "thicknesses increase". change "m. and then decreases linearly with depth to" "m. Then thickness decreases linearly with depth reaching".

42. Line 284-285: change "There" to "From 1.7 m to 3 m depth, there". change "from 1.7 m to 3.0 m to try to replicate" to "to try to capture".

43. Line 309: change "to" to "in order to".

44. Line 355: add "," after "depth".

45. Line 368-371: Line is needed in methods section to describe how simulated versus observed discharge are compared or assessed.

46. Line 371: delete "see".

47. Line 410-415: This text should be moved to Methods.

48. Line 431-432: Delete "the". Change "the development of taliks" to "that talik development".

49. Line 436: change "as shown in Figure 10(a)," to "(Figure 10(a))".

50. Line 437: change "decreased, and the ground temperature in the deep layer (with" to "decreased and the ground temperature in deep layers (depths…"

51. Line 441: add "surface". change "frozen depths" to "depth to the base of permafrost"

52. Line 459: delete "the"

53. Line 466:change "increasing trend of the" to ""increase in"

54. Line 468: change "annual water storage in the top 0-3 m layer" to "annual soil liquid water storage (0-3 m)".

55. Line 469-450: delete "the". Add "soil" before "water". Delete "of the top 0-3 m".

56. Line 476: change "the frozen soil" to "permafrost"

57. Line 483: change "as shown in Figure 10(a)." to "(Figure 10(a))".

58. Line 485: change "as shown in Figure 10(b)." to "(Figure 10(b))".

59. Line 488: delete "the"

60. Line 490-491: change "increased" to "increase in". delete "the".

61. Line 497: add "area of" before "seasonally".

62. Line 498-499: change "runoff in the permafrost area was lower than in the seasonally frozen ground" to "the inverse was true""

63. Line 501: change "1980 and that changed" to "1980, but degraded"

64. Line 502: delete "the".

65. Line 503: delete "the" before "runoff" and add "hydrological" before "recession".

66. Line 510: change "larger" to "large".

67. Line 583-584: change to "In areas with high groundwater flow rates, laterally advected heat flux may increase the thawing of permafrost".

68. Line 592: change "challenging. This work will be done in a future study." to "challenging, but is part of our ongoing research.".

69. Line 602: change "grounds" to "ground".

70. Line 606-607: change "that the increasing trend of" to "A trend of increasing". change "is" to "of".

71. Line 633: delete "in the future".

72. Line 635: At your discretion, I recommend acknowledging that suggestions from reviewers have substantially improved the paper.

73. Line 638: change "for this paper are properly cited and referred in the reference list" to "cited in this paper are available from the references".

74. Figure 1. Change color of boreholes sites to black so the they are immediately visible. Figure looks great!

75. Figure 6. Not sure why the figure looks like this after I exported to Microsoft Word, but anyway, please change "(m3/s to "(m3·s-1)" in axis titles.

76. Figure 7. Please change caption to "…"October). Results from linear regressions are indicated."

77. Figure 8. Please change caption to "…"permafrost. Results from linear regressions are indicated."

78. Figure 11. In order to better separate the two pairs of panels, I suggest moving the "(a) Freezing season" and "(b Thawing season" titles outside of the panels. This will provide a bit of further separation between the pairs. Change caption to "…Data and regression results are shown. The upper…"

79. Figure 12. You use "groundwater" in some places, and "ground water" in others. Please be consistent and make appropriate change. Please change caption to "…"water. "Resu from linear regressions are indicated."

80. Figure 13. Caption: Change "changes" to "showing changes".

81. Table1. Add "." After "1".

82. Table2. Add "." After "2".component (mm·y-1)". If you make this insertion, you can delete "(mm/yr)" from each of the column headers.

**2. Author's responses**

Thanks for handling our manuscript and the detail suggestions to revise the manuscript. Following comments from the editor, we have revised our manuscript. The details are given bellow.

**Comments:**

**1.** Line 21: change "especially" to "particularly".

**Reply:** We have revised as suggested (Please see line 21 in the revised clean version manuscript).

**2.** Line 22: change "distributed physically based" to "distributed, physically-based".

**Reply:** We have revised as suggested (Please see line 22 in the revised clean version manuscript).

**3.** Line 23: change "the long-term (from 1971 to 2013)" to "long-term (1971-2013)".

**Reply:** We have revised as suggested (Please see line 23 in the revised clean version manuscript).

**4.** Line 24-25: Delete "which is located on the".

**Reply:** We have revised as suggested (Please see line 24 in the revised clean version manuscript).

**5.** Line 25: Delete "carefully".

**Reply:** We have revised as suggested (Please see line 25 in the revised clean version manuscript).

**6.** Line 26: Delete "the".

**Reply:** We have revised as suggested (Please see line 26 in the revised clean version manuscript).

**7.** Line 27: change "the" to "spatio-temporal".

Reply: We have revised as suggested (Please see line 26 in the revised clean version manuscript).

**8.** Line 27: delete "the" before "hydrology".

**Reply:** We have revised as suggested (Please see line 27 in the revised clean version manuscript).

**9.** Line 27: change "the results" to "our results".

**Reply:** We have revised as suggested (Please see line 27 in the revised clean version manuscript).

**10.** Line 28: change "showed" to "show".

**Reply:** We have revised as suggested (Please see line 27 in the revised clean version manuscript).

**11.** Line 28: change "permafrost area" to "area with permafrost".

Reply: We have revised as suggested (Please see line 27 in the revised clean version manuscript).

**12.** Line 28-29: change "especially" to "predominantly". Delete "frozen".

**Reply:** We have revised as suggested (Please see line 28-29 in the revised clean version manuscript).

**13.** Line 33: change "the" to "an".

**Reply:** We have revised as suggested (Please see line 32 in the revised clean version manuscript).

**14.** Line 58: Add "(QTP)".

**Reply:** We have revised as suggested (Please see line 57 in the revised clean version manuscript).

**15.** Line 58: change "Intensive" to "Those studies that include intensive".

**Reply:** We have revised as suggested (Please see line 58 in the revised clean version manuscript).

**16.** Line 60: "(e.g., Cite an example or two).

**Reply:** We have revised as "(e.g., Cheng and Wu, 2007; Wu et al., 2010)" (Please see line 59-60 in the revised clean version manuscript).

**17.** Line 60-61: change "have not been" to "are not typically". change "available" to "available in many areas".

**Reply:** We have revised as suggested (Please see line 60-61 in the revised clean version manuscript).

**18.** Line 66: "As an alternat strategy, hydrological".

Reply: We have revised as suggested (Please see line 66 in the revised clean version manuscript).

**19.** Line 69: change "which could" to " "but these cann not".

**Reply:** We have revised as suggested (Please see line 69 in the revised clean version manuscript).

**20.** Line 77: change "for" to "to".

**Reply:** We have revised as suggested (Please see line 77 in the revised clean version manuscript).

**21.** Line 82: change "Qinghai-Tibetan Plateau (QTP)" to "(QTP)".

**Reply:** We have revised as suggested (Please see line 82 in the revised clean version manuscript).

**22.** Line 93: Delete "the".

**Reply:** We have revised as suggested (Please see line 92 in the revised clean version manuscript).

**23.** Line 96: Delete "the".

**Reply:** We have revised as suggested (Please see line 95 in the revised clean version manuscript).

**24.** Line 105: Delete "solid".

**Reply:** We have revised as suggested (Please see line 104 in the revised clean version manuscript).

**25.** Line 107: Change "basin located on the" to "basin,".

**Reply:** We have revised as suggested (Please see line 106 in the revised clean version manuscript).

**26.** Line 108: change "On the basis of previous studies" to "Specifically". Also, append to the end of the previous paragraph.

**Reply:** We have revised as suggested (Please see line 106 in the revised clean version manuscript).

27. Line 115: change "of " to "ranging from".

**Reply:** We have revised as suggested (Please see line 113 in the revised clean version manuscript).

28. Line 121: delete ",".

**Reply:** We have revised as suggested (Please see line 119 in the revised clean version manuscript).

29. Line 122: delete ",".

**Reply:** We have revised as suggested (Please see line 120 in the revised clean version manuscript).

30. Line 122-123: change "," to ";" change "i.e.," to "one at".

**Reply:** We have revised as suggested (Please see line 120-121 in the revised clean version manuscript).

31. Line 124-125: change "and" to "and the other at". Delete "see".

**Reply:** We have revised as suggested (Please see line 121-123 in the revised clean version manuscript).

32. Line 132: change "by" to "from".

**Reply:** We have revised as suggested (Please see line 130 in the revised clean version manuscript).

33. Line 137: Add "(LAI)" after "index".

**Reply:** We have revised as suggested (Please see line 135 in the revised clean version manuscript).

34. Line 140: change "leaf area index (LAI)" to "LAI".

**Reply:** We have revised as suggested (Please see line 138 in the revised clean version manuscript).

35. Line 155: Add "a" before "lack". Change "measurement" to "measurements"

**Reply:** We have revised as suggested (Please see line 152 in the revised clean version manuscript).

36. Line 169-170: change "and the study catchment" to "which". change "a" to "each".

**Reply:** We have revised as suggested (Please see line 166 in the revised clean version manuscript).

37. Line 173-174: change to "Additional hillslope properties include soil and vegetation types".

**Reply:** We have revised as suggested (Please see line 170 in the revised clean version manuscript).

38. Line 260: change "Eq" to "Equation".

**Reply:** We have revised as suggested (Please see line 256 in the revised clean version manuscript).

39. Line 276: change "simulate the permafrost we consider an underground depth of 50 m.We" to "simulate permafrost we".

**Reply:** We have revised as suggested (Please see line 272 in the revised clean version manuscript).

40. Line 282: change "the first layer" to " the first soil layer at the surface". Delete "soil" before "layer".

**Reply:** We have revised as suggested (Please see line 278 in the revised clean version manuscript).

**41.** Line 283-284: change "thickness increases" to "thicknesses increase". change "m. and then decreases linearly with depth to" "m. Then thickness decreases linearly with depth reaching".

**Reply:** We have revised as suggested (Please see line 278-280 in the revised clean version manuscript).

**42.** Line 284-285: change "There" to "From 1.7 m to 3 m depth, there". change "from 1.7 m to 3.0 m to try to replicate" to "to try to capture".

**Reply:** We have revised as suggested (Please see line 280-281 in the revised clean version manuscript).

**43.** Line 309: change "to" to "in order to".

**Reply:** We have revised as suggested (Please see line 312 in the revised clean version manuscript).

**44.** Line 355: add "," after "depth".

**Reply:** We have revised as suggested (Please see line 358 in the revised clean version manuscript).

**45.** Line 368-371: Line is needed in methods section to describe how simulated versus observed discharge are compared or assessed.

Reply: We have added a sentence in the section 3.3 as "The Nash-Sutcliffe efficiency and relative error are calculated using observed and simulated discharge to evaluate the model performance." (Please see line 306-307 in the revised clean version manuscript).

**46.** Line 371: delete "see".

**Reply:** We have revised as suggested (Please see line 374 in the revised clean version manuscript).

**47.** Line 410-415: This text should be moved to Methods.

**Reply:** We have moved it to section 3.2 (Please see line 286-291 in the revised clean version manuscript).

**48.** Line 431-432: Delete "the". Change "the development of taliks" to "that talik development".

**Reply:** We have revised as suggested (Please see line 429-430 in the revised clean version manuscript).

**49.** Line 436: change "as shown in Figure 10(a)," to "(Figure 10(a))".

**Reply:** We have revised as suggested (Please see line 434 in the revised clean version manuscript).

**50.** Line 437: change "decreased, and the ground temperature in the deep layer (with" to "decreased and the ground temperature in deep layers (depths…".

**Reply:** We have revised as suggested (Please see line 435 in the revised clean version manuscript).

**51.** Line 441: add "surface". change "frozen depths" to "depth to the base of permafrost".

**Reply:** We have revised as suggested (Please see line 439-440 in the revised clean version manuscript).

**52.** Line 459: delete "the".

**Reply:** We have revised as suggested (Please see line 457 in the revised clean version manuscript).

**53.** Line 466:change "increasing trend of the" to ""increase in".

**Reply:** We have revised as suggested (Please see line 464 in the revised clean version manuscript).

**54.** Line 468: change "annual water storage in the top 0-3 m layer" to "annual soil liquid water storage (0-3 m)".

**Reply:** We have revised as suggested (Please see line 466-467 in the revised clean version manuscript).

**55.** Line 469-450: delete "the". Add "soil" before "water". Delete "of the top 0-3 m".

**Reply:** We have revised as suggested (Please see line 467 in the revised clean version manuscript).

**56.** Line 476: change "the frozen soil" to "permafrost".

**Reply:** We have revised as suggested (Please see line 474 in the revised clean version manuscript).

**57.** Line 483: change "as shown in Figure 10(a)." to "(Figure 10(a))".

**Reply:** We have revised as suggested (Please see line 481 in the revised clean version manuscript).

**58.** Line 485: change "as shown in Figure 10(b)." to "(Figure 10(b))".

**Reply:** We have revised as suggested (Please see line 483 in the revised clean version manuscript).

**59.** Line 488: delete "the".

**Reply:** We have revised as suggested (Please see line 485 in the revised clean version manuscript).

**60.** Line 490-491: change "increased" to "increase in". delete "the".

**Reply:** We have revised as suggested (Please see line 487-488 in the revised clean version manuscript).

**61.** Line 497: add "area of" before "seasonally".

**Reply:** We have revised as suggested (Please see line 494 in the revised clean version manuscript).

**62.** Line 498-499: change "runoff in the permafrost area was lower than in the seasonally frozen ground" to "the inverse was true".

**Reply:** We have revised as suggested (Please see line 495 in the revised clean version manuscript).

**63.** Line 501: change "1980 and that changed" to "1980, but degraded".

**Reply:** We have revised as suggested (Please see line 497 in the revised clean version manuscript).

**64.** Line 502: delete "the".

**Reply:** We have revised as suggested (Please see line 499 in the revised clean version manuscript).

**65.** Line 503: delete "the" before "runoff" and add "hydrological" before "recession".

**Reply:** We have revised as suggested (Please see line 499 in the revised clean version manuscript).

**66.** Line 510: change "larger" to "large".

**Reply:** We have revised as suggested (Please see line 507 in the revised clean version manuscript).

**67.** Line 583-584: change to "In areas with high groundwater flow rates, laterally advected heat flux may increase the thawing of permafrost".

**Reply:** We have revised as suggested (Please see line 580-581 in the revised clean version manuscript).

**68.** Line 592: change "challenging. This work will be done in a future study." to "challenging, but is part of our ongoing research.".

**Reply:** We have revised as suggested (Please see line 589 in the revised clean version manuscript).

**69.** Line 602: change "grounds" to "ground".

**Reply:** We have revised as suggested (Please see line 599 in the revised clean version manuscript).

**70.** Line 606-607: change "that the increasing trend of" to "A trend of increasing". change "is" to "of".

**Reply:** We have revised as suggested (Please see line 603-604 in the revised clean version manuscript).

**71.** Line 633: delete "in the future".

**Reply:** We have revised as suggested (Please see line 630 in the revised clean version manuscript).

**72.** Line 635: At your discretion, I recommend acknowledging that suggestions from reviewers have substantially improved the paper.

Reply: We have added a sentence "The authors would like to thank the editor and reviewers for their constructive suggestions, which have substantially improved the paper." In the manuscript (Please see line 635-636 in the revised clean version manuscript).

**73.** Line 638: change "for this paper are properly cited and referred in the reference list" to "cited in this paper are available from the references".

**Reply:** We have revised as suggested (Please see line 637 in the revised clean version manuscript).

**74.** Figure 1. Change color of boreholes sites to black so the they are immediately visible. Figure looks great!

**Reply:** We have revised this figure as suggested (Please see Figure 1 in the revised clean version manuscript).

**75.** Figure 6. Not sure why the figure looks like this after I exported to Microsoft Word, but anyway, please change "(m3/s to "(m3·s-1)" in axis titles.

Reply: We have revised this figure as suggested (Please see Figure 6 in the revised clean version manuscript).

**76.** Figure 7. Please change caption to "… "October). Results from linear regressions are indicated."

**Reply:** We have revised this figure as suggested (Please see Figure 7 in the revised clean version manuscript).

**77.** Figure 8. Please change caption to "… "permafrost. Results from linear regressions are indicated."

**Reply:** We have revised this figure as suggested (Please see Figure 8 in the revised clean version manuscript).

**78.** Figure 11. In order to better separate the two pairs of panels, I suggest moving the "(a) Freezing season" and "(b Thawing season" titles outside of the panels. This will provide a bit of further separation between the pairs. Change caption to "…Data and regression results are shown. The upper…".

Reply: We have revised this figure as suggested (Please see Figure 11 in the revised clean version manuscript).

**79.** Figure 12. You use "groundwater" in some places, and "ground water" in others. Please be consistent and make appropriate change. Please change caption to "…"water. "Resu from linear regressions are indicated."

**Reply:** We have revised this figure as suggested (Please see Figure 12 in the revised clean version manuscript). And we use "groundwater" for the whole manuscript to keep consistency.

**80.** Figure 13. Caption: Change "changes" to "showing changes".

**Reply:** We have revised this figure as suggested (Please see Figure 13 in the revised clean version manuscript).

**81.** Table1. Add "." After "1".

**Reply:** We have revised as suggested (Please see Table 1 in the revised clean version manuscript).

**82.** Table2. Add "." After "2". "component (mm·y-1)". If you make this insertion, you can delete "(mm/yr)" from each of the column headers.

Reply: We have revised as suggested (Please see Table 2 in the revised clean version manuscript).

**3. Author's changes in manuscript**

1. Modified Figure 1, Figure 6 and Figure 11 as suggested.

2. Modified figure captions of Figure 7, Figure 8, Figure 11, Figure 12 and Figure 13 as suggested.

3. Modified Table 2 as suggested.

4. Modified all the texts as suggested.

[revised manuscript text omitted]

---

## Editor Decision (ED3)

Dear Dr. Gao,

I have read your revised manuscript entitled "Change in frozen soils and the effects on regional hydrology, upper Heihe Basin, northeastern Qinghai-Tibetan Plateau" (tc-2017-158). Thank you for your revisions. Please make the following edits before submitting for publication.

Best regards,

Peter

Minor comments:

Change title to reflect capitalization as indicated above.

P4, L66: Change "alternate" to "alternative".

Figure 8: "°C" in title of second Y-axis label in panel (a) does not match font in the rest of the figure, nor similar text in Figure7. Also, insert a space between "Temperature" and "(°C)".

Figure 9: In panels (d) and (e), insert a space between "Elevation" and "(m)", and between "permafrost" and "(%)".

Figure 11 caption, Line 934: "thawing season." Becomes "thawing season for the period of 1971 to 2013."